# STNADAM: STOCHASTIC TWO-TRACK NESTEROV-ACCELERATED ADAPTIVE MOMENTUM ESTIMATION

## ABSTRACT

We develop an enhanced version of the Adam algorithm for solving "nonconvex + weakly-convex" composite optimizations, termed Stochastic Two-track Nesterov-accelerated Adaptive Momentum Estimation (STNAdam). A featured difference from the existing accelerated variants of Adam is that STNAdam adopts a novel two-track iteration framework, which maintains two intertwined iteration trajectories including an extrapolation track and a regular update track, governed by Nesterov momentum and Adam-style adaptive conditioning interactively. It aims to promote the formation of a larger update neighborhood, while exploring a better iteration direction continuously. The stochastic gradient in STNAdam is allowed to be provided by arbitrary a variance-reduced gradient estimator, such as SVRG, SAGA and SARAH. The internal hyper-parameters generated along with this can be dynamically scheduled within some iterate-dependent finite intervals. Under the Kurdyka-Łojasiewicz property, we show that the sequence generated by STNAdam almost surely converges to a stationary point of the original problem at an explicit rate. Empirical results on low-light image enhancement are presented to demonstrate the superior performance of our proposed method.

## 1 INTRODUCTION

In recent years, machine learning has achieved remarkable success across various fields, such as computer vision (He et al., 2016; Mozaffari, 2025), natural language processing (Lauriola et al., 2022), and quantitative finance (Su et al., 2017). Many achievements are closely tied to the Adam algorithm and its accelerated variants, which generally entail integrating acceleration techniques into Adam, such as NAdam (Dozat, 2016) and Adam$^+$ (Liu et al., 2020). However, Adam-based algorithms face significant challenges when handling massive network parameters and data sets, thereby prompting the development of stochastic variants of Adam for deep learning problems. For instance, Wang et al. (2019) developed a stochastic Adam variant (SAdam) tailored for strongly convex problems, while Le-Duc et al. (2024) extended SAdam to "nonconvex + convex" composite optimization scenarios. Zhao et al. (2021) further proposed the stochastic Nesterov-accelerated adaptive momentum estimation (SNAdam) algorithm for such composite optimization tasks.

Despite these advancements, several critical issues remain unresolved. The integration of Nesterov acceleration with adaptive learning rates introduces additional complexity, making parameter tuning challenging and often leading to poor generalization. Moreover, in high-dimensional and nonconvex settings, the stochastic nature of gradients can destabilize training dynamics, further degrading algorithm performance. Addressing these challenges requires a deeper exploration of the intricate interplay between adaptive learning rates, momentum, and stochasticity.

In this paper, we focus on developing an enhanced stochastic variant of Adam that can handle the complexities of modern deep learning tasks. More precisely, we consider such a "nonconvex + weakly-convex" composite optimization problem, formulated as

$$\min_{x \in \mathbb{R}^d} \quad \Phi(x) := \frac{1}{N} \sum_{i=1}^{N} f_i(x) + g(x), \tag{1}$$

where each $f_i(x)$ is Lipschitz smooth with modulus $L_i > 0$, and hence their average sum function, written as $f(x)$, is also Lipschitz smooth with modulus $L = \frac{1}{N} \sum_{i=1}^{N} L_i$ (possibly nonconvex);

$g(x)$ is proper, lower semicontinuous (l.s.c.) and weakly-convex with modulus $\tau > 0$, and hence proximal-friendly (possibly nonsmooth), e.g., $g(x) = \mathcal{I}_X(x)$ with $\mathcal{I}_X(\cdot)$ being the indicator function over a simple compact set $X \subseteq \mathbb{R}^d$, or $g(x)$ is some a sparse-induced function (e.g., MCP, SCAD, $\ell_{1/2}$-norm). Specially, if $g(x) \equiv 0$, (1) reduces to a classic distributed optimization problem.

## 1.1 RELATED WORK

In practical, numerous algorithms have been developed for problem (1) or its some special cases, based on the gradient descent method (Nemirovski et al., 2009; Bottou, 2010). Next, we review the related literatures from the following perspectives.

**Deterministic methods**: Ghadimi & Lan (2016) proposed the Nesterov Accelerated Gradient (NAG) method with a fixed step size, which incorporated future gradient weights (i.e., momentum) to generate more informed descent directions. Duchi et al. (2011) introduced adaptivity by scaling step sizes according to the $\ell_2$ norm of all historical gradients, yet leading to infinite gradient accumulation. To mitigate this issue, Tieleman & Hinton (2012) proposed RMSprop, which employed exponential decay for past squared gradients. Their subsequent work, the Adam algorithm (Kingma & Ba, 2014), further integrated momentum (via decaying averages) with RMSprop's adaptive step size mechanism and added bias correction to enhance stability. Dozat (2016) later developed the NAdam algorithm by incorporating Nesterov acceleration into Adam.

**Stochastic methods**: To enhance problem-solving capabilities in deep learning, stochastic gradient descent (SGD) approximations were developed (see Bottou (2010) and the references therein). Then, the stochastic variant of NAG (SNAG) based on stochastic momentum was proposed by Sutskever et al. (2013). Subsequently, to improve generalization and flexibly adjust parameters, several Adam variants with adaptive learning rates were proposed. For instance, Le-Duc et al. (2024) proposed SAdam based on strong convexity, which maintained a fast decay rate while controlling the step size. Reddi et al. (2019) incorporated Nesterov-acceleration technique into Adam, named SNAdam. Xie et al. (2024) further proposed the SAdan algorithm by implicitly computing the future gradient direction to enhance convergence while preserving stochasticity.

**Stochastic gradient variants**: To enhance the convergence rates of SGD algorithms, Driggs et al. (2021) proposed a variance-shrinking gradient estimator as an alternative to the standard SGD estimator, with convergence analysis verifying its variance reduction properties. For nonconvex optimization, several variance-reduced gradient estimators have been developed to drive gradient estimator variance toward zero through modified stochastic gradient directions. Representative methods include SAG (Schmidt et al., 2017), SVRG (Johnson & Zhang, 2013), SAGA (Defazio et al., 2014), SARAH (Ghadimi & Lan, 2012; Nguyen et al., 2017), and so on.

The above discussion covers the popular first-order algorithms and their variants, which have well-established convergence analysis and practical applications. However, existing algorithms still lack efficiency in solving the "nonconvex + weakly-convex" optimization like the form of (1), indicating the need for a more effective iterative framework.

## 1.2 OUR CONTRIBUTIONS

We develop an enhanced version of the Adam algorithm, termed the Stochastic Two-track Nesterov-accelerated Adaptive Momentum Estimation (STNAdam), for problem (1), which adopts a novel two-track iteration framework. Specifically, this paper has the three main contributions.

  (i) **Two-track coupled iteration**: Essentially, we employ Nesterov momentum and Adam-style adaptive conditioning to interactively generate an extrapolation iteration trajectory and a regular update one. This two-track approach attempts to promote the formation of a larger update neighborhood, while exploring a better iteration direction continuously than the single-track versions, such as SGD, SAdam and SNAdam.
  (ii) **General convergence result**: Under the Kurdyka-Łojasiewicz (KŁ) property, we establish almost-sure global convergence of STNAdam to a stationary point of problem (1). Notably, the stochastic gradient in STNAdam is allowed to be provided by arbitrary a variance-reduced gradient estimator, such as SVRG, SAGA, SARAH and SPIDER. And the internal hyper-parameters generated along with this can be dynamically scheduled within some

iterate-dependent finite intervals, removing hand-tuning. This is particularly important in reducing training time and improving generalization.

(iii) **Favorable practical performance**: Our STNAdam yields excellent performance on low-light image enhancement (LIE) tasks, compared to three single-track algorithms, including SGD (Bottou, 2010), SAdam (Kingma & Ba, 2014), and SNAdam (Xie et al., 2024), and five customized algorithms of LIE, including NPE (Fu et al., 2015), DeHz (Dong et al., 2011), LIME (Guo et al., 2017), Retinex-Net (Wei et al., 2018) and LR3M (Ren et al., 2020).

The rest of this paper is organized as follows. Section 2 introduces the STNAdam algorithm in detail. We give its global convergence analysis in Section 3. The empirical results are reported in Section 4. Concluding remarks are made in Section 5. The detailed proofs for the theoretical analysis, along with supplementary experimental results, are provided in the appendix.

## 2   THE PROPOSED METHOD

In this section, we propose an enhanced Adam algorithm for problem (1), termed STNAdam, and then provide adaptive update rules regarding stochastic gradient and hyper-parameters.

We first introduce some paired notations that are calculated using full gradient $\nabla f(x^k)$ and stochastic gradient $\widetilde{\nabla} f(x^k)$ at point $x^k$, respectively, in Table 1.

Table 1: The paired notations derived from full gradient and stochastic gradient, respectively.

| Name | Full gradient calculation | Stochastic gradient calculation |
|---|---|---|
| Momentum estimation (ME) | $m^{k+1} = \mu m^k + (1-\mu)\nabla f(x^k)$ | $\varpi^{k+1} \leftarrow \mu \varpi^k + (1-\mu)\widetilde{\nabla} f(x^k)$ |
| First-time ME correction | $\widehat{m}^{k+1} = \frac{1}{1-\mu^{k+1}} m^{k+1}$ | $\widehat{\varpi}^{k+1} \leftarrow \frac{1}{1-\mu^{k+1}} \varpi^{k+1}$ |
| Second-time ME correction | $\widetilde{m}^{k+1} = \gamma_{k+1}\widehat{m}^{k+1} + (1-\gamma_{k+1})\nabla f(x^k)$ | $\widetilde{\varpi}^{k+1} \leftarrow \gamma_{k+1}\widehat{\varpi}^{k+1} + (1-\gamma_{k+1})\widetilde{\nabla} f(x^k)$ |
| Adaptive learning rate (ALR) | $n_{k+1} = \nu n_k + (1-\nu)\|\nabla f(x^k)\|^2$ | $\pi_{k+1} \leftarrow \nu \pi_k + (1-\nu)\|\widetilde{\nabla} f(x^k)\|^2$ |
| ALR correction | $\widehat{n}^{k+1} = \frac{1}{1-\nu^{k+1}} n^{k+1}$ | $\widehat{\pi}^{k+1} \leftarrow \frac{1}{1-\nu^{k+1}} \pi^{k+1}$ |

It follows from Table 1 that we have

$$\left\|\widehat{m}^{k+1} - m^{k+1}\right\| = \frac{\mu^{k+1}}{1-\mu^{k+1}} \left\|m^{k+1}\right\|; \quad \left\|\widehat{m}^{k} - m^{k}\right\| = \frac{\mu^{k}}{1-\mu^{k}} \left\|m^{k}\right\|. \tag{2}$$

Further, if $\mu \in \left(0, \frac{1}{\sqrt{2}}\right)$, there holds $\frac{\sqrt{2}\mu^{k+1}}{1-\mu^{k+1}} < \frac{\mu^k}{1-\mu^k}$ for any $k \geq 1$.

Next, to be more intelligible to different iterative ideas, we give an iterative trajectory comparison by means of the notations under full gradient, reported in Figure 1.

- **NAG**: Start from $x^k$ along $m^{k+1}$ to get $x^{k+1}$, shown by the black line in Fig. 1(a);

- **Adam**: Employ $\widehat{m}^{k+1}$ to get $x^{k+1}$ in a similar way, shown by the blue line in Fig. 1(b);

- **NAdam**: Use $\widetilde{m}^{k+1}$ to get $x^{k+1}$, shown by the orange line in Fig. 1(c);

- **TNAdam**: Implement two-track iteration along $\widehat{m}^{k+1}$ and $\widetilde{m}^{k+1}$ from iteration point $x^k$ and extrapolation point $\overline{x}^{k+1} = \lambda_{k+1}x^k + (1-\lambda_{k+1})\widetilde{x}^k$ to get $x^{k+1}$ and $\widetilde{x}^{k+1}$, respectively, shown by the red lines in Fig. 1(d).

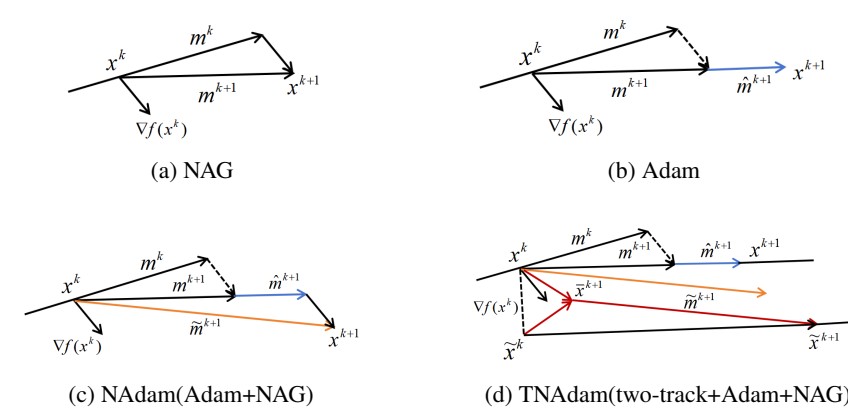

Figure 1: Iterative trajectory comparison of various algorithms.

One observes that distinct from single-track versions, TNAdam adopts a novel two-track iteration framework to promote the formation of a larger update neighborhood, while exploring a better iteration direction continuously. Then, after replacing the above symbols with the corresponding ones under stochastic gradient in Table 1, we obtain the STNAdam algorithm, described in Algorithm 1.

---

**Algorithm 1** Our STNAdam algorithm for problem (1)

---

**Input**: Initialize $x^0$ randomly; Set $\widetilde{x}^0 = x^0$, $\varpi^0 = 0$, $\pi_0 = 0$, $k = 0$; $\mu \in \left(0, \frac{1}{\sqrt{2}}\right)$, $\nu \in [0, 1]$, $\alpha > 0$ and $\varepsilon > 0$.
**Output**: $\widetilde{x}^{k+1}$.

1: **while** a termination criterion is not met **do**
2:     Generate stochastic gradient $\widetilde{\nabla} f(x^k)$ by a variance-reduced gradient estimator;
3:     Randomly select weighted parameters $\gamma_{k+1}, \alpha_{k+1}, \lambda_{k+1}$ within some updated intervals;
4:     Calculate $\varpi^{k+1}, \widehat{\varpi}^{k+1}, \widetilde{\varpi}^{k+1}, \pi_{k+1}, \widehat{\pi}_{k+1}$ according to Table 1;
5:     The iteration update:

$$x^{k+1} \leftarrow \mathcal{P}_g \left(x^k, \widehat{\varpi}^{k+1}, \frac{\alpha}{\sqrt{\widehat{\pi}_{k+1}} + \varepsilon}\right);$$

$$\overline{x}^{k+1} \leftarrow \lambda_{k+1} x^k + (1 - \lambda_{k+1}) \widetilde{x}^k;$$

$$\widetilde{x}^{k+1} \leftarrow \mathcal{P}_g \left(\overline{x}^{k+1}, \widetilde{\varpi}^{k+1}, \frac{\alpha_{k+1}}{\sqrt{\widehat{\pi}_{k+1}} + \varepsilon}\right).$$

6:     Set $k \leftarrow k + 1$.
7: **end while**

---

**Remark 1.** (i) In Step 5, we define the proximal gradient operator (Ghadimi & Lan, 2016)

$$\mathcal{P}_g \left(x, y, t\right) = \arg \min_{u \in \mathbb{R}^d} \left\{g(u) + \langle y, u \rangle + \frac{1}{2t} \|u - x\|^2\right\};$$

(ii) The external termination criterion adopts the following condition

$$\|\widetilde{x}^{k+1} - \widetilde{x}^k\| \leq 10^{-6}.$$

In what follows, we will describe the details of the variance-reduced gradient estimator and the parameter update intervals, i.e., the two underlined parts in STNAdam.

**Variance-reduced gradient estimator**: The stochastic gradient $\widetilde{\nabla} f(x^k)$ is typically generated by employing partial elements $\{\nabla f_1(x^k), \cdots, \nabla f_N(x^k)\}$. The index set of employed elements (a.k.a.,

mini-batch) $B_k \subset \{1, \cdots, N\} = [N]$ is selected uniformly at random from all possible subsets of $[N]$ with fixed size $b(\ll N)$, e.g., the SGD estimator. It is well known that SGD does not exhibit variance reduction, whereas other widely used gradient estimators, such as SAGA and SARAH, possess this property. They can be mathematically formulated as follows:

- SGD (Bottou, 2010): $\widetilde{\nabla} f(x^k)_{\text{SGD}} = \frac{1}{b} \sum_{i \in B_k} \nabla f_i(x^k)$.
- SAGA (Defazio et al., 2014):

$$\widetilde{\nabla} f(x^k)_{\text{SAGA}} = \frac{1}{b} \sum_{i \in B_k} \left( \nabla f_i(x^k) - \nabla f_i(\varphi_i^k) \right) + \frac{1}{N} \sum_{j=1}^{N} \nabla f_j(\varphi_j^k),$$

  where $\varphi_i^{k+1} = x^k$ if $i \in B_k$; otherwise $\varphi_i^{k+1} = \varphi_i^k$.
- SARAH (Ghadimi & Lan, 2012; Nguyen et al., 2017):

$$\widetilde{\nabla} f(x^k)_{\text{SARAH}} = \begin{cases} \nabla f(x^k), & \text{with probability} \quad p \in (0, 1), \\ \frac{1}{b} \sum_{i \in B_k} \left( \nabla f_i(x^k) - \nabla f_i(x^{k-1}) \right) + \widetilde{\nabla} f(x^{k-1})_{\text{SARAH}}, & \text{otherwise.} \end{cases}$$

Together with the entries $m^{k+1}$ and $\varpi^{k+1}$ in Table 1, it is not hard to obtain the following update formulas of $\widehat{\varpi}^{k+1}$ with respect to $\widehat{m}_i^{k+1}$ under the SGD, SAGA, and SARAH gradient estimators, respectively. Similar formulas hold for $\widetilde{\varpi}^{k+1}$ with respect to $\widetilde{m}_i^{k+1}$, so we omit them.

- SGD: $\widehat{\varpi}_{\text{SGD}}^{k+1} = \frac{1}{b} \sum_{i \in B_k} \widehat{m}_i^{k+1}$.
- SAGA: $\widehat{\varpi}_{\text{SAGA}}^{k+1} = \frac{1}{b} \sum_{i \in B_k} \left( \widehat{m}_i^{k+1} - \widehat{w}_i^{k+1} \right) + \frac{1}{N} \sum_{j=1}^{N} \widehat{w}_j^{k+1}$,
  where $\widehat{w}_i^{k+1} = \widehat{m}_i^k$ if $i \in B_k$; otherwise $\widehat{w}_i^{k+1} = \widehat{w}_i^k$.
- SARAH: $\widehat{\varpi}_{\text{SARAH}}^{k+1} = \begin{cases} \widehat{m}^{k+1}, & \text{with probability} \quad p \in (0, 1), \\ \frac{1}{b} \sum_{i \in B_k} \left( \widehat{m}_i^{k+1} - \widehat{m}_i^k \right) + \widehat{\varpi}_{\text{SARAH}}^k, & \text{otherwise.} \end{cases}$

**Remark 2.** When STNAdam is equipped with a certain estimator, e.g., the SGD estimator, we call it STNAdam-SGD. Then, STNAdam-SAGA and STNAdam-SARAH are similar.

Finally, we review a technical lemma regarding variance-reduced gradient estimator, of which the proof is analogous to that presented in Bertsekas & Tsitsiklis (1989); Wang & Han (2023).

**Lemma 1.** *Let $\{x^k\}$ and $\{\widetilde{x}^k\}$ be the sequences generated by Algorithm 1 with some a gradient estimator. Then, this gradient estimator is called variance-reduced with nonnegative constants $V_1, V_2, V_\Upsilon$, and $\rho \in (0, 1]$ if the following conditions hold:*

(i) *[MSE bound] There exist the sequences of $\{\Upsilon_k\}$ and $\{\Gamma_k\}$ with $\Upsilon_k = \sum_{i=1}^n (v_i^k)^2$ and $\Gamma_k = \sum_{i=1}^n v_i^k$ for a random variable $v_i^k \in \mathbb{R}_+$, $i \in [n]$ such that*

$$\mathbb{E}_k \left[ \left\| \widehat{m}^{k+1} - \widehat{\varpi}^{k+1} \right\|^2 + \left\| \widetilde{m}^{k+1} - \widetilde{\varpi}^{k+1} \right\|^2 \right] \le \Upsilon_k + V_1 \left( \left\| \nabla f(x^k) - m^k \right\|^2 + \left\| x^k - x^{k-1} \right\|^2 \right), \quad (3)$$

$$\mathbb{E}_k \left[ \left\| \widehat{m}^{k+1} - \widehat{\varpi}^{k+1} \right\| + \left\| \widetilde{m}^{k+1} - \widetilde{\varpi}^{k+1} \right\| \right] \le \Gamma_k + V_2 \left( \left\| \nabla f(x^k) - m^k \right\| + \left\| x^k - x^{k-1} \right\| \right), \quad (4)$$

*where $\mathbb{E}_k(\cdot)$ denotes the expectation conditional on the first $k$ iterations.*

(ii) *[Geometric decay] The sequence $\{\Upsilon_k\}$ decays geometrically:*

$$\mathbb{E}_k[\Upsilon_{k+1}] \le (1 - \rho)\Upsilon_k + V_\Upsilon \left( \left\| \nabla f(x^k) - m^k \right\|^2 + \left\| x^k - x^{k-1} \right\|^2 \right). \quad (5)$$

(iii) *[Convergence of estimator] If $\{x^k\}$ satisfies*

$$\lim_{k \to \infty} \mathbb{E} \left\| \nabla f(x^k) - m^k \right\|^2 = 0;$$

$$\lim_{k \to \infty} \mathbb{E} \left\| x^k - x^{k-1} \right\|^2 = 0,$$

*then we have $\mathbb{E}[\Upsilon_k] \to 0$ and $\mathbb{E}[\Gamma_k] \to 0$, where $\mathbb{E}(\cdot)$ is the full expectation.*

**Adaptive Update of Parameters**: The parameters $\gamma_{k+1}, \alpha_{k+1}$ and $\lambda_{k+1}$ for any $k \geq 0$ in Algorithm 1 are randomly selected within the following updated intervals.

(i) The second-order decay factor $\gamma_{k+1}$ used in the entry $\widetilde{\varpi}^{k+1}$ of Table 1:

$$\gamma_{k+1} \in \left( \underline{\gamma}, \overline{\gamma} \right) \subseteq (0,1), \tag{6}$$

where $\underline{\gamma} = 1 - \dfrac{\sqrt{2}\sqrt{(1-\mu^k)^2 \left[ (1-2\mu^2)M - 4s(V_1 + V_\Upsilon/\rho) \right] - 4}}{16}, \overline{\gamma} = 1; V_1, V_2, V_\Upsilon \geq 0, \rho \in (0,1]$ are defined in Lemma 1; $M$ and $s$ are the parameters defined in (9).

(ii) The weighted parameter $\lambda_{k+1}$ in Step 5:

$$\lambda_{k+1} \in \left( \underline{\lambda}, \overline{\lambda} \right) \subseteq (0,1), \tag{7}$$

where $\underline{\lambda} = \dfrac{14 - \sqrt{6 - 10\delta}}{20}, \overline{\lambda} = \dfrac{14 + \sqrt{6 - 10\delta}}{20}$ and $\delta = \dfrac{16\underline{\alpha} \left( \tau + \dfrac{1}{2s} + \dfrac{sL^2}{2} + L \right)}{\sqrt{\widehat{\pi}_{k+1}} + \varepsilon}$.

(iii) The stepsize $\alpha_{k+1}$ in Step 5:

$$\alpha_{k+1} \in \left( \underline{\alpha}, \overline{\alpha} \right) \subseteq (0,1), \tag{8}$$

where $\underline{\alpha} = \dfrac{\alpha(\sqrt{\widehat{\pi}_{k+1}} + \varepsilon)}{2(\sqrt{\widehat{\pi}_{k+1}} + \varepsilon) + 2\alpha(\tau + L)}$ and $\overline{\alpha} = 1$.

**Remark 3.** It is easy to obtain that the lower bounds $\underline{\gamma}$ in (6) and $\underline{\lambda}$ in (7) exceed 0 and do not approach 0. And the lower bound $\underline{\alpha}$ in (8) can also hold this property, provided that the stepsize $\alpha \in (0,1)$ is fixed and the moduli $L$ and $\tau$ are appropriately increased if necessary.

## 3 CONVERGENCE ANALYSIS

In this section, we make the convergence analysis for STNAdam, i.e., Algorithm 1. For convenience, let $\{\theta^k\} = \left\{ \left( \widetilde{x}^k, x^k \right) \right\}$ be the sequence generated by STNAdam with variance-reduced gradient estimator, and $\Phi^k = \Phi(\theta^k) = \Phi(\widetilde{x}^k) + \Phi(x^k)$. Moreover, we make such a mild assumption.

**Assumption 1.** *The objective function $\Phi$ in (1) is coercive, namely, if $\|x\| \to +\infty$, $\Phi(x) \to +\infty$.*

Before proceeding, we define the following energy function sequence:

$$\begin{aligned}
G^k &\equiv G(\widetilde{x}^k, x^k, x^{k-1}) \\
&= \Phi^k + \frac{4s}{\rho} \Upsilon_k + \left( M - 8s \left( 2\underline{\gamma}^2 - 4\underline{\gamma} + 3 \right) \right) \left\| \nabla f(x^k) - m^k \right\|^2 + H \left\| x^k - x^{k-1} \right\|^2 \\
&\quad + \left( \frac{\sqrt{\widehat{\pi}_k} + \varepsilon}{2\overline{\alpha}} + \frac{\sqrt{\widehat{\pi}_k} + \varepsilon}{2\alpha} - \tau - \frac{1}{2s} \right) \left\| \widetilde{x}^k - x^k \right\|^2 + \left( \frac{D(\mu^k)^2}{(1-\mu^k)^2} - Z \right) \left\| m^k \right\|^2,
\end{aligned} \tag{9}$$

where $M, H, Z$ and $D$ are parameters within some certain intervals; $s > 0$ is a parameter of the inequality $ab \leq \dfrac{s}{2}a^2 + \dfrac{1}{2s}b^2$. Please refer to Lemma A.1 in Appendix for details.

> **Step 1.** We firstly establish a foundational lemma, estimating the expected decrease of the energy function sequence with increasing iterations.

**Lemma 2.** *Under Assumption 1, for any $k \geq 1$, we have*

(i) $\mathbb{E}_k \left[ G^{k+1} \right] \leq G^k - A_1 \left\| \widetilde{x}^{k+1} - \widetilde{x}^k \right\|^2 - A_2 \left\| x^{k+1} - x^k \right\|^2 - A_3 \left\| \widetilde{x}^{k+1} - x^k \right\|^2 - A_4 \left\| x^{k+1} - \widetilde{x}^k \right\|^2$

$$- A_5 \left\| \widetilde{x}^k - x^k \right\|^2 - A_6 \left\| \nabla f(x^k) - m^k \right\|^2 - A_7 \left\| x^k - x^{k-1} \right\|^2 - A_8 \left\| m^{k+1} \right\|^2, \tag{10}$$

*where each $A_i > 0$ is given in Appendix Lemma A.1.*

(ii) $\displaystyle\sum_{k=0}^{\infty} \mathbb{E} \left( \left\| \widetilde{x}^{k+1} - \widetilde{x}^k \right\|^2 + \left\| x^{k+1} - x^k \right\|^2 + \left\| \widetilde{x}^{k+1} - x^k \right\|^2 + \left\| x^{k+1} - \widetilde{x}^k \right\|^2 + \left\| \widetilde{x}^k - x^k \right\|^2 \right.$

$$\left. + \left\| \nabla f(x^k) - m^k \right\|^2 + \left\| x^k - x^{k-1} \right\|^2 + \left\| m^{k+1} \right\|^2 \right) < +\infty,$$

*and hence the sequence $\left\{ \mathbb{E} \left\| \widetilde{x}^{k+1} - \widetilde{x}^k \right\|^2 \right\}$ is summable.*

**Step 2.** Then, we derive some important properties for the subgradient of $\Phi^{k+1}$ and the set of accumulation points of $\{\theta^k\}$, defined by

$$\Omega := \{\hat{\theta} : \exists \ \{\theta^{k_l}\} \subseteq \{\theta^k\} \ \text{s.t.} \ \theta^{k_l} \to \hat{\theta} \ \text{as} \ l \to \infty\}.$$

**Lemma 3.** [Boundedness of subgradient] *For $k \geq 0$, define*

$$\omega_1^k = \nabla f(x^k) - \widehat{\varpi}^k - \frac{\sqrt{\widehat{\pi}_k} + \varepsilon}{\alpha}(x^k - x^{k-1}); \ \omega_2^k = \nabla f(\widetilde{x}^k) - \widetilde{\varpi}^k - \frac{\sqrt{\widehat{\pi}_k} + \varepsilon}{\alpha}(x^k - x^{k-1}).$$

*Then, under Assumption 1, we have $\left(\omega_1^{k+1}, \omega_2^{k+1}\right) \in \partial\Phi(\theta^{k+1})$, and there exists a $\varrho > 0$ such that*

$$\begin{aligned}
\mathbb{E}_k \left\|\omega^{k+1}\right\| &\leq \varrho \left(\mathbb{E}_k \left\|\widetilde{x}^{k+1} - \widetilde{x}^k\right\| + \mathbb{E}_k \left\|x^{k+1} - x^k\right\| + \mathbb{E}_k \left\|\widetilde{x}^{k+1} - x^k\right\| + \mathbb{E}_k \left\|x^{k+1} - \widetilde{x}^k\right\| \right. \\
&\quad \left. + \left\|\widetilde{x}^k - x^k\right\| + \left\|\nabla f(x^k) - m^k\right\| + \left\|x^k - x^{k-1}\right\| + \left\|m^{k+1}\right\| \right) + \Gamma_k.
\end{aligned} \tag{11}$$

**Lemma 4.** [Properties of $\Omega$] *Under Assumption 1, we have*

(1) $\sum_{k=1}^{\infty} \left\|\widetilde{x}^k - \widetilde{x}^{k-1}\right\|^2 < \infty$ *almost surely (a.s.), and* $\left\|\widetilde{x}^k - \widetilde{x}^{k-1}\right\| \to 0$ *a.s.;*

(2) $\mathbb{E}[\Phi(\theta^k)] \to \Phi^*$, *where* $\Phi^* \in [\Phi_0, \infty)$;    (3) $\mathbb{E}[\text{dist}(0, \partial\Phi(\theta^k))] \to 0$;

(4) $\Omega$ *is nonempty, and* $\mathbb{E}[\text{dist}(0, \partial\Phi(\theta^*))] = 0$, $\forall \theta^* \in \Omega$;    (5) $\text{dist}(\theta^k, \Omega) \to 0$ *a.s.;*

(6) $\Omega$ *is a.s. compact and connected;*    (7) $\mathbb{E}[\Phi(\theta^*)] = \Phi^*$, $\forall \theta^* \in \Omega$.

**Step 3.** Next, we show that the sequence $\{\widetilde{x}^k\}$ converges to a stationary point of problem (1) in expectation by means of the KŁ inequality.

**Lemma 5.** [KŁ inequality] *Suppose that $\Phi$ is semialgebraic with KŁ exponent $\vartheta \in [0, 1)$. If $\widetilde{x}^k$ is not a stationary point of $\Phi$ after a finite number of iterations, then there must exist a $l > 0$ and a nondegenerate concave function $\varphi$ such that*

$$\varphi'(\mathbb{E}[\Phi(\theta^k) - \Phi_k^*])\mathbb{E}[\text{dist}(0, \partial\Phi(\theta^k))] \geq 1, \ \forall k \geq l,$$

*where $\Phi_k^*$ is a nondecreasing sequence converging to $\mathbb{E}[\Phi(\theta^*)]$ for any $\theta^* \in \Omega$.*

Now, we are ready to establish the convergence of the sequence $\{\widetilde{x}^k\}$ in expectation.

**Theorem 1.** *Assume that the conditions of Lemma 5 hold. Then, there hold:*

(i) *Either $\widetilde{x}^k$ is a stationary point after a finite number of iterations or $\{\widetilde{x}^k\}$ satisfies the finite-length property in expectation:*

$$\sum_{k=0}^{\infty} \mathbb{E} \left\|\widetilde{x}^{k+1} - \widetilde{x}^k\right\| < \infty,$$

*and there exists an integer $l$ such that, for all $i > l$,*

$$\begin{aligned}
\sum_{k=l}^{i} &\mathbb{E} \left(\left\|\widetilde{x}^{k+1} - \widetilde{x}^k\right\| + \left\|x^{k+1} - x^k\right\| + \left\|\widetilde{x}^{k+1} - x^k\right\| + \left\|x^{k+1} - \widetilde{x}^k\right\| + \left\|\widetilde{x}^k - x^k\right\| \right. \\
&\left. + \left\|\nabla f(x^k) - m^k\right\| + \left\|x^k - x^{k-1}\right\| + \left\|m^{k+1}\right\|\right) \\
\leq &\sqrt{\mathbb{E} \left\|\widetilde{x}^l - \widetilde{x}^{l-1}\right\|^2} + \sqrt{\mathbb{E} \left\|x^l - x^{l-1}\right\|^2} + \sqrt{\mathbb{E} \left\|\widetilde{x}^l - x^{l-1}\right\|^2} + \sqrt{\mathbb{E} \left\|x^l - \widetilde{x}^{l-1}\right\|^2} \\
&+ \sqrt{\mathbb{E} \left\|\widetilde{x}^{l-1} - x^{l-1}\right\|^2} + \sqrt{\mathbb{E} \left\|\nabla f(x^{l-1}) - m^{l-1}\right\|^2} + \sqrt{\mathbb{E} \left\|x^{l-1} - x^{l-2}\right\|^2} \\
&+ \sqrt{\mathbb{E} \left\|m^l\right\|^2} + \frac{2\sqrt{n}}{K\rho} \sqrt{\mathbb{E}[\Upsilon_{l-1}]} + \frac{4K}{A} \triangle^{l,i+1},
\end{aligned} \tag{12}$$

*where $K = \varrho + \frac{2\sqrt{nV_\Upsilon}}{\rho}$ with $\varrho$ defined in Lemma 3; $A = \min_{i \in [8]} \{A_i\} > 0$, defined in Lemma 2;*

*$\triangle^{\overline{k}, \underline{k}} = \mathbb{E}[G^{\overline{k}} - \Phi_{\overline{k}}^*] - \mathbb{E}[G^{\underline{k}} - \Phi_{\underline{k}}^*]$ for any $\overline{k} \geq \underline{k} \in \mathbb{Z}_+$.*

(ii) *The sequence $\{\widetilde{x}^k\}$ converges to a stationary point of $\Phi$ in expectation.*

**Step 5.** Finally, we provide a general convergence rate of the sequences $\{\widetilde{x}^k\}$ in expectation.

Furthermore, we adopt the form of the desingularization function proposed by Robbins & Siegmund (1971), i.e., $\varphi(r) = ar^{1-\vartheta}$ (Robbins & Siegmund, 1971), where $a > 0$ and the KŁ exponent $\vartheta \in [0,1)$. Then, for some $C > 0$, we have

$$\left(\mathbb{E}[\Phi(\theta^k) - \Phi_k^*]\right)^{\vartheta} \leq C\mathbb{E}\|\xi\|, \ \ \forall \xi \in \partial\Phi(x). \tag{13}$$

**Theorem 2.** *Assume the conditions of Lemma 5 hold. Let $\{\widetilde{x}^k\} \to \widetilde{x}^*$, then there hold:*

(i) *If $\vartheta \in (0, \frac{1}{2}]$, there exist $d_1 > 0$ and $\zeta \in [1-\rho, 1)$ such that $\mathbb{E}\left\|\widetilde{x}^k - \tilde{x}^*\right\| \leq d_1\zeta^k$.*

(ii) *If $\vartheta \in (\frac{1}{2}, 1)$, there exists a constant $d_2 > 0$ such that $\mathbb{E}\left\|\widetilde{x}^k - \tilde{x}^*\right\| \leq d_2 k^{-\frac{1-\vartheta}{2\vartheta-1}}$.*

(iii) *If $\vartheta = 0$, there exists a $m \in \mathbb{N}$ such that $\mathbb{E}[\Phi(\widetilde{x}^k)] = \mathbb{E}[\Phi(\tilde{x}^*)]$ for all $k \geq l$.*

## 4 NUMERICAL RESULTS

In this section, we evaluate the effectiveness of STNAdam-SGD, STNAdam-SAGA and STNAdam-SARAH on low-light image enhancement (LIE), compared with the SGD (Bottou, 2010), SAdam (Kingma & Ba, 2014), and SNAdam (Xie et al., 2024) methods. Additional comparisons are also made with the customized algorithms of LIE, including NPE (Fu et al., 2015), DeHz (Dong et al., 2011), LIME (Guo et al., 2017), Retinex-Net (Wei et al., 2018) and LR3M (Ren et al., 2020).

Specifically, we consider the following LIE model (Ren et al., 2020):

$$\min_{R,L} \quad \|R \circ L - S\|_2^2 + \hbar\|\nabla L\|_{1/2}^{1/2} + \ell\|\text{NN}_i(R)\|_* + \eta\|\nabla R - G\|_2^2, \tag{14}$$

where $S$ is the observed image; $G$ is its adjusted gradient; $R$ is the reflectance layer; $L$ is the illumination layer; $\text{NN}_i(\cdot)$ is an extraction operation that collects similar patches to the $i$-th position. Model (14) can be converted to (1), as long as let $f(R,L) = \sum_{i=1}^{N}[\|R \circ L - S_i\|_2^2 + \eta\|\nabla_i R - G_i\|_2^2]$, $g(R,L) = \hbar\|\nabla L\|_{1/2}^{1/2} + \ell\|\text{NN}_i(R)\|_*$. Then, we further adopt the training framework of Retinex-Net (Wei et al., 2018) for (14), where the data sources and detailed process are given in Appendix.

Table 2: Numerical results of the eleven algorithms for LIE on the LOL dataset.

| Algorithm | PSNR↑ | SSIM↑ | LPIPS↓ | Time(s) | Algorithm | PSNR↑ | SSIM↑ | LPIPS↓ | Time(s) |
|---|---|---|---|---|---|---|---|---|---|
| Input | 7.3465 | 0.4090 | 0.4431 | - | SGD | 14.8024 | 0.6438 | 0.2692 | 2.85e-05 |
| NPE | 13.3294 | 0.6056 | 0.2789 | 3.47e-05 | SAdam | 16.3781 | 0.7050 | 0.1235 | 5.79e-05 |
| DeHz | 15.0894 | 0.6769 | 0.1690 | 3.39e-05 | SNAdam | 17.1359 | 0.7945 | 0.0984 | 2.81e-05 |
| LIME | 16.2409 | 0.6995 | 0.2160 | 3.28e-05 | STNAdam-SGD | 18.0631 | 0.8194 | 0.0856 | 3.18e-05 |
| LR3M | 16.9564 | 0.7168 | 0.1452 | 3.04e-05 | STNAdam-SAGA | 21.0502 | 0.8886 | 0.0663 | 3.12e-05 |
| Retinex-Net | 18.4396 | 0.8205 | 0.0794 | 7.63e-05 | STNAdam-SARAH | **22.2581** | **0.9062** | **0.0501** | **2.64e-05** |

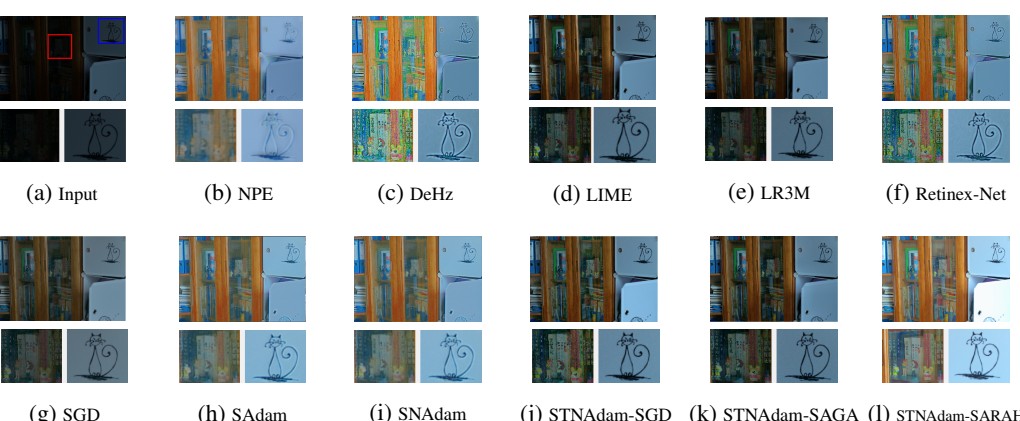

(a) Input    (b) NPE    (c) DeHz    (d) LIME    (e) LR3M    (f) Retinex-Net

(g) SGD    (h) SAdam    (i) SNAdam    (j) STNAdam-SGD    (k) STNAdam-SAGA    (l) STNAdam-SARAH

Figure 2: Visualization results of the eleven algorithms on the LOL dataset.

We report the numerical results of various methods in terms of three image evaluation metrics, including PSNR, SSIM, and LPIPS, in Table 2. The corresponding visualization results are shown in Fig. 2. From Table 2 and Fig. 2, we make the following observations.

- Compared with all other methods, our STNAdam-SARAH shows absolute advantages since it achieves the highest values for PSNR and SSIM, and the lowest value for LPIPS. Moreover, STNAdam-SAGA and STNAdam-SGD occupy the second and third positions in terms of the values of PSNR, SSIM, and LPIPS, respectively.

- Our STNAdam-SARAH yields the most favourable image restoration output since it avoids dark edges and produces clearer results, whereas SAdam and other variants yield blurry and shadowy images (e.g., the kitten doodle). Compared to all the customized algorithms of LIE, our method effectively illuminates objects against dark backgrounds without over-exposure, unlike DeHz, which exhibits partial overexposure.

Table 3: Numerical results of various methods for LIE with noise on the LOL dataset.

| Algorithm | Wardrobe | | | | Doll | | | |
|---|---|---|---|---|---|---|---|---|
| | PSNR↑ | SSIM↑ | LPIPS↓ | Time(s) | PSNR↑ | SSIM↑ | LPIPS↓ | Time(s) |
| LIME | 15.9424 | 0.8542 | 0.1082 | 3.55e-05 | 12.9593 | 0.8372 | 0.1257 | 5.06e-05 |
| LR3M | 16.6897 | 0.8842 | 0.0778 | 2.92e-05 | 13.8628 | 0.8712 | 0.0934 | 3.65e-05 |
| Retinex-Net | 17.1421 | 0.9087 | 0.0721 | 2.96e-05 | 16.7618 | 0.9033 | 0.0782 | 3.22e-05 |
| STNAdam-SARAH | **20.9119** | **0.9781** | **0.0385** | **2.34e-05** | **19.9958** | **0.9581** | **0.0421** | **2.93e-05** |

Based on the above results, we select the three best ones from the customized algorithms of LIE, i.e., LIME, LR3M and Retinex-Net. Then, we further evaluate the joint denoising performance using STNAdam-SARAH against the three alternatives. The comparison results are reported in Table 3 and Fig. 3. We make the following observations.

- Our STNAdam-SARAH is superior to the other three methods in terms of preserving image quality since it achieves the optimal quantitative metrics for enhanced images, as shown in Table 3. Moreover, our method outperforms the other three approaches in terms of speed.

- In reflectance-based denoising tasks, our STNAdam-SARAH performs the best since it preserves details more effectively, as illustrated in Fig. 3, whereas LIME, LR3M and Retinex-Net blur edges and reduce color contrast.

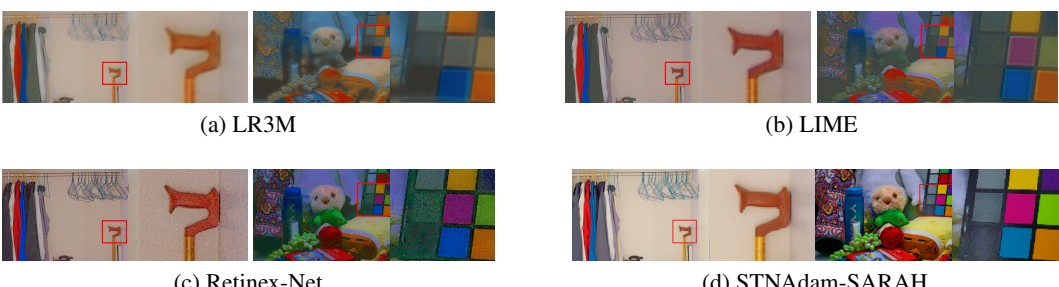

(a) LR3M                                                    (b) LIME

(c) Retinex-Net                                        (d) STNAdam-SARAH

Figure 3: Joint denoising comparison results of various methods on the LOL dataset.

## 5 CONCLUDING REMARKS

In this paper, we propose the STNAdam algorithm to solve "nonconvex + weakly-convex" composite optimizations. This algorithm adopts a novel two-track iteration framework, and is essentially an enhanced version of stochastic Adam, combining Adam and Nesterov-accelerated technique. Under the Kurdyka-Łojasiewicz property, we establish the global convergence of STNAdam in expectation. Finally, we perform numerical tests on low-light image enhancement tasks to demonstrate the superiority of STNAdam.

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

# A APPENDIX

The appendix is organized as follows:

- The proofs of the important Lemmas 2, 3, 4 and 5 used in convergence analysis are given in Section A.1.
- The proof of Theorem 1 (converge to a stationary point) is provided in Section A.2.
- The proof of Theorem 2 (convergence rate) is provided in Section A.3.
- Experimental details and additional experimental results are provided in Section A.4.

In what follows, unless otherwise specified, let $\{\theta^k\} = \left\{\left(\widetilde{x}^k, x^k\right)\right\}$ be the sequence generated by Algorithm 1 with variance-reduced gradient estimator.

## A.1 IMPORTANT LEMMAS NEEDED FOR CONVERGENCE ANALYSIS

**Lemma A.1.** [Lemma 2] *Under Assumption 1, for any $k \geq 1$, we have*

(i)

$$
\begin{aligned}
\mathbb{E}_k\left[G^{k+1}\right] \leq G^k &- A_1\left\|\widetilde{x}^{k+1} - \widetilde{x}^k\right\|^2 - A_2\left\|x^{k+1} - x^k\right\|^2 - A_3\left\|\widetilde{x}^{k+1} - x^k\right\|^2 - A_4\left\|x^{k+1} - \widetilde{x}^k\right\|^2 \\
&- A_5\left\|\widetilde{x}^k - x^k\right\|^2 - A_6\left\|\nabla f(x^k) - m^k\right\|^2 - A_7\left\|x^k - x^{k-1}\right\|^2 - A_8\left\|m^{k+1}\right\|^2,
\end{aligned}
$$

(A.15)

*where*

- $A_1 = \frac{\sqrt{\widehat{\pi}_{k+1}} + \varepsilon}{8\overline{\alpha}} - \frac{1}{2s} > 0;$

- $A_2 = \frac{\sqrt{\widehat{\pi}_{k+1}} + \varepsilon}{\alpha} - \frac{1}{s} - \frac{3(\sqrt{\widehat{\pi}_{k+1}} + \varepsilon)}{4\underline{\alpha}} - \left(8s\left(2\underline{\gamma}^2 - 4\underline{\gamma} + 3\right) + 2M\right)L^2 - H > 0;$

- $A_3 = \frac{5(\sqrt{\widehat{\pi}_{k+1}} + \varepsilon)}{8\overline{\alpha}} - \frac{\tau}{2} - \frac{\sqrt{\widehat{\pi}_{k+1}} + \varepsilon}{2\alpha} - \frac{L}{2} - \frac{1}{2s} > 0;$

- $A_4 = \frac{\sqrt{\widehat{\pi}_{k+1}} + \varepsilon}{2\alpha} - \frac{\tau}{2} - \frac{L}{2} - \frac{\sqrt{\widehat{\pi}_{k+1}} + \varepsilon}{4\underline{\alpha}} > 0;$

- $A_5 = \frac{(56\lambda - 40\lambda^2 - 19)(\sqrt{\widehat{\pi}_{k+1}} + \varepsilon)}{16\underline{\alpha}} - \tau - \frac{1}{2s} - \frac{sL^2}{2} - L > 0;$

- $A_6 = (1 - 2\mu^2)M - 8s\left(2\underline{\gamma}^2 - 4\underline{\gamma} + 3\right) - 4s(V_1 + V_\Upsilon/\rho) > 0;$

- $A_7 = H - 4s(V_1 + V_\Upsilon/\rho) > 0;$

- $A_8 = Z - \frac{2D(\mu^{k+1})^2}{(1 - \mu^{k+1})^2}.$

(ii)

$$
\begin{aligned}
\sum_{k=0}^{\infty}\mathbb{E}\Big(&\left\|\widetilde{x}^{k+1} - \widetilde{x}^k\right\|^2 + \left\|x^{k+1} - x^k\right\|^2 + \left\|\widetilde{x}^{k+1} - x^k\right\|^2 + \left\|x^{k+1} - \widetilde{x}^k\right\|^2 + \left\|\widetilde{x}^k - x^k\right\|^2 \\
&+ \left\|\nabla f(x^k) - m^k\right\|^2 + \left\|x^k - x^{k-1}\right\|^2 + \left\|m^{k+1}\right\|^2\Big) < +\infty,
\end{aligned}
$$

*and hence the sequence $\left\{\mathbb{E}\left\|\widetilde{x}^{k+1} - \widetilde{x}^k\right\|^2\right\}$ is summable.*

*Proof.* According to Lemma 1 (Descent Lemma) in (Bolte et al., 2014), we have the inequalities

$$
f(x^k) - f(\widetilde{x}^k) \leq \left\langle\nabla f(x^k), x^k - \widetilde{x}^k\right\rangle + \frac{L}{2}\left\|\widetilde{x}^k - x^k\right\|^2,
$$

$$
f(\widetilde{x}^{k+1}) - f(x^k) \leq \left\langle\nabla f(x^k), \widetilde{x}^{k+1} - x^k\right\rangle + \frac{L}{2}\left\|\widetilde{x}^{k+1} - x^k\right\|^2,
$$

which implies that

$$f(\widetilde{x}^{k+1}) \le f(\widetilde{x}^k) + \left\langle \nabla f(x^k), \widetilde{x}^{k+1} - \widetilde{x}^k \right\rangle + \frac{L}{2} \left\| x^k - \widetilde{x}^k \right\|^2 + \frac{L}{2} \left\| \widetilde{x}^{k+1} - x^k \right\|^2. \tag{A.16}$$

Similarly, we obtain

$$f(x^{k+1}) \le f(x^k) + \left\langle \nabla f(\widetilde{x}^k), x^{k+1} - x^k \right\rangle + \frac{L}{2} \left\| \widetilde{x}^k - x^k \right\|^2 + \frac{L}{2} \left\| x^{k+1} - \widetilde{x}^k \right\|^2. \tag{A.17}$$

Now, according to the definition of $\mathcal{P}_g(x, y, t)$, using the optimality condition of the Step 5 of the Algorithm 1, for any $x \in \mathbb{R}^d$, we can obtain

$$\begin{aligned}
&\frac{\sqrt{\widehat{\pi}_{k+1}} + \varepsilon}{2\alpha} \left( \left\| \widetilde{x}^k - x^k \right\|^2 - \left\| x^{k+1} - x^k \right\|^2 - \left\| x^{k+1} - \widetilde{x}^k \right\|^2 \right) \\
&= \frac{\sqrt{\widehat{\pi}_{k+1}} + \varepsilon}{\alpha} \left\langle x^{k+1} - x^k, \widetilde{x}^k - x^{k+1} \right\rangle \\
&= \left\langle \widehat{\varpi}^{k+1} + v, x^{k+1} - \widetilde{x}^k \right\rangle,
\end{aligned} \tag{A.18}$$

for some $v \in \partial g(x^{k+1})$. By the weakly convexity of $g(\cdot)$, we have

$$g(x^{k+1}) - g(\widetilde{x}^k) \le \left\langle v, x^{k+1} - \widetilde{x}^k \right\rangle + \frac{\tau}{2} \left\| x^{k+1} - \widetilde{x}^k \right\|^2. \tag{A.19}$$

Then together (A.18) with (A.19), we have

$$\begin{aligned}
g(x^{k+1}) \le\, & g(\widetilde{x}^k) + \left\langle \widehat{\varpi}^{k+1}, \widetilde{x}^k - x^{k+1} \right\rangle + \frac{\tau}{2} \left\| x^{k+1} - \widetilde{x}^k \right\|^2 \\
& + \frac{\sqrt{\widehat{\pi}_{k+1}} + \varepsilon}{2\alpha} \left[ \left\| \widetilde{x}^k - x^k \right\|^2 - \left\| x^{k+1} - x^k \right\|^2 - \left\| x^{k+1} - \widetilde{x}^k \right\|^2 \right].
\end{aligned} \tag{A.20}$$

Similarly, we obtain

$$\begin{aligned}
g(x^{k+1}) \le\, & g(\widetilde{x}^{k+1}) + \left\langle \widehat{\varpi}^{k+1}, \widetilde{x}^{k+1} - x^{k+1} \right\rangle + \frac{\tau}{2} \left\| x^{k+1} - \widetilde{x}^{k+1} \right\|^2 \\
& + \frac{\sqrt{\widehat{\pi}_{k+1}} + \varepsilon}{2\alpha} \left[ \left\| \widetilde{x}^{k+1} - x^k \right\|^2 - \left\| x^{k+1} - x^k \right\|^2 - \left\| x^{k+1} - \widetilde{x}^{k+1} \right\|^2 \right].
\end{aligned} \tag{A.21}$$

Likewise, by the definition of $\mathcal{P}_g(x, y, t)$ for the optimality condition of the Step 5 of the Algorithm 1, together with the weakly convexity of $g(\cdot)$, we have

$$\begin{aligned}
g(\widetilde{x}^{k+1}) \le\, & g(x^{k+1}) + \left\langle \widetilde{\varpi}^{k+1}, x^{k+1} - \widetilde{x}^{k+1} \right\rangle + \frac{\tau}{2} \left\| \widetilde{x}^{k+1} - x^{k+1} \right\|^2 \\
& + \frac{\sqrt{\widehat{\pi}_{k+1}} + \varepsilon}{2\alpha_{k+1}} \left[ \left\| x^{k+1} - \overline{x}^{k+1} \right\|^2 - \left\| \widetilde{x}^{k+1} - \overline{x}^{k+1} \right\|^2 - \left\| \widetilde{x}^{k+1} - x^{k+1} \right\|^2 \right].
\end{aligned} \tag{A.22}$$

Similarly, we obtain

$$\begin{aligned}
g(\widetilde{x}^{k+1}) \le\, & g(x^k) + \left\langle \widetilde{\varpi}^{k+1}, x^k - \widetilde{x}^{k+1} \right\rangle + \frac{\tau}{2} \left\| \widetilde{x}^{k+1} - x^k \right\|^2 \\
& + \frac{\sqrt{\widehat{\pi}_{k+1}} + \varepsilon}{2\alpha_{k+1}} \left[ \left\| x^k - \overline{x}^{k+1} \right\|^2 - \left\| \widetilde{x}^{k+1} - \overline{x}^{k+1} \right\|^2 - \left\| \widetilde{x}^{k+1} - x^k \right\|^2 \right].
\end{aligned} \tag{A.23}$$

Adding (A.20) and (A.22), we have

$$\begin{aligned}
& g(\widetilde{x}^{k+1}) \\
\le\, & g(\widetilde{x}^k) + \left\langle \widetilde{\varpi}^{k+1} - \widehat{\varpi}^{k+1}, x^{k+1} - \widetilde{x}^{k+1} \right\rangle + \left\langle \widehat{\varpi}^{k+1}, \widetilde{x}^k - \widetilde{x}^{k+1} \right\rangle + \frac{\sqrt{\widehat{\pi}_{k+1}} + \varepsilon}{2\alpha} \left\| \widetilde{x}^k - x^k \right\|^2 \\
& - \frac{\sqrt{\widehat{\pi}_{k+1}} + \varepsilon}{2\alpha} \left\| x^{k+1} - x^k \right\|^2 - \left( \frac{\sqrt{\widehat{\pi}_{k+1}} + \varepsilon}{2\alpha} - \frac{\tau}{2} \right) \left\| x^{k+1} - \widetilde{x}^k \right\|^2 + \frac{\sqrt{\widehat{\pi}_{k+1}} + \varepsilon}{2\alpha_{k+1}} \left\| x^{k+1} - \overline{x}^{k+1} \right\|^2 \\
& - \frac{\sqrt{\widehat{\pi}_{k+1}} + \varepsilon}{2\alpha_{k+1}} \left\| \widetilde{x}^{k+1} - \overline{x}^{k+1} \right\|^2 - \left( \frac{\sqrt{\widehat{\pi}_{k+1}} + \varepsilon}{2\alpha_{k+1}} - \frac{\tau}{2} \right) \left\| \widetilde{x}^{k+1} - x^{k+1} \right\|^2.
\end{aligned} \tag{A.24}$$

Adding (A.21) and (A.23), we have

$$g(x^{k+1})$$

$$\leq g(x^k) + \left\langle \widetilde{\varpi}^{k+1} - \widehat{\varpi}^{k+1}, x^k - \widetilde{x}^{k+1} \right\rangle + \left\langle \widehat{\varpi}^{k+1}, x^k - x^{k+1} \right\rangle + \frac{\sqrt{\widehat{\pi}_{k+1}} + \varepsilon}{2\alpha} \left\| \widetilde{x}^{k+1} - x^k \right\|^2$$

$$- \frac{\sqrt{\widehat{\pi}_{k+1}} + \varepsilon}{2\alpha} \left\| x^{k+1} - x^k \right\|^2 - \left( \frac{\sqrt{\widehat{\pi}_{k+1}} + \varepsilon}{2\alpha} - \frac{\tau}{2} \right) \left\| x^{k+1} - \widetilde{x}^{k+1} \right\|^2 + \frac{\sqrt{\widehat{\pi}_{k+1}} + \varepsilon}{2\alpha_{k+1}} \left\| x^k - \overline{x}^{k+1} \right\|^2$$

$$- \frac{\sqrt{\widehat{\pi}_{k+1}} + \varepsilon}{2\alpha_{k+1}} \left\| \widetilde{x}^{k+1} - \overline{x}^{k+1} \right\|^2 - \left( \frac{\sqrt{\widehat{\pi}_{k+1}} + \varepsilon}{2\alpha_{k+1}} - \frac{\tau}{2} \right) \left\| \widetilde{x}^{k+1} - x^k \right\|^2 .$$

$$\tag{A.25}$$

According to (A.16) and (A.24), we have

$$\Phi(\widetilde{x}^{k+1})$$

$$\leq \Phi(\widetilde{x}^k) + \left\langle \widetilde{\varpi}^{k+1} - \widehat{\varpi}^{k+1}, x^{k+1} - \widetilde{x}^{k+1} \right\rangle + \left\langle \nabla f(x^k) - \widehat{\varpi}^{k+1}, \widetilde{x}^{k+1} - \widetilde{x}^k \right\rangle$$

$$+ \left( \frac{L}{2} + \frac{\sqrt{\widehat{\pi}_{k+1}} + \varepsilon}{2\alpha} \right) \left\| \widetilde{x}^k - x^k \right\|^2 + \frac{L}{2} \left\| \widetilde{x}^{k+1} - x^k \right\|^2 - \frac{\sqrt{\widehat{\pi}_{k+1}} + \varepsilon}{2\alpha} \left\| x^{k+1} - x^k \right\|^2$$

$$- \left( \frac{\sqrt{\widehat{\pi}_{k+1}} + \varepsilon}{2\alpha} - \frac{\tau}{2} \right) \left\| x^{k+1} - \widetilde{x}^k \right\|^2 + \frac{\sqrt{\widehat{\pi}_{k+1}} + \varepsilon}{2\alpha_{k+1}} \left\| x^{k+1} - \overline{x}^{k+1} \right\|^2$$

$$- \frac{\sqrt{\widehat{\pi}_{k+1}} + \varepsilon}{2\alpha_{k+1}} \left\| \widetilde{x}^{k+1} - \overline{x}^{k+1} \right\|^2 - \left( \frac{\sqrt{\widehat{\pi}_{k+1}} + \varepsilon}{2\alpha_{k+1}} - \frac{\tau}{2} \right) \left\| \widetilde{x}^{k+1} - x^{k+1} \right\|^2 .$$

$$\tag{A.26}$$

According to (A.17) and (A.25), we have

$$\Phi(x^{k+1})$$

$$\leq \Phi(x^k) + \left\langle \widetilde{\varpi}^{k+1} - \widehat{\varpi}^{k+1}, x^k - \widetilde{x}^{k+1} \right\rangle + \left\langle \nabla f(\widetilde{x}^k) - \widehat{\varpi}^{k+1}, x^{k+1} - x^k \right\rangle + \frac{L}{2} \left\| \widetilde{x}^k - x^k \right\|^2$$

$$+ \frac{L}{2} \left\| x^{k+1} - \widetilde{x}^k \right\|^2 - \left( \frac{\sqrt{\widehat{\pi}_{k+1}} + \varepsilon}{2\alpha_{k+1}} - \frac{\tau}{2} - \frac{\sqrt{\widehat{\pi}_{k+1}} + \varepsilon}{2\alpha} \right) \left\| \widetilde{x}^{k+1} - x^k \right\|^2$$

$$- \frac{\sqrt{\widehat{\pi}_{k+1}} + \varepsilon}{2\alpha} \left\| x^{k+1} - x^k \right\|^2 - \left( \frac{\sqrt{\widehat{\pi}_{k+1}} + \varepsilon}{2\alpha} - \frac{\tau}{2} \right) \left\| x^{k+1} - \widetilde{x}^{k+1} \right\|^2$$

$$+ \frac{\sqrt{\widehat{\pi}_{k+1}} + \varepsilon}{2\alpha_{k+1}} \left\| x^k - \overline{x}^{k+1} \right\|^2 - \frac{\sqrt{\widehat{\pi}_{k+1}} + \varepsilon}{2\alpha_{k+1}} \left\| \widetilde{x}^{k+1} - \overline{x}^{k+1} \right\|^2 .$$

$$\tag{A.27}$$

Adding (A.26) and (A.27), we have

$$\Phi(\widetilde{x}^{k+1}) + \Phi(x^{k+1})$$

$$\leq \Phi(\widetilde{x}^k) + \Phi(x^k) + \left\langle \widetilde{\varpi}^{k+1} - \widehat{\varpi}^{k+1}, x^{k+1} - \widetilde{x}^{k+1} \right\rangle + \left\langle \widetilde{\varpi}^{k+1} - \widehat{\varpi}^{k+1}, x^k - \widetilde{x}^{k+1} \right\rangle$$

$$+ \left\langle \nabla f(x^k) - \widehat{\varpi}^{k+1}, \widetilde{x}^{k+1} - \widetilde{x}^k \right\rangle + \left\langle \nabla f(x^k) - \widehat{\varpi}^{k+1}, x^{k+1} - x^k \right\rangle + \left\langle \nabla f(\widetilde{x}^k) - \nabla f(x^k), x^{k+1} - x^k \right\rangle$$

$$+ \left( L + \frac{\sqrt{\widehat{\pi}_{k+1}} + \varepsilon}{2\alpha} \right) \left\| \widetilde{x}^k - x^k \right\|^2 - \left( \frac{\sqrt{\widehat{\pi}_{k+1}} + \varepsilon}{2\alpha_{k+1}} - \frac{\tau}{2} - \frac{\sqrt{\widehat{\pi}_{k+1}} + \varepsilon}{2\alpha} - \frac{L}{2} \right) \left\| \widetilde{x}^{k+1} - x^k \right\|^2$$

$$- \frac{\sqrt{\widehat{\pi}_{k+1}} + \varepsilon}{\alpha} \left\| x^{k+1} - x^k \right\|^2 - \left( \frac{\sqrt{\widehat{\pi}_{k+1}} + \varepsilon}{2\alpha} - \frac{\tau}{2} - \frac{L}{2} \right) \left\| x^{k+1} - \widetilde{x}^k \right\|^2$$

$$- \left( \frac{\sqrt{\widehat{\pi}_{k+1}} + \varepsilon}{2\alpha_{k+1}} + \frac{\sqrt{\widehat{\pi}_{k+1}} + \varepsilon}{2\alpha} - \tau \right) \left\| \widetilde{x}^{k+1} - x^{k+1} \right\|^2 + \frac{\sqrt{\widehat{\pi}_{k+1}} + \varepsilon}{2\alpha_{k+1}} \left\| x^{k+1} - \overline{x}^{k+1} \right\|^2$$

$$+ \frac{\sqrt{\widehat{\pi}_{k+1}} + \varepsilon}{2\alpha_{k+1}} \left\| x^k - \overline{x}^{k+1} \right\|^2 - \frac{\sqrt{\widehat{\pi}_{k+1}} + \varepsilon}{\alpha_{k+1}} \left\| \widetilde{x}^{k+1} - \overline{x}^{k+1} \right\|^2$$

$$\leq \Phi(\widetilde{x}^k) + \Phi(x^k) + s\left\|\widetilde{\varpi}^{k+1} - \widehat{\varpi}^{k+1}\right\|^2 + \frac{1}{2s}\left\|x^{k+1} - \widetilde{x}^{k+1}\right\|^2 + \frac{1}{2s}\left\|\widetilde{x}^{k+1} - x^k\right\|^2$$

$$+ s\left\|\nabla f(x^k) - \widehat{\varpi}^{k+1}\right\|^2 + \frac{1}{2s}\left\|\widetilde{x}^{k+1} - \widetilde{x}^k\right\|^2 + \frac{1}{2s}\left\|x^{k+1} - x^k\right\|^2 + \frac{sL^2}{2}\left\|\widetilde{x}^k - x^k\right\|^2$$

$$+ \frac{1}{2s}\left\|x^{k+1} - x^k\right\|^2 + \left(L + \frac{\sqrt{\widehat{\pi}_{k+1}} + \varepsilon}{2\alpha}\right)\left\|\widetilde{x}^k - x^k\right\|^2 - \left(\frac{\sqrt{\widehat{\pi}_{k+1}} + \varepsilon}{2\alpha_{k+1}} - \frac{\tau}{2} - \frac{\sqrt{\widehat{\pi}_{k+1}} + \varepsilon}{2\alpha} - \frac{L}{2}\right)$$

$$\left\|\widetilde{x}^{k+1} - x^k\right\|^2 - \frac{\sqrt{\widehat{\pi}_{k+1}} + \varepsilon}{\alpha}\left\|x^{k+1} - x^k\right\|^2 - \left(\frac{\sqrt{\widehat{\pi}_{k+1}} + \varepsilon}{2\alpha} - \frac{\tau}{2} - \frac{L}{2}\right)\left\|x^{k+1} - \widetilde{x}^k\right\|^2$$

$$- \left(\frac{\sqrt{\widehat{\pi}_{k+1}} + \varepsilon}{2\alpha_{k+1}} + \frac{\sqrt{\widehat{\pi}_{k+1}} + \varepsilon}{2\alpha} - \tau\right)\left\|\widetilde{x}^{k+1} - x^{k+1}\right\|^2 + \frac{\sqrt{\widehat{\pi}_{k+1}} + \varepsilon}{8\alpha_{k+1}}\left\|x^{k+1} - \overline{x}^{k+1}\right\|^2$$

$$+ \frac{3(\sqrt{\widehat{\pi}_{k+1}} + \varepsilon)}{8\alpha_{k+1}}\left\|x^{k+1} - \overline{x}^{k+1}\right\|^2 + \frac{\sqrt{\widehat{\pi}_{k+1}} + \varepsilon}{2\alpha_{k+1}}\left\|x^k - \overline{x}^{k+1}\right\|^2 - \frac{\sqrt{\widehat{\pi}_{k+1}} + \varepsilon}{2\alpha_{k+1}}\left\|\widetilde{x}^{k+1} - \overline{x}^{k+1}\right\|^2$$

$$- \frac{\sqrt{\widehat{\pi}_{k+1}} + \varepsilon}{2\alpha_{k+1}}\left\|\widetilde{x}^{k+1} - \overline{x}^{k+1}\right\|^2$$

$$\leq \Phi(\widetilde{x}^k) + \Phi(x^k) + s\left\|\widetilde{\varpi}^{k+1} - \widehat{\varpi}^{k+1}\right\|^2 + s\left\|\nabla f(x^k) - \widehat{\varpi}^{k+1}\right\|^2 + \frac{1}{2s}\left\|\widetilde{x}^{k+1} - \widetilde{x}^k\right\|^2$$

$$+ \left(\frac{sL^2}{2} + L + \frac{\sqrt{\widehat{\pi}_{k+1}} + \varepsilon}{2\alpha}\right)\left\|\widetilde{x}^k - x^k\right\|^2 - \left(\frac{\sqrt{\widehat{\pi}_{k+1}} + \varepsilon}{2\alpha_{k+1}} - \frac{\tau}{2} - \frac{\sqrt{\widehat{\pi}_{k+1}} + \varepsilon}{2\alpha} - \frac{L}{2} - \frac{1}{2s}\right)$$

$$\left\|\widetilde{x}^{k+1} - x^k\right\|^2 - \left(\frac{\sqrt{\widehat{\pi}_{k+1}} + \varepsilon}{\alpha} - \frac{1}{s}\right)\left\|x^{k+1} - x^k\right\|^2 - \left(\frac{\sqrt{\widehat{\pi}_{k+1}} + \varepsilon}{2\alpha} - \frac{\tau}{2} - \frac{L}{2}\right)\left\|x^{k+1} - \widetilde{x}^k\right\|^2$$

$$- \left(\frac{\sqrt{\widehat{\pi}_{k+1}} + \varepsilon}{2\alpha_{k+1}} + \frac{\sqrt{\widehat{\pi}_{k+1}} + \varepsilon}{2\alpha} - \tau - \frac{1}{2s}\right)\left\|\widetilde{x}^{k+1} - x^{k+1}\right\|^2 + \frac{\sqrt{\widehat{\pi}_{k+1}} + \varepsilon}{4\alpha_{k+1}}\left\|x^{k+1} - \widetilde{x}^k\right\|^2$$

$$+ \frac{\sqrt{\widehat{\pi}_{k+1}} + \varepsilon}{4\alpha_{k+1}}\left\|\overline{x}^{k+1} - \widetilde{x}^k\right\|^2 + \frac{3(\sqrt{\widehat{\pi}_{k+1}} + \varepsilon)}{4\alpha_{k+1}}\left\|x^{k+1} - x^k\right\|^2 + \frac{3(\sqrt{\widehat{\pi}_{k+1}} + \varepsilon)}{4\alpha_{k+1}}\left\|\overline{x}^{k+1} - x^k\right\|^2$$

$$+ \frac{\sqrt{\widehat{\pi}_{k+1}} + \varepsilon}{2\alpha_{k+1}}\left\|\overline{x}^{k+1} - x^k\right\|^2 - \frac{\sqrt{\widehat{\pi}_{k+1}} + \varepsilon}{4\alpha_{k+1}}\left\|\widetilde{x}^{k+1} - \widetilde{x}^k\right\|^2 + \frac{\sqrt{\widehat{\pi}_{k+1}} + \varepsilon}{2\alpha_{k+1}}\left\|\overline{x}^{k+1} - \widetilde{x}^k\right\|^2$$

$$- \frac{\sqrt{\widehat{\pi}_{k+1}} + \varepsilon}{4\alpha_{k+1}}\left\|\widetilde{x}^{k+1} - x^k\right\|^2 + \frac{\sqrt{\widehat{\pi}_{k+1}} + \varepsilon}{2\alpha_{k+1}}\left\|\overline{x}^{k+1} - x^k\right\|^2$$

$$\leq \Phi(\widetilde{x}^k) + \Phi(x^k) + 8s\left(2\gamma_{k+1}^2 - 4\gamma_{k+1} + 3\right)\left\|\nabla f(x^{k+1}) - m^{k+1}\right\|^2 + 4s\left\|\widehat{m}^{k+1} - \widehat{\varpi}^{k+1}\right\|^2$$

$$+ 4s\left\|\widetilde{m}^{k+1} - \widetilde{\varpi}^{k+1}\right\|^2 + \left(\frac{sL^2}{2} + L + \frac{\sqrt{\widehat{\pi}_{k+1}} + \varepsilon}{2\alpha} + \frac{(7(1-\lambda_{k+1})^2 + 3\lambda_{k+1}^2)(\sqrt{\widehat{\pi}_{k+1}} + \varepsilon)}{4\alpha_{k+1}}\right)$$

$$\left\|\widetilde{x}^k - x^k\right\|^2 - \left(\frac{3(\sqrt{\widehat{\pi}_{k+1}} + \varepsilon)}{4\alpha_{k+1}} - \frac{\tau}{2} - \frac{\sqrt{\widehat{\pi}_{k+1}} + \varepsilon}{2\alpha} - \frac{L}{2} - \frac{1}{2s}\right)\left\|\widetilde{x}^{k+1} - x^k\right\|^2$$

$$- \left(\frac{\sqrt{\widehat{\pi}_{k+1}} + \varepsilon}{\alpha} - \frac{1}{s} - \frac{3(\sqrt{\widehat{\pi}_{k+1}} + \varepsilon)}{4\alpha_{k+1}} - 8s\left(2\gamma_{k+1}^2 - 4\gamma_{k+1} + 3\right)L^2\right)\left\|x^{k+1} - x^k\right\|^2$$

$$- \left(\frac{\sqrt{\widehat{\pi}_{k+1}} + \varepsilon}{2\alpha} - \frac{\tau}{2} - \frac{L}{2} - \frac{\sqrt{\widehat{\pi}_{k+1}} + \varepsilon}{4\alpha_{k+1}}\right)\left\|x^{k+1} - \widetilde{x}^k\right\|^2 - \left(\frac{\sqrt{\widehat{\pi}_{k+1}} + \varepsilon}{2\alpha_{k+1}} + \frac{\sqrt{\widehat{\pi}_{k+1}} + \varepsilon}{2\alpha} - \tau - \frac{1}{2s}\right)$$

$$\left\|\widetilde{x}^{k+1} - x^{k+1}\right\|^2 - \left(\frac{\sqrt{\widehat{\pi}_{k+1}} + \varepsilon}{4\alpha_{k+1}} - \frac{1}{2s}\right)\left\|\widetilde{x}^{k+1} - \widetilde{x}^k\right\|^2 + \frac{D(\mu^{k+1})^2}{(1 - \mu^{k+1})^2}\left\|m^{k+1}\right\|^2$$

$$\leq \Phi(\widetilde{x}^k) + \Phi(x^k) + 8s\left(2\underline{\gamma}^2 - 4\underline{\gamma} + 3\right)\left\|\nabla f(x^{k+1}) - m^{k+1}\right\|^2 + 4s\left\|\widehat{m}^{k+1} - \widehat{\varpi}^{k+1}\right\|^2$$

$$+ 4s\left\|\widetilde{m}^{k+1} - \widetilde{\varpi}^{k+1}\right\|^2 + \left(\frac{sL^2}{2} + L + \frac{\sqrt{\widehat{\pi}_{k+1}} + \varepsilon}{2\alpha} + \frac{(7(1-\lambda)^2 + 3\lambda^2)(\sqrt{\widehat{\pi}_{k+1}} + \varepsilon)}{4\underline{\alpha}}\right)$$

$$\left\|\widetilde{x}^k - x^k\right\|^2 - \left(\frac{3(\sqrt{\widehat{\pi}_{k+1}} + \varepsilon)}{4\overline{\alpha}} - \frac{\tau}{2} - \frac{\sqrt{\widehat{\pi}_{k+1}} + \varepsilon}{2\alpha} - \frac{L}{2} - \frac{1}{2s}\right)\left\|\widetilde{x}^{k+1} - x^k\right\|^2$$

$$- \left(\frac{\sqrt{\widehat{\pi}_{k+1}} + \varepsilon}{\alpha} - \frac{1}{s} - \frac{3(\sqrt{\widehat{\pi}_{k+1}} + \varepsilon)}{4\underline{\alpha}} - 8s\left(2\underline{\gamma}^2 - 4\underline{\gamma} + 3\right)\right)L^2\left\|x^{k+1} - x^k\right\|^2$$

$$- \left(\frac{\sqrt{\widehat{\pi}_{k+1}} + \varepsilon}{2\alpha} - \frac{\tau}{2} - \frac{L}{2} - \frac{\sqrt{\widehat{\pi}_{k+1}} + \varepsilon}{4\underline{\alpha}}\right)\left\|x^{k+1} - \widetilde{x}^k\right\|^2 - \left(\frac{\sqrt{\widehat{\pi}_{k+1}} + \varepsilon}{2\overline{\alpha}} + \frac{\sqrt{\widehat{\pi}_{k+1}} + \varepsilon}{2\alpha} - \tau - \frac{1}{2s}\right)$$

$$\left\|\widetilde{x}^{k+1} - x^{k+1}\right\|^2 - \left(\frac{\sqrt{\widehat{\pi}_{k+1}} + \varepsilon}{4\overline{\alpha}} - \frac{1}{2s}\right)\left\|\widetilde{x}^{k+1} - \widetilde{x}^k\right\|^2 - \frac{D(\mu^{k+1})^2}{(1 - \mu^{k+1})^2}\left\|m^{k+1}\right\|^2$$

$$+ \frac{D(\mu^k)^2}{(1 - \mu^k)^2}\left\|m^k\right\|^2 + \frac{2D(\mu^{k+1})^2}{(1 - \mu^{k+1})^2}\left\|m^{k+1}\right\|^2 - \frac{D(\mu^k)^2}{(1 - \mu^k)^2}\left\|m^k\right\|^2,$$

$$(A.28)$$

where the second inequality holds from the fact that $ab \leq \frac{s}{2}a^2 + \frac{1}{2s}b^2$, $\forall a,\ b \in \mathbb{R}$, the thrid inequality holds from the fact that $(a + b)^2 \leq 2a^2 + 2b^2$, $\forall a,\ b \in \mathbb{R}$. The fourth inequality is obtained from the fact that Step 5 of Algorithm 1 and the following inequality:

$$\left\|\widetilde{\varpi}^{k+1} - \widehat{\varpi}^{k+1}\right\|^2 + \left\|\nabla f(x^k) - \widehat{\varpi}^{k+1}\right\|^2$$

$$\leq 4\left\|\widehat{m}^{k+1} - \widehat{\varpi}^{k+1}\right\|^2 + 2\left\|\widehat{m}^{k+1} - \widetilde{\varpi}^{k+1}\right\|^2 + 2\left\|\nabla f(x^k) - \widehat{m}^{k+1}\right\|^2$$

$$\leq 4\left\|\widehat{m}^{k+1} - \widehat{\varpi}^{k+1}\right\|^2 + 4\left\|\widetilde{m}^{k+1} - \widetilde{\varpi}^{k+1}\right\|^2 + 4\left\|\widehat{m}^{k+1} - \widetilde{m}^{k+1}\right\|^2 + 2\left\|\nabla f(x^k) - \widehat{m}^{k+1}\right\|^2$$

$$= 4\left\|\widehat{m}^{k+1} - \widehat{\varpi}^{k+1}\right\|^2 + 4\left\|\widetilde{m}^{k+1} - \widetilde{\varpi}^{k+1}\right\|^2 + (4(1 - \gamma_{k+1})^2 + 2)\left\|\nabla f(x^k) - \widehat{m}^{k+1}\right\|^2$$

$$\leq 4\left\|\widehat{m}^{k+1} - \widehat{\varpi}^{k+1}\right\|^2 + 4\left\|\widetilde{m}^{k+1} - \widetilde{\varpi}^{k+1}\right\|^2 + 4\left(2\gamma_{k+1}^2 - 4\gamma_{k+1} + 3\right)\left\|\nabla f(x^k) - m^{k+1}\right\|^2$$

$$+ 4\left(2\gamma_{k+1}^2 - 4\gamma_{k+1} + 3\right)\left\|\widehat{m}^{k+1} - m^{k+1}\right\|^2$$

$$\overset{(5)}{\leq} 4\left\|\widehat{m}^{k+1} - \widehat{\varpi}^{k+1}\right\|^2 + 4\left\|\widetilde{m}^{k+1} - \widetilde{\varpi}^{k+1}\right\|^2 + 8\left(2\gamma_{k+1}^2 - 4\gamma_{k+1} + 3\right)\left\|\nabla f(x^{k+1}) - m^{k+1}\right\|^2$$

$$+ 8\left(2\gamma_{k+1}^2 - 4\gamma_{k+1} + 3\right)L^2\left\|x^{k+1} - x^k\right\|^2 + \frac{D(\mu^{k+1})^2}{(1 - \mu^{k+1})^2}\left\|m^{k+1}\right\|^2,$$

where $D = 4\left(2\underline{\gamma}^2 - 4\underline{\gamma} + 3\right)$. This is equivalent to

$$\Phi(\widetilde{x}^{k+1}) + \Phi(x^{k+1}) + \left(M - 8s\left(2\underline{\gamma}^2 - 4\underline{\gamma} + 3\right)\right)\left\|\nabla f(x^{k+1}) - m^{k+1}\right\|^2$$

$$+ \left(\frac{\sqrt{\widehat{\pi}_{k+1}} + \varepsilon}{2\overline{\alpha}} + \frac{\sqrt{\widehat{\pi}_{k+1}} + \varepsilon}{2\alpha} - \tau - \frac{1}{2s}\right)\left\|\widetilde{x}^{k+1} - x^{k+1}\right\|^2 + \left(\frac{D(\mu^{k+1})^2}{(1 - \mu^{k+1})^2} - Z\right)\left\|m^{k+1}\right\|^2$$

$$\leq \Phi(\widetilde{x}^k) + \Phi(x^k) + \left(M - 8s\left(2\underline{\gamma}^2 - 4\underline{\gamma} + 3\right)\right)\left\|\nabla f(x^k) - m^k\right\|^2 + \left(\frac{\sqrt{\widehat{\pi}_k} + \varepsilon}{2\overline{\alpha}} + \frac{\sqrt{\widehat{\pi}_k} + \varepsilon}{2\alpha} - \tau - \frac{1}{2s}\right)$$

$$\left\|\widetilde{x}^k - x^k\right\|^2 + \left(\frac{D(\mu^k)^2}{(1 - \mu^k)^2} - Z\right)\left\|m^k\right\|^2 + 4s\left[\left\|\widehat{m}^{k+1} - \widehat{\varpi}^{k+1}\right\|^2 + \left\|\widetilde{m}^{k+1} - \widetilde{\varpi}^{k+1}\right\|^2\right]$$

$$- \left((1 - 2\mu^2)M - 8s\left(2\underline{\gamma}^2 - 4\underline{\gamma} + 3\right)\right)\left\|\nabla f(x^k) - m^k\right\|^2$$

$$- \left(\frac{(56\lambda - 40\lambda^2 - 19)(\sqrt{\widehat{\pi}_{k+1}} + \varepsilon)}{16\underline{\alpha}} - \tau - \frac{1}{2s} - \frac{sL^2}{2} - L\right)\left\|\widetilde{x}^k - x^k\right\|^2$$

$$- \left(\frac{5(\sqrt{\widehat{\pi}_{k+1}} + \varepsilon)}{8\overline{\alpha}} - \frac{\tau}{2} - \frac{\sqrt{\widehat{\pi}_{k+1}} + \varepsilon}{2\alpha} - \frac{L}{2} - \frac{1}{2s}\right)\left\|\widetilde{x}^{k+1} - x^k\right\|^2$$

$$- \left( \frac{\sqrt{\widehat{\pi}_{k+1}} + \varepsilon}{\alpha} - \frac{1}{s} - \frac{3(\sqrt{\widehat{\pi}_{k+1}} + \varepsilon)}{4\underline{\alpha}} - \left( 8s \left( 2\underline{\gamma}^2 - 4\underline{\gamma} + 3 \right) + 2M \right) L^2 \right) \left\| x^{k+1} - x^k \right\|^2$$

$$- \left( \frac{\sqrt{\widehat{\pi}_{k+1}} + \varepsilon}{2\alpha} - \frac{\tau}{2} - \frac{L}{2} - \frac{\sqrt{\widehat{\pi}_{k+1}} + \varepsilon}{4\underline{\alpha}} \right) \left\| x^{k+1} - \widetilde{x}^k \right\|^2 - \left( \frac{\sqrt{\widehat{\pi}_{k+1}} + \varepsilon}{8\overline{\alpha}} - \frac{1}{2s} \right) \left\| \widetilde{x}^{k+1} - \widetilde{x}^k \right\|^2$$

$$- \left( Z - \frac{2D(\mu^{k+1})^2}{(1 - \mu^{k+1})^2} \right) \left\| m^{k+1} \right\|^2 - \left( \frac{D(\mu^k)^2}{(1 - \mu^k)^2} - Z \right) \left\| m^k \right\|^2 .$$

$$\text{(A.29)}$$

Since $\dfrac{2(\mu^{k+1})^2}{(1 - \mu^{k+1})^2} < \dfrac{(\mu^k)^2}{(1 - \mu^k)^2}$, $\mu \in \left( 0, \frac{1}{\sqrt{2}} \right)$, we can drop the nonpositive terms $\left\| m^k \right\|^2$ by setting

$$Z \in \left( \frac{2D(\mu^{k+1})^2}{(1 - \mu^{k+1})^2}, \frac{D(\mu^k)^2}{(1 - \mu^k)^2} \right) .$$

Further, applying the conditional expectation operator $\mathbb{E}_k$, we can bound the MSE terms using (3). This gives

$$\mathbb{E}_k \left[ \Phi(\widetilde{x}^{k+1}) + \Phi(x^{k+1}) + \left( M - 8s \left( 2\underline{\gamma}^2 - 4\underline{\gamma} + 3 \right) \right) \left\| \nabla f(x^{k+1}) - m^{k+1} \right\|^2 \right.$$

$$+ \left( \frac{\sqrt{\widehat{\pi}_{k+1}} + \varepsilon}{2\overline{\alpha}} + \frac{\sqrt{\widehat{\pi}_{k+1}} + \varepsilon}{2\alpha} - \tau - \frac{1}{2s} \right) \left\| \widetilde{x}^{k+1} - x^{k+1} \right\|^2 + \left( \frac{D(\mu^{k+1})^2}{(1 - \mu^{k+1})^2} - Z \right) \left\| m^{k+1} \right\|^2$$

$$\left. + H \left\| x^{k+1} - x^k \right\|^2 \right]$$

$$\leq \Phi(\widetilde{x}^k) + \Phi(x^k) + \left( M - 8s \left( 2\underline{\gamma}^2 - 4\underline{\gamma} + 3 \right) \right) \left\| \nabla f(x^k) - m^k \right\|^2 + \left( \frac{\sqrt{\widehat{\pi}_k} + \varepsilon}{2\overline{\alpha}} + \frac{\sqrt{\widehat{\pi}_k} + \varepsilon}{2\alpha} - \tau - \frac{1}{2s} \right)$$

$$\left\| \widetilde{x}^k - x^k \right\|^2 + \left( \frac{D(\mu^k)^2}{(1 - \mu^k)^2} - Z \right) \left\| m^k \right\|^2 + H \left\| x^k - x^{k-1} \right\|^2 + 4s\Upsilon_k$$

$$- \left( (1 - 2\mu^2)M - 8s \left( 2\underline{\gamma}^2 - 4\underline{\gamma} + 3 \right) - 4sV_1 \right) \left\| \nabla f(x^k) - m^k \right\|^2$$

$$- \left( \frac{(56\lambda - 40\lambda^2 - 19)(\sqrt{\widehat{\pi}_{k+1}} + \varepsilon)}{16\underline{\alpha}} - \tau - \frac{1}{2s} - \frac{sL^2}{2} - L \right) \left\| \widetilde{x}^k - x^k \right\|^2$$

$$- \left( \frac{5(\sqrt{\widehat{\pi}_{k+1}} + \varepsilon)}{8\overline{\alpha}} - \frac{\tau}{2} - \frac{\sqrt{\widehat{\pi}_{k+1}} + \varepsilon}{2\alpha} - \frac{L}{2} - \frac{1}{2s} \right) \left\| \widetilde{x}^{k+1} - x^k \right\|^2$$

$$- \left( \frac{\sqrt{\widehat{\pi}_{k+1}} + \varepsilon}{\alpha} - \frac{1}{s} - \frac{3(\sqrt{\widehat{\pi}_{k+1}} + \varepsilon)}{4\underline{\alpha}} - \left( 8s \left( 2\underline{\gamma}^2 - 4\underline{\gamma} + 3 \right) + 2M \right) L^2 - H \right) \left\| x^{k+1} - x^k \right\|^2$$

$$- \left( \frac{\sqrt{\widehat{\pi}_{k+1}} + \varepsilon}{2\alpha} - \frac{\tau}{2} - \frac{L}{2} - \frac{\sqrt{\widehat{\pi}_{k+1}} + \varepsilon}{4\underline{\alpha}} \right) \left\| x^{k+1} - \widetilde{x}^k \right\|^2 - \left( \frac{\sqrt{\widehat{\pi}_{k+1}} + \varepsilon}{8\overline{\alpha}} - \frac{1}{2s} \right) \left\| \widetilde{x}^{k+1} - \widetilde{x}^k \right\|^2$$

$$- \left( H - 4sV_1 \right) \left\| x^k - x^{k-1} \right\|^2 - \left( Z - \frac{2D(\mu^{k+1})^2}{(1 - \mu^{k+1})^2} \right) \left\| m^{k+1} \right\|^2 .$$

$$\text{(A.30)}$$

Next, we use (5) to say that

$$4s\Upsilon_k \leq \frac{4s}{\rho} \left( -\mathbb{E}_k \Upsilon_{k+1} + \Upsilon_k + V_\Upsilon \left( \left\| \nabla f(x^k) - m^k \right\|^2 + \left\| x^k - x^{k-1} \right\|^2 \right) \right) .$$

Combining these inequalities, we have

$$\mathbb{E}_k \left[ \Phi(\widetilde{x}^{k+1}) + \Phi(x^{k+1}) + \frac{4s}{\rho}\Upsilon_{k+1} + \left( M - 8s \left( 2\underline{\gamma}^2 - 4\underline{\gamma} + 3 \right) \right) \left\| \nabla f(x^{k+1}) - m^{k+1} \right\|^2 \right.$$

$$+ \left( \frac{\sqrt{\widehat{\pi}_{k+1}} + \varepsilon}{2\overline{\alpha}} + \frac{\sqrt{\widehat{\pi}_{k+1}} + \varepsilon}{2\alpha} - \tau - \frac{1}{2s} \right) \left\| \widetilde{x}^{k+1} - x^{k+1} \right\|^2 + \left( \frac{D(\mu^{k+1})^2}{(1 - \mu^{k+1})^2} - Z \right) \left\| m^{k+1} \right\|^2$$

$$+H\left\|x^{k+1}-x^k\right\|^2\Big]$$

$$\leq\Phi(\widetilde{x}^k)+\Phi(x^k)+\frac{4s}{\rho}\Upsilon_k+\left(M-8s\left(2\underline{\gamma}^2-4\underline{\gamma}+3\right)\right)\left\|\nabla f(x^k)-m^k\right\|^2$$

$$+\left(\frac{\sqrt{\widehat{\pi}_k}+\varepsilon}{2\overline{\alpha}}+\frac{\sqrt{\widehat{\pi}_k}+\varepsilon}{2\alpha}-\tau-\frac{1}{2s}\right)\left\|\widetilde{x}^k-x^k\right\|^2+\left(\frac{D(\mu^k)^2}{(1-\mu^k)^2}-Z\right)\left\|m^k\right\|^2+H\left\|x^k-x^{k-1}\right\|^2$$

$$-\left((1-2\mu^2)M-8s\left(2\underline{\gamma}^2-4\underline{\gamma}+3\right)-4s(V_1+V_\Upsilon/\rho)\right)\left\|\nabla f(x^k)-m^k\right\|^2$$

$$-\left(\frac{(56\lambda-40\lambda^2-19)(\sqrt{\widehat{\pi}_{k+1}}+\varepsilon)}{16\underline{\alpha}}-\tau-\frac{1}{2s}-\frac{sL^2}{2}-L\right)\left\|\widetilde{x}^k-x^k\right\|^2$$

$$-\left(\frac{5(\sqrt{\widehat{\pi}_{k+1}}+\varepsilon)}{8\overline{\alpha}}-\frac{\tau}{2}-\frac{\sqrt{\widehat{\pi}_{k+1}}+\varepsilon}{2\alpha}-\frac{L}{2}-\frac{1}{2s}\right)\left\|\widetilde{x}^{k+1}-x^k\right\|^2$$

$$-\left(\frac{\sqrt{\widehat{\pi}_{k+1}}+\varepsilon}{\alpha}-\frac{1}{s}-\frac{3(\sqrt{\widehat{\pi}_{k+1}}+\varepsilon)}{4\underline{\alpha}}-\left(8s\left(2\underline{\gamma}^2-4\underline{\gamma}+3\right)+2M\right)L^2-H\right)\left\|x^{k+1}-x^k\right\|^2$$

$$-\left(\frac{\sqrt{\widehat{\pi}_{k+1}}+\varepsilon}{2\alpha}-\frac{\tau}{2}-\frac{L}{2}-\frac{\sqrt{\widehat{\pi}_{k+1}}+\varepsilon}{4\underline{\alpha}}\right)\left\|x^{k+1}-\widetilde{x}^k\right\|^2-\left(\frac{\sqrt{\widehat{\pi}_{k+1}}+\varepsilon}{8\overline{\alpha}}-\frac{1}{2s}\right)\left\|\widetilde{x}^{k+1}-\widetilde{x}^k\right\|^2$$

$$-\left(H-4s(V_1+V_\Upsilon/\rho)\right)\left\|x^k-x^{k-1}\right\|^2-\left(Z-\frac{2D(\mu^{k+1})^2}{(1-\mu^{k+1})^2}\right)\left\|m^{k+1}\right\|^2.$$

$$(A.31)$$

Now, we need to make sure that the last eight terms on the right-hand side of the inequality in (A.31) are all negative. It's worth noting that since $L$ and $H$ are both related to $s$, in order to get the range of values for $L$ and $H$, we can simply calculate the range of values for $s$ (taking the intersection) by using the terms related to $L$ and $H$ as well as the term related to $s$. Then we can get reasonable ranges of values for $L$ and $H$ at the same time. By setting

$$\mu\in\left(0,\frac{1}{\sqrt{2}}\right),\ s\in\left(\frac{4\overline{\alpha}}{\sqrt{\widehat{\pi}_{k+1}}+\varepsilon},\frac{2\overline{\alpha}^2\left((56\lambda-40\lambda^2-19)(\sqrt{\widehat{\pi}_{k+1}}+\varepsilon)-16\underline{\alpha}\tau\right)}{\left[(\sqrt{\widehat{\pi}_{k+1}}+\varepsilon)(5-4\overline{\alpha})-4\overline{\alpha}\tau\right]^2\underline{\alpha}}\right),$$

$$M>\left(2(2\underline{\gamma}^2-4\underline{\gamma}+3)+V_1+V_\Upsilon/\rho\right)\frac{4s}{1-2\mu^2},\alpha_{k+1}\in\left(\frac{\alpha(\sqrt{\widehat{\pi}_{k+1}}+\varepsilon)}{2(\sqrt{\widehat{\pi}_{k+1}}+\varepsilon)+2\alpha(\tau+L)},1\right),$$

$$\gamma_{k+1}\in\left(1-\frac{\sqrt{2}}{16}\sqrt{(1-\mu^k)^2\left[(1-2\mu^2)M-4s(V_1+V_\Upsilon/\rho)\right]-4},1\right),$$

$$H\in\left(4s(V_1+V_\Upsilon/\rho),\frac{\sqrt{\widehat{\pi}_{k+1}}+\varepsilon}{\alpha}-\frac{1}{s}-\frac{3(\sqrt{\widehat{\pi}_{k+1}}+\varepsilon)}{4\underline{\alpha}}-\left(8s\left(2\underline{\gamma}^2-4\underline{\gamma}+3\right)+2M\right)L^2\right),$$

$$\lambda_{k+1}\in\left(\frac{14-\sqrt{6-10\delta}}{20},\frac{14+\sqrt{6-10\delta}}{20}\right)\text{ with }\delta=\frac{16\alpha_1\left(\tau+\frac{1}{2s}+\frac{sL^2}{2}+L\right)}{\sqrt{\widehat{\pi}_{k+1}}+\varepsilon},$$

we can obtain

$$G^{k+1}\leq G^k-A_1\left\|\widetilde{x}^{k+1}-\widetilde{x}^k\right\|^2-A_2\left\|x^{k+1}-x^k\right\|^2-A_3\left\|\widetilde{x}^{k+1}-x^k\right\|^2-A_4\left\|x^{k+1}-\widetilde{x}^k\right\|^2$$
$$-A_5\left\|\widetilde{x}^k-x^k\right\|^2-A_6\left\|\nabla f(x^k)-m^k\right\|^2-A_7\left\|x^k-x^{k-1}\right\|^2-A_8\left\|m^{k+1}\right\|^2,$$

$$(A.32)$$

where

- $A_1=\frac{\sqrt{\widehat{\pi}_{k+1}}+\varepsilon}{8\overline{\alpha}}-\frac{1}{2s}>0;$

- $A_2=\frac{\sqrt{\widehat{\pi}_{k+1}}+\varepsilon}{\alpha}-\frac{1}{s}-\frac{3(\sqrt{\widehat{\pi}_{k+1}}+\varepsilon)}{4\underline{\alpha}}-\left(8s\left(2\underline{\gamma}^2-4\underline{\gamma}+3\right)+2M\right)L^2-H>0;$

- $A_3 = \frac{5(\sqrt{\widehat{\pi}_{k+1}}+\varepsilon)}{8\overline{\alpha}} - \frac{\tau}{2} - \frac{\sqrt{\widehat{\pi}_{k+1}}+\varepsilon}{2\alpha} - \frac{L}{2} - \frac{1}{2s} > 0$;

- $A_4 = \frac{\sqrt{\widehat{\pi}_{k+1}}+\varepsilon}{2\alpha} - \frac{\tau}{2} - \frac{L}{2} - \frac{\sqrt{\widehat{\pi}_{k+1}}+\varepsilon}{4\underline{\alpha}} > 0$;

- $A_5 = \frac{(56\lambda-40\lambda^2-19)(\sqrt{\widehat{\pi}_{k+1}}+\varepsilon)}{16\underline{\alpha}} - \tau - \frac{1}{2s} - \frac{sL^2}{2} - L > 0$;

- $A_6 = (1 - 2\mu^2)M - 8s\left(2\underline{\gamma}^2 - 4\underline{\gamma} + 3\right) - 4s(V_1 + V_\Upsilon/\rho) > 0$;

- $A_7 = H - 4s(V_1 + V_\Upsilon/\rho) > 0$ and $A_8 = Z - \frac{2D(\mu^{k+1})^2}{(1-\mu^{k+1})^2}$.

So we prove the first claim.

(ii) We apply the full expectation operator to (A.32) and sum the resulting inequality from $k = 0$ to $T - 1$,

$$\mathbb{E}\left[G^T\right] + A_1 \sum_{k=0}^{T-1} \mathbb{E}\left\|\widetilde{x}^{k+1} - \widetilde{x}^k\right\|^2 + A_2 \sum_{k=0}^{T-1} \mathbb{E}\left\|x^{k+1} - x^k\right\|^2 + A_3 \sum_{k=0}^{T-1} \mathbb{E}\left\|\widetilde{x}^{k+1} - x^k\right\|^2$$
$$+ A_4 \sum_{k=0}^{T-1} \mathbb{E}\left\|x^{k+1} - \widetilde{x}^k\right\|^2 + A_5 \sum_{k=0}^{T-1} \mathbb{E}\left\|\widetilde{x}^k - x^k\right\|^2 + A_6 \sum_{k=0}^{T-1} \mathbb{E}\left\|\nabla f(x^k) - m^k\right\|^2$$
$$+ A_7 \sum_{k=0}^{T-1} \mathbb{E}\left\|x^k - x^{k-1}\right\|^2 + A_8 \sum_{k=0}^{T-1} \mathbb{E}\left\|m^{k+1}\right\|^2$$
$$\leq G^0.$$

Since $\Phi$ is bounded from below, i.e., there exists a constant $\Phi_0$ such that $\Phi(x) \geq \Phi_0$. So we have the facts that $\Phi_0 \leq G^T$,

$$A_1 \sum_{k=0}^{T-1} \mathbb{E}\left\|\widetilde{x}^{k+1} - \widetilde{x}^k\right\|^2 + A_2 \sum_{k=0}^{T-1} \mathbb{E}\left\|x^{k+1} - x^k\right\|^2 + A_3 \sum_{k=0}^{T-1} \mathbb{E}\left\|\widetilde{x}^{k+1} - x^k\right\|^2$$
$$+ A_4 \sum_{k=0}^{T-1} \mathbb{E}\left\|x^{k+1} - \widetilde{x}^k\right\|^2 + A_5 \sum_{k=0}^{T-1} \mathbb{E}\left\|\widetilde{x}^k - x^k\right\|^2 + A_6 \sum_{k=0}^{T-1} \mathbb{E}\left\|\nabla f(x^k) - m^k\right\|^2 \quad \text{(A.33)}$$
$$+ A_7 \sum_{k=0}^{T-1} \mathbb{E}\left\|x^k - x^{k-1}\right\|^2 + A_8 \sum_{k=0}^{T-1} \mathbb{E}\left\|m^{k+1}\right\|^2$$
$$\leq G^0 - \Phi_0.$$

Taking the limit $T \to +\infty$, we have the sequence $\left\{\mathbb{E}\left\|\widetilde{x}^{k+1} - \widetilde{x}^k\right\|^2\right\}$ is summable. $\qquad\square$

**Lemma A.2.** [Lemma 3] *For $k \geq 0$, define*

$$\omega_1^{k+1} = \nabla f(x^{k+1}) - \widehat{\varpi}^{k+1} - \frac{\sqrt{\widehat{\pi}_{k+1}}+\varepsilon}{\alpha}(x^{k+1} - x^k),$$

$$\omega_2^{k+1} = \nabla f(\widetilde{x}^{k+1}) - \widetilde{\varpi}^{k+1} - \frac{\sqrt{\widehat{\pi}_{k+1}}+\varepsilon}{\alpha}(x^{k+1} - x^k).$$

*Then, under Assumption 1, we have $\left(\omega_1^{k+1}, \omega_2^{k+1}\right) \in \partial\Phi(\theta^{k+1})$, and there exists $\varrho > 0$, such that*

$$\mathbb{E}_k\left\|\omega^{k+1}\right\|$$
$$\leq \varrho\left(\mathbb{E}_k\left\|\widetilde{x}^{k+1} - \widetilde{x}^k\right\| + \mathbb{E}_k\left\|x^{k+1} - x^k\right\| + \mathbb{E}_k\left\|\widetilde{x}^{k+1} - x^k\right\| + \mathbb{E}_k\left\|x^{k+1} - \widetilde{x}^k\right\| + \left\|\widetilde{x}^k - x^k\right\|$$
$$+ \left\|\nabla f(x^k) - m^k\right\| + \left\|x^k - x^{k-1}\right\| + \mathbb{E}_k\left\|m^{k+1}\right\|\right) + \Gamma_k.$$
$$\text{(A.34)}$$

*Proof.* For simplicity, let $\Phi(\theta) := \Phi(x) + \Phi(y)$, we can obtain $\partial\Phi(\theta^{k+1}) = \begin{bmatrix} \nabla f(x^{k+1}) \\ \nabla f(\widetilde{x}^{k+1}) \end{bmatrix} + \partial g(x^{k+1}) \times \partial g(\widetilde{x}^{k+1})$. Hence $0 \in \partial\Phi(\theta^{k+1})$ is equivalent to

$$\begin{cases} 0 \in \nabla f(x^{k+1}) + \partial g(x^{k+1}), \\ 0 \in \nabla f(\widetilde{x}^{k+1}) + \partial g(\widetilde{x}^{k+1}). \end{cases}$$

From the iterative scheme, we know

$$x^{k+1} \in \arg\min_{x \in \mathbb{R}^d} \left\{ g(x) + \langle \widehat{\varpi}^{k+1}, x \rangle + \frac{\sqrt{\widehat{\pi}_{k+1}} + \varepsilon}{2\alpha} \left\| x - x^k \right\|^2 \right\},$$

$$\widetilde{x}^{k+1} \in \arg\min_{x \in \mathbb{R}^d} \left\{ g(x) + \langle \widetilde{\varpi}^{k+1}, x \rangle + \frac{\sqrt{\widehat{\pi}_{k+1}} + \varepsilon}{2\alpha_{k+1}} \left\| x - \overline{x}^{k+1} \right\|^2 \right\}.$$

By the first-order optimality condition, we have

$$0 \in \partial g(x^{k+1}) + \widehat{\varpi}^{k+1} + \frac{\sqrt{\widehat{\pi}_{k+1}} + \varepsilon}{\alpha}(x^{k+1} - x^k),$$

$$0 \in \partial g(\widetilde{x}^{k+1}) + \widetilde{\varpi}^{k+1} + \frac{\sqrt{\widehat{\pi}_{k+1}} + \varepsilon}{\alpha_{k+1}}(\widetilde{x}^{k+1} - \overline{x}^{k+1}),$$

which implies that $\left(\omega_1^{k+1}, \omega_2^{k+1}\right) \in \partial\Phi(\theta^{k+1})$.

All that remains is to bound the norm of $\omega_1^{k+1}$ and $\omega_2^{k+1}$. Combined with the boundedness of $\left\{x^k\right\}$ and $\left\{\widetilde{x}^k\right\}$, there exists a $\varrho > 0$ such that

$$\left\| \omega_1^{k+1} \right\|$$

$$= \left\| \nabla f(x^{k+1}) - \widehat{\varpi}^{k+1} - \frac{\sqrt{\widehat{\pi}_{k+1}} + \varepsilon}{\alpha}(x^{k+1} - x^k) \right\|$$

$$\leq \left\| \widehat{m}^{k+1} - \widehat{\varpi}^{k+1} \right\| + \left\| \nabla f(x^{k+1}) - \widehat{m}^{k+1} \right\| + \frac{\sqrt{\widehat{\pi}_{k+1}} + \varepsilon}{\alpha} \left\| x^{k+1} - x^k \right\|$$

$$\leq \left\| \widehat{m}^{k+1} - \widehat{\varpi}^{k+1} \right\| + \left\| \nabla f(x^{k+1}) - m^{k+1} \right\| + \left\| \widehat{m}^{k+1} - m^{k+1} \right\| + \frac{\sqrt{\widehat{\pi}_{k+1}} + \varepsilon}{\alpha} \left\| x^{k+1} - x^k \right\|$$

$$\leq \left\| \widehat{m}^{k+1} - \widehat{\varpi}^{k+1} \right\| + \left\| \nabla f(x^{k+1}) - \nabla f(x^k) \right\| + \mu \left\| \nabla f(x^k) - m^k \right\| + \frac{\mu^{k+1}}{1 - \mu^{k+1}} \left\| m^{k+1} \right\|$$

$$+ \frac{\sqrt{\widehat{\pi}_{k+1}} + \varepsilon}{\alpha} \left\| x^{k+1} - x^k \right\|$$

$$\leq \left\| \widehat{m}^{k+1} - \widehat{\varpi}^{k+1} \right\| + \mu \left\| \nabla f(x^k) - m^k \right\| + \left( L + \frac{\sqrt{\widehat{\pi}_{k+1}} + \varepsilon}{\alpha} \right) \left\| x^{k+1} - x^k \right\| + \frac{\mu^{k+1}}{1 - \mu^{k+1}} \left\| m^{k+1} \right\|,$$

and

$$\left\| \omega_2^{k+1} \right\|$$

$$= \left\| \nabla f(\widetilde{x}^{k+1}) - \widetilde{\varpi}^{k+1} - \frac{\sqrt{\widehat{\pi}_{k+1}} + \varepsilon}{\alpha_{k+1}}(\widetilde{x}^{k+1} - \overline{x}^{k+1}) \right\|$$

$$\leq \left\| \nabla f(\widetilde{x}^{k+1}) - \widetilde{m}^{k+1} \right\| + \left\| \widetilde{m}^{k+1} - \widetilde{\varpi}^{k+1} \right\| + \frac{\sqrt{\widehat{\pi}_{k+1}} + \varepsilon}{\alpha_{k+1}} \left\| \widetilde{x}^{k+1} - \widetilde{x}^k \right\| + \frac{\lambda_{k+1}(\sqrt{\widehat{\pi}_{k+1}} + \varepsilon)}{\alpha_{k+1}} \left\| \widetilde{x}^k - x^k \right\|$$

$$\leq \left\| \nabla f(\widetilde{x}^{k+1}) - \nabla f(x^k) \right\| + \gamma_{k+1} \left\| \nabla f(x^k) - \widehat{m}^{k+1} \right\| + \left\| \widetilde{m}^{k+1} - \widetilde{\varpi}^{k+1} \right\| + \frac{\sqrt{\widehat{\pi}_{k+1}} + \varepsilon}{\alpha_{k+1}} \left\| \widetilde{x}^{k+1} - \widetilde{x}^k \right\|$$

$$+ \frac{\lambda_{k+1}(\sqrt{\widehat{\pi}_{k+1}} + \varepsilon)}{\alpha_{k+1}} \left\| \widetilde{x}^k - x^k \right\|$$

$$\leq \left\| \widetilde{m}^{k+1} - \widetilde{\varpi}^{k+1} \right\| + L \left\| \widetilde{x}^{k+1} - x^k \right\| + \gamma_{k+1} \left\| m^{k+1} - \nabla f(x^k) \right\| + \gamma_{k+1} \left\| \widehat{m}^{k+1} - m^{k+1} \right\|$$

$$+ \frac{\sqrt{\widehat{\pi}_{k+1}} + \varepsilon}{\alpha_{k+1}} \left\| \widetilde{x}^{k+1} - \widetilde{x}^k \right\| + \frac{\lambda_{k+1}(\sqrt{\widehat{\pi}_{k+1}} + \varepsilon)}{\alpha_{k+1}} \left\| \widetilde{x}^k - x^k \right\|$$

$$\overset{(5)}{\leq} \left\| \widetilde{m}^{k+1} - \widetilde{\varpi}^{k+1} \right\| + L \left\| \widetilde{x}^{k+1} - x^k \right\| + \mu\gamma_{k+1} \left\| \nabla f(x^k) - m^k \right\| + \frac{\gamma_{k+1}\mu^{k+1}}{1 - \mu^{k+1}} \left\| m^{k+1} \right\|$$

$$+ \frac{\sqrt{\widehat{\pi}_{k+1}} + \varepsilon}{\alpha_{k+1}} \left\| \widetilde{x}^{k+1} - \widetilde{x}^k \right\| + \frac{\lambda_{k+1}(\sqrt{\widehat{\pi}_{k+1}} + \varepsilon)}{\alpha_{k+1}} \left\| \widetilde{x}^k - x^k \right\|.$$

Applying the conditional expectation operator and using (4) to bound the MSE terms, we can obtain

$$\mathbb{E}_k \left[ \left\| \omega_1^{k+1} \right\| + \left\| \omega_2^{k+1} \right\| \right]$$

$$\leq \mathbb{E}_k \left[ \left\| \widetilde{m}^{k+1} - \widetilde{\varpi}^{k+1} \right\| + \left\| \widehat{m}^{k+1} - \widehat{\varpi}^{k+1} \right\| + \mu\left(\gamma_{k+1} + 1\right) \left\| \nabla f(x^k) - m^k \right\| + \left( L + \frac{\sqrt{\widehat{\pi}_{k+1}} + \varepsilon}{\alpha} \right) \right.$$

$$\left\| x^{k+1} - x^k \right\| + \frac{\mu^{k+1}(\gamma_{k+1} + 1)}{1 - \mu^{k+1}} \left\| m^{k+1} \right\| + L \left\| \widetilde{x}^{k+1} - x^k \right\| + \frac{\sqrt{\widehat{\pi}_{k+1}} + \varepsilon}{\alpha_{k+1}} \left\| \widetilde{x}^{k+1} - \widetilde{x}^k \right\|$$

$$\left. + \frac{\lambda_{k+1}(\sqrt{\widehat{\pi}_{k+1}} + \varepsilon)}{\alpha_{k+1}} \left\| \widetilde{x}^k - x^k \right\| \right]$$

$$\leq \mathbb{E}_k \left[ \left( \mu\left(\gamma_{k+1} + 1\right) + V_2 \right) \left\| \nabla f(x^k) - m^k \right\| + \left( L + \frac{\sqrt{\widehat{\pi}_{k+1}} + \varepsilon}{\alpha} \right) \left\| x^{k+1} - x^k \right\| + \frac{\mu^{k+1}(\gamma_{k+1} + 1)}{1 - \mu^{k+1}} \right.$$

$$\left\| m^{k+1} \right\| + L \left\| \widetilde{x}^{k+1} - x^k \right\| + \frac{\sqrt{\widehat{\pi}_{k+1}} + \varepsilon}{\alpha_{k+1}} \left\| \widetilde{x}^{k+1} - \widetilde{x}^k \right\| + \frac{\lambda_{k+1}(\sqrt{\widehat{\pi}_{k+1}} + \varepsilon)}{\alpha_{k+1}} \left\| \widetilde{x}^k - x^k \right\|$$

$$\left. + V_2 \left\| x^k - x^{k-1} \right\| \right] + \Gamma_k$$

$$\leq \varrho \left( \mathbb{E}_k \left\| \widetilde{x}^{k+1} - \widetilde{x}^k \right\| + \mathbb{E}_k \left\| x^{k+1} - x^k \right\| + \mathbb{E}_k \left\| \widetilde{x}^{k+1} - x^k \right\| + \mathbb{E}_k \left\| x^{k+1} - \widetilde{x}^k \right\| + \left\| \widetilde{x}^k - x^k \right\| \right.$$

$$\left. + \left\| \nabla f(x^k) - m^k \right\| + \left\| x^k - x^{k-1} \right\| + \mathbb{E}_k \left\| m^{k+1} \right\| \right) + \Gamma_k,$$

where $\varrho = \mu(\overline{\gamma} + 1) + L + \frac{\mu^{k+1}(\overline{\gamma}+1)}{1-\mu^{k+1}} + \frac{\sqrt{\widehat{\pi}_{k+1}}+\varepsilon}{\underline{\alpha}} + \frac{\sqrt{\widehat{\pi}_{k+1}}+\varepsilon}{\alpha} + V_2$. This completes the proof. $\square$

Then, we present the supermartingale convergence theorem dav, which is used to establish almost sure (a.s.) convergence for STNAdam (Algorithm 1).

**Lemma A.3.** [Supermartingale convergence] *Let* $\left\{ X^k \right\}$ *and* $\left\{ Y^k \right\}$ *be the sequences of bounded, nonnegative random variables such that* $X^k, Y^k$ *depend only on the first* $k$ *iterations of Algorithm 1. If for* $k \geq 0$,

$$\mathbb{E}_k[X^{k+1} + Y^k] \leq X^k, \tag{A.35}$$

*then* $\sum_{k=0}^{\infty} Y^k < +\infty$ *a.s. and* $\left\{ X^k \right\}$ *converges a.s.*

Then, we derive some important properties for the subgradient of $\Phi^{k+1}$ and the set of accumulation points of $\left\{ \theta^k \right\}$, defined by

$$\Omega := \{ \hat{\theta} : \exists \left\{ \theta^{k_l} \right\} \subseteq \left\{ \theta^k \right\} \text{ s.t. } \theta^{k_l} \to \hat{\theta} \text{ as } l \to \infty \}.$$

**Lemma A.4.** [Lemma 4, Properties of $\Omega$] *Under Assumption 1, we have*

(1) $\sum_{k=1}^{\infty} \left\| \widetilde{x}^k - \widetilde{x}^{k-1} \right\|^2 < \infty$ *a.s.,* $\left\| \widetilde{x}^k - \widetilde{x}^{k-1} \right\| \to 0$ *a.s.;*

(2) $\mathbb{E}[\Phi(\theta^k)] \to \Phi^*$, *where* $\Phi^* \in [\Phi_0, \infty)$;

(3) $\mathbb{E}[\text{dist}(0, \partial\Phi(\theta^k))] \to 0$;

(4) $\Omega$ *is nonempty, and* $\mathbb{E}[\text{dist}(0, \partial\Phi(\theta^*))] = 0, \forall \theta^* \in \Omega$;

(5) $\text{dist}(\theta^k, \Omega) \to 0$ *a.s.;*

(6) $\Omega$ is a.s. compact and connected;

(7) $\mathbb{E}[\Phi(\theta^*)] = \Phi^*,\ \forall \theta^* \in \Omega$.

*Proof.* By Lemma 2, we have claim (1) holds.

According to (A.15), the supermartingale convergence theorem ensures $\{G^k\}$ converges to a finite, positive random variable. Because $\|\widetilde{x}^k - \widetilde{x}^{k-1}\| \to 0$ a.s., $\|x^k - x^{k-1}\| \to 0$ a.s., $\|\widetilde{x}^k - x^{k-1}\| \to 0$ a.s., $\|x^k - \widetilde{x}^{k-1}\| \to 0$ a.s., $\|\widetilde{x}^{k-1} - x^{k-1}\| \to 0$ a.s., $\|\nabla f(x^{k-1}) - m^{k-1}\| \to 0$ a.s., $\|x^{k-1} - x^{k-2}\| \to 0$ a.s., and $\|m^k\| \to 0$ a.s. Moreover, $\widehat{\varpi}$ and $\widetilde{\varpi}$ are variance-reduced, so $\mathbb{E}[\Upsilon_k] \to 0$, we can say

$$\lim_{k\to\infty} \mathbb{E}[G^k] = \lim_{k\to\infty} \mathbb{E}[\Phi(\theta^k)] \in [\Phi_0, \infty),$$

which implys claim (2).

Claim (3) holds because, by Lemma 3, we know that

$$\mathbb{E}\left\|\omega^k\right\|$$
$$\leq \varrho \mathbb{E}\left(\left\|\widetilde{x}^k - \widetilde{x}^{k-1}\right\| + \left\|x^k - x^{k-1}\right\| + \left\|\widetilde{x}^k - x^{k-1}\right\| + \left\|x^k - \widetilde{x}^{k-1}\right\| + \left\|\widetilde{x}^{k-1} - x^{k-1}\right\|$$
$$+ \left\|\nabla f(x^{k-1}) - m^{k-1}\right\| + \left\|x^{k-1} - x^{k-2}\right\| + \left\|m^k\right\|\right) + \mathbb{E}[\Gamma_{k-1}].$$

Combined with $\mathbb{E}\left\|\widetilde{x}^k - \widetilde{x}^{k-1}\right\| \to 0$, $\mathbb{E}\left\|x^k - x^{k-1}\right\| \to 0$, $\mathbb{E}\left\|\widetilde{x}^k - x^{k-1}\right\| \to 0$, $\mathbb{E}\left\|x^k - \widetilde{x}^{k-1}\right\| \to 0$, $\mathbb{E}\left\|\widetilde{x}^{k-1} - x^{k-1}\right\| \to 0$, $\mathbb{E}\left\|\nabla f(x^{k-1}) - m^{k-1}\right\| \to 0$, $\mathbb{E}\left\|m^k\right\| \to 0$ and $\mathbb{E}[\Gamma_{k-1}] \to 0$. This ensures that $\mathbb{E}\left\|w^k\right\| \to 0$. Since $w^k$ is one element of $\partial\Phi(\theta^k)$, we obtain $\mathbb{E}\mathrm{dist}(0, \partial\Phi(\theta^k)) \leq \mathbb{E}\left\|w^k\right\| \to 0$.

To prove claim (4), suppose $\widetilde{x}^*$ is a limit point of the sequence $\{\widetilde{x}^k\}$ (a limit point must exist because we assume the objective function $\Phi$ is coercive, so the sequence $\{\widetilde{x}^k\}$ is bounded). This means there exists a subsequence $\{\widetilde{x}^{k_j}\}$ satisfying $\lim_{j\to\infty} \widetilde{x}^{k_j} = \widetilde{x}^*$. Furthermore, by the variance-reduced property of $\widetilde{\varpi}^{k_j}$, we have $\mathbb{E}\left\|\widetilde{\varpi}^{k_j} - \widetilde{m}^{k_j}\right\|^2 \to 0$. Because $f$ and $g$ are lower semicontinuous, we have

$$\liminf_{j\to\infty} f(\widetilde{x}^{k_j}) \geq f(\widetilde{x}^*),$$
$$\liminf_{j\to\infty} g(\widetilde{x}^{k_j}) \geq g(\widetilde{x}^*). \tag{A.36}$$

By the update rule for $\widetilde{x}^{k_j}$, letting $x = \widetilde{x}^*$, we have

$$g(\widetilde{x}^{k_j}) + \langle \widetilde{x}^{k_j}, \widetilde{\varpi}^{k_j} \rangle + \frac{\sqrt{\widehat{\pi}_{k_j}} + \varepsilon}{2\alpha_{k_j}} \left\|\widetilde{x}^{k_j} - \overline{x}^{k_j}\right\|^2 \leq g(\widetilde{x}^*) + \langle \widetilde{x}^*, \widetilde{\varpi}^{k_j} \rangle + \frac{\sqrt{\widehat{\pi}_{k_j}} + \varepsilon}{2\alpha_{k_j}} \left\|\widetilde{x}^* - \overline{x}^{k_j}\right\|^2.$$

Taking the expectation and taking the limit $j \to \infty$,

$$\limsup_{j\to\infty} g(\widetilde{x}^{k_j}) \leq \limsup_{j\to\infty} g(\widetilde{x}^*) + \langle \widetilde{x}^* - \widetilde{x}^{k_j}, \widetilde{m}^{k_j} \rangle + \langle \widetilde{x}^* - \widetilde{x}^{k_j}, \widetilde{\varpi}^{k_j} - \widetilde{m}^{k_j} \rangle + \frac{\sqrt{\widetilde{\pi}_{k_j}} + \varepsilon}{2\alpha_{k_j}} \left\|\overline{x}^{k_j} - \widetilde{x}^*\right\|^2.$$

The second term on the right goes to zero because $\widetilde{x}^{k_j} \to \widetilde{x}^*$ and $\{\widetilde{m}^{k_j}\}$ is bounded. The thrid term is zero almost surely because it is bounded above by $\left\|\widetilde{x}^* - \widetilde{x}^{k_j}\right\|^2$, and $\widetilde{\varpi}^{k_j} - \widetilde{m}^{k_j} \to 0$ a.s. So $\limsup_{j\to\infty} g(\widetilde{x}^{k_j}) \leq g(\widetilde{x}^*)$ a.s., which, together with (A.36), implies that $\lim_{j\to\infty} g(\widetilde{x}^{k_j}) = g(\widetilde{x}^*)$ a.s. Similarly, we have $\lim_{j\to\infty} f(\widetilde{x}^{k_j}) = f(\widetilde{x}^*)$ a.s. A similar conclusion holds at $x^{k_j}$, hence

$$\lim_{j\to\infty} \Phi(\widetilde{x}^{k_j}) = \Phi(\widetilde{x}^*)\ \text{a.s},$$
$$\lim_{j\to\infty} \Phi(x^{k_j}) = \Phi(x^*)\ \text{a.s.} \tag{A.37}$$

Claim (3) ensures that $\mathbb{E}\mathrm{dist}(0, \partial\Phi(\theta^k)) \to 0$. Combining (A.37) and the fact that the subdifferential of $\Phi$ is closed, we have $\mathbb{E}\mathrm{dist}(0, \partial\Phi(\theta^*)) = 0$.

Claims (5) and (6) hold for any sequence satisfying $\|\widetilde{x}^k - \widetilde{x}^{k-1}\| \to 0$ a.s. and $\|x^k - x^{k-1}\| \to 0$ a.s. (this fact is used in the same context in Bolte et al. (2014); Damek (2016).

Finally, we must show that $\Phi$ has constant expectation over $\Omega$. From claim (2), we have $\mathbb{E}[\Phi(\theta^k)] \to \Phi^*$, which implies $\mathbb{E}[\Phi(\theta^{k_j})] \to \Phi^*$ for every subsequence $\{\theta^{k_j}\}$ converging to some $\theta^* \in \Omega$. In the proof of claim (4), we show that $\Phi(\theta^{k_j}) \to \Phi(\theta^*)$ a.s., so $\mathbb{E}[\Phi(\theta^*)] = \Phi^*$ for all $\theta^* \in \Omega$. $\qquad\square$

**Lemma A.5.** [Lemma 5, KŁ inequality] *Suppose that $\Phi$ is semialgebraic with KŁ exponent $\vartheta \in [0, 1)$. If $\widetilde{x}^k$ is not a stationary point of $\Phi$ after a finite number of iterations, then there must exist a $l > 0$ and a nondegenerate concave function $\varphi$ such that*

$$\varphi'(\mathbb{E}[\Phi(\theta^k) - \Phi_k^*])\mathbb{E}[\mathrm{dist}(0, \partial\Phi(\theta^k))] \geq 1, \ \ \forall k \geq l,$$

*where $\Phi_k^*$ is a nondecreasing sequence converging to $\mathbb{E}[\Phi(\theta^*)]$ for any $\theta^* \in \Omega$.*

*Proof.* First, we show that $\mathbb{E}[\Phi(\theta^k)]$ satisfies the KŁ property. Recall that $b$ is the minibatch size. Let $\overline{n} = \binom{n}{b}$ be the number of possible gradient estimates in one iteration, and let $\{\theta_i^k\}_{i=1}^{\overline{n}^k}$ be the set of possible values for $\theta^k$. Considering $\mathbb{E}[\Phi]$ as a function of $\{\theta_i^k\}_{i=1}^{\overline{n}^k}$, we have

$$\mathbb{E}[\Phi(\theta^k)] = \frac{1}{\overline{n}^k} \sum_{i=1}^{\overline{n}^k} \Phi(\theta_i^k).$$

Because $\mathbb{E}[\Phi(\theta^k)]$ can be written as $\sum_i f_i(x^i)$ where $f_i$ are KŁ functions with exponent $\vartheta$, $\mathbb{E}[\Phi(\theta^k)]$ (as a function of $\{\theta_i^k\}_{i=1}^{\overline{n}^k}$) is also KŁ with exponent $\vartheta$. Hence, $\mathbb{E}[\Phi]$ satisfies the KŁ inequality at every point in its domain. Therefore, for every point $(\theta_1^k, \theta_2^k, \ldots, \theta_{\overline{n}^k}^k)$ in a neighborhood $U_k$ of $(\overline{\theta}_1^k, \overline{\theta}_2^k, \ldots, \overline{\theta}_{\overline{n}^k}^k)$ and satisfying

$$\frac{1}{\overline{n}^k} \sum_{i=1}^{\overline{n}^k} \Phi(\overline{\theta}_i^k) < \frac{1}{\overline{n}^k} \sum_{i=1}^{\overline{n}^k} \Phi(\theta_i^k) < \frac{1}{\overline{n}^k} \sum_{i=1}^{\overline{n}^k} \Phi(\overline{\theta}_i^k) + \varepsilon_k \tag{A.38}$$

for some $\varepsilon_k > 0$, the KŁ inequality holds with the desingularizing function $\varphi_k$:

$$\varphi'\left(\frac{1}{\overline{n}^k} \sum_{i=1}^{\overline{n}^k} \Phi(\theta_i^k) - \frac{1}{\overline{n}^k} \sum_{i=1}^{\overline{n}^k} \Phi(\overline{\theta}_i^k)\right) \mathrm{dist}\left(0, \frac{1}{\overline{n}^k} \sum_{i=1}^{\overline{n}^k} \partial\Phi(\theta_i^k)\right) \geq 1. \tag{A.39}$$

There always exists a choice of $(\overline{\theta}_1^k, \overline{\theta}_2^k, \ldots, \overline{\theta}_{\overline{n}^k}^k)$ satisfying (A.38) unless $\mathbb{E}[\Phi(\theta^k)])$ is a local minimum. Lemma A.4, claim (5), implies $\mathrm{dist}(\theta^k, \Omega) \to 0$ a.s., and claims (2) and (7) imply $\mathbb{E}[\Phi(\theta^k)] \to \mathbb{E}[\Phi(\theta^*)]$, so we can choose $\overline{\theta}^k$ such that $\frac{1}{\overline{n}^k} \sum_{i=1}^{\overline{n}^k} \Phi(\overline{\theta}_i^k) \to \mathbb{E}[\Phi(\theta^*)]$ as well. To summarize, we have shown that there exists a sequence $(\overline{\theta}_1^k, \overline{\theta}_2^k, \ldots, \overline{\theta}_{\overline{n}^k}^k)$ such that

(1) the point $(\theta_1^k, \theta_2^k, \ldots, \theta_{\overline{n}^k}^k)$ lies in a neighborhood $U_k$ of $(\overline{\theta}_1^k, \overline{\theta}_2^k, \ldots, \overline{\theta}_{\overline{n}^k}^k)$;

(2) the inequality (A.38) is satisfied;

(3) we have $\frac{1}{\overline{n}^k} \sum_{i=1}^{\overline{n}^k} \Phi(\overline{\theta}_i^k) \to \mathbb{E}[\Phi(\theta^*)]$.

Points (1) and (2) imply the KŁ inequality (A.39). This ensures that the KŁ inequality holds at every iteration, but the desingularizing function $\varphi_k$ changes every iteration. We now show that the KŁ inequality holds using a single function $\varphi$.

Because $\Phi$ is semialgebraic with KŁ exponent $\vartheta$, each desingularizing function is of the form $\varphi_k(s) = a_k s^{1-\vartheta}$. Each $a_k$ is bounded, so $a_{\max} = \max\{a_k\}_{k\geq 1}$ is bounded, and inequality (A.39) holds with the desingularizing function $\varphi_{\max}(s) = a_{\max} s^{1-\vartheta}$.

Let $\Phi_k^* = \min_{j\geq k} \frac{1}{\overline{n}^j} \sum_{i=1}^{\overline{n}^j} \Phi(\overline{\theta}_i^j)$. It is clear that $\Phi_k^*$ is nondecreasing and $\Phi_k^* \to \mathbb{E}[\Phi(\theta^*)]$. From point (3), we can say there exists an index $l$ and a constant $a$ such that for all $k \geq l$,

$$a\left(\frac{1}{\overline{n}^k} \sum_{i=1}^{\overline{n}^k} \Phi(\theta_i^k) - \Phi_k^*\right)^{-\vartheta} \geq a_{\max}\left(\frac{1}{\overline{n}^k} \sum_{i=1}^{\overline{n}^k} \Phi(\theta_i^k) - \frac{1}{\overline{n}^k} \sum_{i=1}^{\overline{n}^k} \Phi(\overline{\theta}_i^k)\right)^{-\vartheta}. \tag{A.40}$$

The constant $a$ exists, we can take $a$ to be

$$\max_{k \geq 1}\left\{\left(\frac{\frac{1}{\overline{n}^k}\sum_{i=1}^{\overline{n}^k}\Phi(\theta_i^k)-\Phi_k^*}{\frac{1}{\overline{n}^k}\sum_{i=1}^{\overline{n}^k}\Phi(\theta_i^k)-\frac{1}{\overline{n}^k}\sum_{i=1}^{\overline{n}^k}\Phi(\overline{\theta}_i^k)}\right)^{\vartheta}\right\}_{k\geq 1}, \tag{A.41}$$

which is bounded. To see this, we acknowledge that this ratio is bounded for every $k$, and

$$\lim_{k\to\infty}\left(\frac{\frac{1}{\overline{n}^k}\sum_{i=1}^{\overline{n}^k}\Phi(\theta_i^k)-\Phi_k^*}{\frac{1}{\overline{n}^k}\sum_{i=1}^{\overline{n}^k}\Phi(\theta_i^k)-\frac{1}{\overline{n}^k}\sum_{i=1}^{\overline{n}^k}\Phi(\overline{\theta}_i^k)}\right)=\lim_{k\to\infty}\left(\frac{\frac{1}{\overline{n}^k}\sum_{i=1}^{\overline{n}^k}\Phi(\theta_i^k)-\mathbb{E}[\Phi(\theta^*)]}{\frac{1}{\overline{n}^k}\sum_{i=1}^{\overline{n}^k}\Phi(\theta_i^k)-\mathbb{E}[\Phi(\theta^*)]}\right)=1. \tag{A.42}$$

Therefore, with $\varphi(s)=as^{1-\vartheta}$, we have

$$\varphi'(\mathbb{E}[\Phi(\theta^k)-\Phi_k^*])\text{dist}(0,\mathbb{E}\partial\Phi(\theta^k))\geq\varphi'_{\max}(\mathbb{E}[\Phi(\theta^k)-\Phi_k^*])\text{dist}(0,\mathbb{E}\partial\Phi(\theta^k))\geq 1,\;\;\forall k>l.$$

The desired inequality follows from Jensen's inequality and the convexity of $\theta\longmapsto\text{dist}(0,\theta)$. $\quad\square$

### A.2 PROOF OF THEOREM 1 (CONVERGE TO A STATIONARY POINT)

**Theorem A.1.** [Theorem 1] *Assume that the conditions of Lemma A.5 hold. Then, there hold:*

(i) *Either $\widetilde{x}^k$ is a stationary point after a finite number of iterations or $\{\widetilde{x}^k\}$ satisfies the finite-length property in expectation:*

$$\sum_{k=0}^{\infty}\mathbb{E}\left\|\widetilde{x}^{k+1}-\widetilde{x}^k\right\|<\infty,$$

*and there exists an integer $l$ so that, for all $i>l$,*

$$\sum_{k=l}^{i}\left(\mathbb{E}\left\|\widetilde{x}^{k+1}-\widetilde{x}^k\right\|+\mathbb{E}\left\|x^{k+1}-x^k\right\|+\mathbb{E}\left\|\widetilde{x}^{k+1}-x^k\right\|+\mathbb{E}\left\|x^{k+1}-\widetilde{x}^k\right\|+\mathbb{E}\left\|\widetilde{x}^k-x^k\right\|\right.$$

$$\left.+\mathbb{E}\left\|\nabla f(x^k)-m^k\right\|+\mathbb{E}\left\|x^k-x^{k-1}\right\|+\left\|m^{k+1}\right\|\right)\leq T_i^l$$

$$\leq\sqrt{\mathbb{E}\left\|\widetilde{x}^l-\widetilde{x}^{l-1}\right\|^2}+\sqrt{\mathbb{E}\left\|x^l-x^{l-1}\right\|^2}+\sqrt{\mathbb{E}\left\|\widetilde{x}^l-x^{l-1}\right\|^2}+\sqrt{\mathbb{E}\left\|x^l-\widetilde{x}^{l-1}\right\|^2}$$

$$+\sqrt{\mathbb{E}\left\|\widetilde{x}^{l-1}-x^{l-1}\right\|^2}+\sqrt{\mathbb{E}\left\|\nabla f(x^{l-1})-m^{l-1}\right\|^2}+\sqrt{\mathbb{E}\left\|x^{l-1}-x^{l-2}\right\|^2}$$

$$+\sqrt{\mathbb{E}\left\|m^l\right\|^2}+\frac{2\sqrt{n}}{K\rho}\sqrt{\mathbb{E}[\Upsilon_{l-1}]}+\frac{4K}{A}\triangle^{l,i+1}, \tag{A.43}$$

*where*

- *$K=\varrho+\frac{2\sqrt{nV_{\Upsilon}}}{\rho}$, $\varrho$ is seen in Lemma A.2;*

- *$A=\min_{i\in[8]}\{A_i\}>0$, defined in Lemma A.1;*

- *$\triangle^{\overline{k},\underline{k}}=\mathbb{E}[G_{\overline{k}}^{\overline{k}}-\Phi_{\overline{k}}^*]-\mathbb{E}[G_{\underline{k}}^{\underline{k}}-\Phi_{\underline{k}}^*]$ for any $\overline{k}\geq\underline{k}\in\mathbb{Z}_+$.*

(ii) *$\{\widetilde{x}^k\}$ generated by Algorithm 1 converge to a stationary point of $\Phi$ in expectation.*

*Proof.* (i) If $\vartheta\in(0,\frac{1}{2})$, then $\Phi$ satisfies the KŁ property with exponent $\frac{1}{2}$, so we consider only the case $\vartheta\in[\frac{1}{2},1)$. By Lemma A.5, there exists a function $\varphi_0(r)=ar^{1-\theta}$ such that

$$\varphi_0'(\mathbb{E}[\Phi(\theta^k)-\Phi_k^*])\mathbb{E}\text{dist}(0,\partial\Phi(\theta^k))\geq 1,\;\;\forall k>l.$$

Lemma A.2 provides a bound on $\mathbb{E}\text{dist}(0,\partial\Phi(\theta^k))$.

$$\mathbb{E}\mathrm{dist}(0, \partial\Phi(\theta^k)) \leq \mathbb{E}\left\|w^k\right\|$$

$$\leq \varrho\mathbb{E}\left(\left\|\widetilde{x}^k - \widetilde{x}^{k-1}\right\| + \left\|x^k - x^{k-1}\right\| + \left\|\widetilde{x}^k - x^{k-1}\right\| + \left\|x^k - \widetilde{x}^{k-1}\right\| + \left\|\widetilde{x}^{k-1} - x^{k-1}\right\|\right.$$

$$\left. + \left\|\nabla f(x^{k-1}) - m^{k-1}\right\| + \left\|x^{k-1} - x^{k-2}\right\| + \left\|m^k\right\|\right) + \mathbb{E}[\Gamma_{k-1}]$$

$$\leq \varrho\left(\sqrt{\mathbb{E}\left\|\widetilde{x}^k - \widetilde{x}^{k-1}\right\|^2} + \sqrt{\mathbb{E}\left\|x^k - x^{k-1}\right\|^2} + \sqrt{\mathbb{E}\left\|\widetilde{x}^k - x^{k-1}\right\|^2} + \sqrt{\mathbb{E}\left\|x^k - \widetilde{x}^{k-1}\right\|^2}\right.$$

$$\left. + \sqrt{\mathbb{E}\left\|\widetilde{x}^{k-1} - x^{k-1}\right\|^2} + \sqrt{\mathbb{E}\left\|\nabla f(x^{k-1}) - m^{k-1}\right\|^2} + \sqrt{\mathbb{E}\left\|x^{k-1} - x^{k-2}\right\|^2} + \sqrt{\mathbb{E}\left\|m^k\right\|^2}\right)$$

$$+ \sqrt{n\mathbb{E}[\Upsilon_{k-1}]}. \tag{A.44}$$

The final inequality is Jensen's inequality. Because $\Gamma_k = \sum_{i=1}^n v_i^k$ for some nonnegative random variables $v_i^k$, we can say $\mathbb{E}[\Gamma_k] = \mathbb{E}\sum_{i=1}^n v_i^k \leq \mathbb{E}\sqrt{n\sum_{i=1}^n (v_i^k)^2} \leq \sqrt{n\mathbb{E}[\Upsilon_k]}$. So we can bound the term $\sqrt{\mathbb{E}[\Upsilon_k]}$ using (5):

$$\sqrt{\mathbb{E}[\Upsilon_k]} \leq \sqrt{(1-\rho)\mathbb{E}[\Upsilon_{k-1}] + V_\Upsilon\mathbb{E}\left(\left\|\nabla f(x^{k-1}) - m^{k-1}\right\|^2 + \left\|x^{k-1} - x^{k-2}\right\|^2\right)}$$

$$\leq \sqrt{(1-\rho)}\sqrt{\mathbb{E}[\Upsilon_{k-1}]} + \sqrt{V_\Upsilon}\left(\sqrt{\mathbb{E}\left\|\nabla f(x^{k-1}) - m^{k-1}\right\|^2} + \sqrt{\mathbb{E}\left\|x^{k-1} - x^{k-2}\right\|^2}\right)$$

$$\leq (1 - \frac{\rho}{2})\sqrt{\mathbb{E}[\Upsilon_{k-1}]} + \sqrt{V_\Upsilon}\left(\sqrt{\mathbb{E}\left\|\nabla f(x^{k-1}) - m^{k-1}\right\|^2} + \sqrt{\mathbb{E}\left\|x^{k-1} - x^{k-2}\right\|^2}\right). \tag{A.45}$$

The final inequality uses the fact that $\sqrt{1-\rho} = 1 - \frac{\rho}{2} - \frac{\rho^2}{8} - \cdots$. This implies that

$$\sqrt{n\mathbb{E}[\Upsilon_{k-1}]}$$

$$\leq \frac{2\sqrt{n}}{\rho}\left(\sqrt{\mathbb{E}[\Upsilon_{k-1}]} - \sqrt{\mathbb{E}[\Upsilon_k]}\right) + \frac{2\sqrt{nV_\Upsilon}}{\rho}\left(\sqrt{\mathbb{E}\left\|\nabla f(x^{k-1}) - m^{k-1}\right\|^2} + \sqrt{\mathbb{E}\left\|x^{k-1} - x^{k-2}\right\|^2}\right). \tag{A.46}$$

Then, from (A.44) and (A.46), we have

$$\mathbb{E}\mathrm{dist}(0, \partial\Phi(\theta^k))$$

$$\leq \varrho\left(\sqrt{\mathbb{E}\left\|\widetilde{x}^k - \widetilde{x}^{k-1}\right\|^2} + \sqrt{\mathbb{E}\left\|x^k - x^{k-1}\right\|^2} + \sqrt{\mathbb{E}\left\|\widetilde{x}^k - x^{k-1}\right\|^2} + \sqrt{\mathbb{E}\left\|x^k - \widetilde{x}^{k-1}\right\|^2}\right.$$

$$\left. + \sqrt{\mathbb{E}\left\|\widetilde{x}^{k-1} - x^{k-1}\right\|^2} + \sqrt{\mathbb{E}\left\|\nabla f(x^{k-1}) - m^{k-1}\right\|^2} + \sqrt{\mathbb{E}\left\|x^{k-1} - x^{k-2}\right\|^2} + \sqrt{\mathbb{E}\left\|m^k\right\|^2}\right)$$

$$+ \frac{2\sqrt{nV_\Upsilon}}{\rho}\left(\sqrt{\mathbb{E}\left\|\nabla f(x^{k-1}) - m^{k-1}\right\|^2} + \sqrt{\mathbb{E}\left\|x^{k-1} - x^{k-2}\right\|^2}\right) + \frac{2\sqrt{n}}{\rho}\left(\sqrt{\mathbb{E}[\Upsilon_{k-1}]} - \sqrt{\mathbb{E}[\Upsilon_k]}\right)$$

$$\leq K\left(\sqrt{\mathbb{E}\left\|\widetilde{x}^k - \widetilde{x}^{k-1}\right\|^2} + \sqrt{\mathbb{E}\left\|x^k - x^{k-1}\right\|^2} + \sqrt{\mathbb{E}\left\|\widetilde{x}^k - x^{k-1}\right\|^2} + \sqrt{\mathbb{E}\left\|x^k - \widetilde{x}^{k-1}\right\|^2}\right.$$

$$\left. + \sqrt{\mathbb{E}\left\|\widetilde{x}^{k-1} - x^{k-1}\right\|^2} + \sqrt{\mathbb{E}\left\|\nabla f(x^{k-1}) - m^{k-1}\right\|^2} + \sqrt{\mathbb{E}\left\|x^{k-1} - x^{k-2}\right\|^2} + \sqrt{\mathbb{E}\left\|m^k\right\|^2}\right)$$

$$+ \frac{2\sqrt{n}}{\rho}\left(\sqrt{\mathbb{E}[\Upsilon_{k-1}]} - \sqrt{\mathbb{E}[\Upsilon_k]}\right),$$

where $K = \varrho + \frac{2\sqrt{nV_\Upsilon}}{\rho}$. Define $C^k$ to be the right side of this inequality:

$$C^k = K\left(\sqrt{\mathbb{E}\left\|\widetilde{x}^k - \widetilde{x}^{k-1}\right\|^2} + \sqrt{\mathbb{E}\left\|x^k - x^{k-1}\right\|^2} + \sqrt{\mathbb{E}\left\|\widetilde{x}^k - x^{k-1}\right\|^2} + \sqrt{\mathbb{E}\left\|x^k - \widetilde{x}^{k-1}\right\|^2}\right.$$

$$\left. + \sqrt{\mathbb{E}\left\|\widetilde{x}^{k-1} - x^{k-1}\right\|^2} + \sqrt{\mathbb{E}\left\|\nabla f(x^{k-1}) - m^{k-1}\right\|^2} + \sqrt{\mathbb{E}\left\|x^{k-1} - x^{k-2}\right\|^2} + \sqrt{\mathbb{E}\left\|m^k\right\|^2}\right)$$

$$+ \frac{2\sqrt{n}}{\rho}\left(\sqrt{\mathbb{E}[\Upsilon_{k-1}]} - \sqrt{\mathbb{E}[\Upsilon_k]}\right).$$

We then have

$$\varphi_0'(\mathbb{E}[\Phi(\theta^k) - \Phi_k^*])C^k \geq 1, \ \ \forall k > l. \tag{A.47}$$

By the definition of $\varphi_0$, this is equivalent to

$$\frac{a(1 - \vartheta)C^k}{(\mathbb{E}[\Phi(\theta^k) - \Phi_k^*])^\theta} \geq 1, \ \ \forall k > l. \tag{A.48}$$

We would like to hold the inequality above for $G^k$ rather than $\Phi(\theta^k)$. Replace $\mathbb{E}\Phi(\theta^k)$ with $\mathbb{E}[G^k]$ by introducing a term of

$$\mathcal{O}\left(\left(\mathbb{E}\left[\left\|\nabla f(x^k) - m^k\right\|^2 + \left\|\widetilde{x}^k - x^k\right\|^2 + \left\|m^k\right\|^2 + \left\|x^k - x^{k-1}\right\|^2 + \Upsilon_k\right]\right)^\theta\right)$$

in the denominator. We show that inequality (A.48) still holds after this adjustment because these terms are small compared to $C^k$. Indeed, the quantity

$$C^k \geq c_1 \left(\sqrt{\mathbb{E}\left\|\widetilde{x}^k - \widetilde{x}^{k-1}\right\|^2} + \sqrt{\mathbb{E}\left\|x^k - x^{k-1}\right\|^2} + \sqrt{\mathbb{E}\left\|\widetilde{x}^k - x^{k-1}\right\|^2} + \sqrt{\mathbb{E}\left\|x^k - \widetilde{x}^{k-1}\right\|^2}\right.$$

$$+ \sqrt{\mathbb{E}\left\|\widetilde{x}^{k-1} - x^{k-1}\right\|^2} + \sqrt{\mathbb{E}\left\|\nabla f(x^{k-1}) - m^{k-1}\right\|^2} + \sqrt{\mathbb{E}\left\|x^{k-1} - x^{k-2}\right\|^2} + \sqrt{\mathbb{E}\left\|m^k\right\|^2}$$

$$\left. + \sqrt{\mathbb{E}[\Upsilon_{k-1}]}\right)$$

for some constant $c_1 > 0$. And because $\mathbb{E}\left\|\nabla f(x^{k-1}) - m^{k-1}\right\|^2 \to 0$, $\mathbb{E}\left\|x^k - x^{k-1}\right\|^2 \to 0$, $\mathbb{E}\left\|\widetilde{x}^k - \widetilde{x}^{k-1}\right\|^2 \to 0$, $\mathbb{E}\left\|m^k\right\|^2 \to 0$, $\mathbb{E}[\Upsilon_k] \to 0$ and $\vartheta > \frac{1}{2}$, there exists an index $l$ and constants $c_2, c_3 > 0$ such that

$$\left(\mathbb{E}[G^k - \Phi(\theta^k)]\right)^\vartheta$$

$$= \left(\mathbb{E}\left[\frac{4s}{\rho}\Upsilon_k + \left(M - 8s\left(2\underline{\gamma}^2 - 4\underline{\gamma} + 3\right)\right)\left\|\nabla f(x^k) - m^k\right\|^2 + \left(\frac{\sqrt{\widehat{\pi}_k} + \varepsilon}{2\overline{\alpha}} + \frac{\sqrt{\widehat{\pi}_k} + \varepsilon}{2\alpha} - \tau - \frac{1}{2s}\right)\right.\right.$$

$$\left.\left.\left\|\widetilde{x}^k - x^k\right\|^2 + \left(\frac{D(\mu^k)^2}{(1 - \mu^k)^2} - Z\right)\left\|m^k\right\|^2 + H\left\|x^k - x^{k-1}\right\|^2\right]\right)^\vartheta$$

$$\leq c_2 \left(\left(\mathbb{E}\left[\Upsilon_{k-1} + \left\|\nabla f(x^k) - m^k\right\|^2 + \left\|\widetilde{x}^k - x^k\right\|^2 + \left\|m^k\right\|^2 + \left\|x^k - x^{k-1}\right\|^2\right]\right)^\vartheta\right)$$

$$\leq c_3 C^k, \ \ \forall k > l.$$

The first inequality uses (5). Because the terms above are small compared to $C^k$, there exists a constant $d$ such that $c_3 < d < +\infty$ and

$$\frac{ad(1 - \vartheta)C^k}{(\mathbb{E}[\Phi(\theta^k) - \Phi_k^*])^\vartheta + (\mathbb{E}[G^k - \Phi(\theta^k)])^\vartheta} \geq 1, \ \ \forall k > l,$$

For $\vartheta \in [\frac{1}{2}, 1)$, using the fact that $(a + b)^\vartheta \leq a^\vartheta + b^\vartheta$ for all $a, b \geq 0$, we have

$$\frac{ad(1 - \vartheta)C^k}{(\mathbb{E}[G^k - \Phi_k^*])^\vartheta} = \frac{ad(1 - \vartheta)C^k}{(\mathbb{E}[\Phi(\theta^k) - \Phi_k^* + G^k - \Phi(\theta^k)])^\vartheta}$$

$$\geq \frac{ad(1 - \vartheta)C^k}{(\mathbb{E}[\Phi(\theta^k) - \Phi_k^*])^\theta + (\mathbb{E}[G^k - \Phi(\theta^k)])^\vartheta} \geq 1, \ \ \forall k > l.$$

Therefore, with $\varphi(r) = adr^{1-\vartheta}$,

$$\varphi'(\mathbb{E}[G^k - \Phi_k^*])C^k \geq 1, \ \ \forall k > l. \tag{A.49}$$

By the concavity of $\varphi$,

$$\varphi(\mathbb{E}[G^k - \Phi_k^*]) - \varphi(\mathbb{E}[G^{k+1} - \Phi_{k+1}^*])$$

$$\geq \varphi'(\mathbb{E}[G^k - \Phi_k^*])(\mathbb{E}[G^k - \Phi_k^* + \Phi_{k+1}^* - G^{k+1}])$$

$$\geq \varphi'(\mathbb{E}[G^k - \Phi_k^*])(\mathbb{E}[G^k - G^{k+1}]),$$

where the last inequality follows from the fact that $\Phi_k^*$ is nondecreasing. With $\triangle^{\overline{k},\underline{k}} = \mathbb{E}[G^{\overline{k}} - \Phi_{\overline{k}}^*] - \mathbb{E}[G^{\underline{k}} - \Phi_{\underline{k}}^*]$, we have shown

$$\triangle^{k,k+1} C^k \geq \mathbb{E}[G^k - G^{k+1}], \ \forall k > l.$$

Using Lemma A.1, we can bound $\mathbb{E}[G^k - G^{k+1}]$ below by both $\mathbb{E}\left\|\widetilde{x}^{k+1} - \widetilde{x}^k\right\|$, $\mathbb{E}\left\|x^{k+1} - x^k\right\|$, $\mathbb{E}\left\|\widetilde{x}^{k+1} - x^k\right\|$, $\mathbb{E}\left\|x^{k+1} - \widetilde{x}^k\right\|$, $\mathbb{E}\left\|\widetilde{x}^k - x^k\right\|$, $\mathbb{E}\left\|\nabla f(x^k) - m^k\right\|$, $\mathbb{E}\left\|x^k - x^{k-1}\right\|$ and $\mathbb{E}\left\|m^{k+1}\right\|$. Specifically,

$$\triangle^{k,k+1} C^k$$

$$\geq A_1 \mathbb{E}\left\|\widetilde{x}^{k+1} - \widetilde{x}^k\right\|^2 + A_2 \mathbb{E}\left\|x^{k+1} - x^k\right\|^2 + A_3 \mathbb{E}\left\|\widetilde{x}^{k+1} - x^k\right\|^2 + A_4 \mathbb{E}\left\|x^{k+1} - \widetilde{x}^k\right\|^2$$

$$+ A_5 \mathbb{E}\left\|\widetilde{x}^k - x^k\right\|^2 + A_6 \mathbb{E}\left\|\nabla f(x^k) - m^k\right\|^2 + A_7 \mathbb{E}\left\|x^k - x^{k-1}\right\|^2 + A_8 \mathbb{E}\left\|m^{k+1}\right\|^2$$

$$\geq A \left( \mathbb{E}\left\|\widetilde{x}^{k+1} - \widetilde{x}^k\right\|^2 + \mathbb{E}\left\|x^{k+1} - x^k\right\|^2 + \mathbb{E}\left\|\widetilde{x}^{k+1} - x^k\right\|^2 + \mathbb{E}\left\|x^{k+1} - \widetilde{x}^k\right\|^2 + \mathbb{E}\left\|\widetilde{x}^k - x^k\right\|^2 \right.$$

$$\left. + \mathbb{E}\left\|\nabla f(x^k) - m^k\right\|^2 + \mathbb{E}\left\|x^k - x^{k-1}\right\|^2 + \mathbb{E}\left\|m^{k+1}\right\|^2 \right),$$

(A.50)

where $A = \min\{A_i\} > 0$, $i = 1, \cdots, 8$, $A_i$ are set as in Lemma A.1. Let us use the first of these inequalities to begin. Applying Young's inequality to (A.50) yields

$$\sqrt{\mathbb{E}\left\|\widetilde{x}^{k+1} - \widetilde{x}^k\right\|^2} + \sqrt{\mathbb{E}\left\|x^{k+1} - x^k\right\|^2} + \sqrt{\mathbb{E}\left\|\widetilde{x}^{k+1} - x^k\right\|^2} + \sqrt{\mathbb{E}\left\|x^{k+1} - \widetilde{x}^k\right\|^2}$$

$$+ \sqrt{\mathbb{E}\left\|\widetilde{x}^k - x^k\right\|^2} + \sqrt{\mathbb{E}\left\|\nabla f(x^k) - m^k\right\|^2} + \sqrt{\mathbb{E}\left\|x^k - x^{k-1}\right\|^2} + \sqrt{\mathbb{E}\left\|m^{k+1}\right\|^2}$$

$$\leq 2\sqrt{\begin{array}{l} \mathbb{E}\left\|\widetilde{x}^{k+1} - \widetilde{x}^k\right\|^2 + \mathbb{E}\left\|x^{k+1} - x^k\right\|^2 + \mathbb{E}\left\|\widetilde{x}^{k+1} - x^k\right\|^2 + \mathbb{E}\left\|x^{k+1} - \widetilde{x}^k\right\|^2 + \mathbb{E}\left\|\widetilde{x}^k - x^k\right\|^2 \\ + \mathbb{E}\left\|\nabla f(x^k) - m^k\right\|^2 + \mathbb{E}\left\|x^k - x^{k-1}\right\|^2 + \mathbb{E}\left\|m^{k+1}\right\|^2 \end{array}}$$

$$\leq 2\sqrt{A^{-1} C^k \triangle^{k,k+1}} \leq \frac{C^k}{2K} + \frac{2K \triangle^{k,k+1}}{A}$$

$$\leq \frac{1}{2} \left( \sqrt{\mathbb{E}\left\|\widetilde{x}^k - \widetilde{x}^{k-1}\right\|^2} + \sqrt{\mathbb{E}\left\|x^k - x^{k-1}\right\|^2} + \sqrt{\mathbb{E}\left\|\widetilde{x}^k - x^{k-1}\right\|^2} + \sqrt{\mathbb{E}\left\|x^k - \widetilde{x}^{k-1}\right\|^2} \right.$$

$$\left. + \sqrt{\mathbb{E}\left\|\widetilde{x}^{k-1} - x^{k-1}\right\|^2} + \sqrt{\mathbb{E}\left\|\nabla f(x^{k-1}) - m^{k-1}\right\|^2} + \sqrt{\mathbb{E}\left\|x^{k-1} - x^{k-2}\right\|^2} + \sqrt{\mathbb{E}\left\|m^k\right\|^2} \right)$$

$$+ \frac{\sqrt{n}}{K\rho} \left( \sqrt{\mathbb{E}[\Upsilon_{k-1}]} - \sqrt{\mathbb{E}[\Upsilon_k]} \right) + \frac{2K \triangle^{k,k+1}}{A}.$$

(A.51)

Summing inequality (A.51) from $k = l$ to $k = i$, set

$$T_i^l = \sum_{k=l}^{i} \left( \sqrt{\mathbb{E}\left\|\widetilde{x}^{k+1} - \widetilde{x}^k\right\|^2} + \sqrt{\mathbb{E}\left\|x^{k+1} - x^k\right\|^2} + \sqrt{\mathbb{E}\left\|\widetilde{x}^{k+1} - x^k\right\|^2} + \sqrt{\mathbb{E}\left\|x^{k+1} - \widetilde{x}^k\right\|^2} \right.$$

$$\left. + \sqrt{\mathbb{E}\left\|\widetilde{x}^k - x^k\right\|^2} + \sqrt{\mathbb{E}\left\|\nabla f(x^k) - m^k\right\|^2} + \sqrt{\mathbb{E}\left\|x^k - x^{k-1}\right\|^2} + \sqrt{\mathbb{E}\left\|m^{k+1}\right\|^2} \right).$$

(A.52)

Then

$$T_i^l \leq \frac{1}{2} T_{i-1}^{l-1} + \frac{\sqrt{n}}{K\rho} \left( \sqrt{\mathbb{E}[\Upsilon_{l-1}]} - \sqrt{\mathbb{E}[\Upsilon_i]} \right) + \frac{2K \triangle^{l,i+1}}{A},$$

which implies that

$$\frac{1}{2} T_i^l \leq \frac{1}{2} \left( \sqrt{\mathbb{E}\left\|\widetilde{x}^l - \widetilde{x}^{l-1}\right\|^2} + \sqrt{\mathbb{E}\left\|x^l - x^{l-1}\right\|^2} + \sqrt{\mathbb{E}\left\|\widetilde{x}^l - x^{l-1}\right\|^2} + \sqrt{\mathbb{E}\left\|x^l - \widetilde{x}^{l-1}\right\|^2} \right.$$

$$\left. + \sqrt{\mathbb{E}\left\|\widetilde{x}^{l-1} - x^{l-1}\right\|^2} + \sqrt{\mathbb{E}\left\|\nabla f(x^{l-1}) - m^{l-1}\right\|^2} + \sqrt{\mathbb{E}\left\|x^{l-1} - x^{l-2}\right\|^2} + \sqrt{\mathbb{E}\left\|m^l\right\|^2} \right)$$

$$+ \frac{\sqrt{n}}{K\rho} \left( \sqrt{\mathbb{E}[\Upsilon_{l-1}]} - \sqrt{\mathbb{E}[\Upsilon_i]} \right) + \frac{2K}{A} \triangle^{l,i+1}.$$

Dropping the nonpositive terms $-\sqrt{\mathbb{E}[\Upsilon_i]}$, and applying Jensen's inequality to the terms on the left gives, this shows that

$$
\begin{aligned}
\sum_{k=l}^{i} & \left( \mathbb{E}\left\|\widetilde{x}^{k+1} - \widetilde{x}^k\right\| + \mathbb{E}\left\|x^{k+1} - x^k\right\| + \mathbb{E}\left\|\widetilde{x}^{k+1} - x^k\right\| + \mathbb{E}\left\|x^{k+1} - \widetilde{x}^k\right\| + \mathbb{E}\left\|\widetilde{x}^k - x^k\right\| \right. \\
& \left. + \mathbb{E}\left\|\nabla f(x^k) - m^k\right\| + \mathbb{E}\left\|x^k - x^{k-1}\right\| + \mathbb{E}\left\|m^{k+1}\right\| \right) \le T_i^l \\
\le & \sqrt{\mathbb{E}\left\|\widetilde{x}^l - \widetilde{x}^{l-1}\right\|^2} + \sqrt{\mathbb{E}\left\|x^l - x^{l-1}\right\|^2} + \sqrt{\mathbb{E}\left\|\widetilde{x}^l - x^{l-1}\right\|^2} + \sqrt{\mathbb{E}\left\|x^l - \widetilde{x}^{l-1}\right\|^2} \\
& + \sqrt{\mathbb{E}\left\|\widetilde{x}^{l-1} - x^{l-1}\right\|^2} + \sqrt{\mathbb{E}\left\|\nabla f(x^{l-1}) - m^{l-1}\right\|^2} + \sqrt{\mathbb{E}\left\|x^{l-1} - x^{l-2}\right\|^2} + \sqrt{\mathbb{E}\left\|m^l\right\|^2} \\
& + \frac{2\sqrt{n}}{K\rho}\sqrt{\mathbb{E}[\Upsilon_{l-1}]} + \frac{4K}{A}\triangle^{l,i+1}.
\end{aligned}
$$

$$(A.53)$$

The term $\lim_{i\to\infty}\triangle^{l,i+1}$ is bounded because $\mathbb{E}[G^k]$ is bounded due to Lemma A.1. Letting $i \to \infty$, we prove the assertion.

(ii) An immediate consequence of claim (i) is that the sequence $\left\{\widetilde{x}^k\right\}$ converges in expectation to a stationary point. This is because, for any $\overline{k}, \underline{k} \in \mathbb{N}$ with $\overline{k} \ge \underline{k}$, $\mathbb{E}\left\|\widetilde{x}^{\overline{k}} - \widetilde{x}^{\underline{k}}\right\| = \mathbb{E}\left\|\sum_{k=\underline{k}}^{\overline{k}-1}(\widetilde{x}^{k+1} - \widetilde{x}^k)\right\| \le \sum_{k=\underline{k}}^{\overline{k}-1} \mathbb{E}\left\|\widetilde{x}^{k+1} - \widetilde{x}^k\right\|$, and the finite length property implies this final sum converges to zero. This proves claim (ii). $\qquad\square$

### A.3 PROOF OF THEOREM 2 (CONVERGE RATE)

**Theorem A.2.** [Theorem 2] *Assume that the conditions of Lemma A.5 hold. Let $\left\{\widetilde{x}^k\right\} \to \widetilde{x}^*$, then the following statements hold:*

(i) *If $\vartheta \in (0, \frac{1}{2}]$, there exist $d_1 > 0$ and $\zeta \in [1 - \rho, 1)$ such that $\mathbb{E}\left\|\widetilde{x}^k - \tilde{x}^*\right\| \le d_1\zeta^k$.*

(ii) *If $\vartheta \in (\frac{1}{2}, 1)$, there exists a constant $d_2 > 0$ such that $\mathbb{E}\left\|\widetilde{x}^k - \tilde{x}^*\right\| \le d_2 k^{-\frac{1-\vartheta}{2\vartheta-1}}$.*

(iii) *If $\vartheta = 0$, there exists a $m \in \mathbb{N}$ such that $\mathbb{E}[\Phi(\widetilde{x}^k)] = \mathbb{E}[\Phi(\tilde{x}^*)]$ for all $k \ge l$.*

*Proof.* As in the proof of Theorem 1, if $\vartheta \in (0, \frac{1}{2})$, then $\Phi$ satisfies the KŁ property with exponent $\frac{1}{2}$, so we consider only the case $\vartheta \in [\frac{1}{2}, 1)$. Let

$$
\begin{aligned}
T^l = \sum_{k=l}^{\infty} & \left( \sqrt{\mathbb{E}\left\|\widetilde{x}^{k+1} - \widetilde{x}^k\right\|^2} + \sqrt{\mathbb{E}\left\|x^{k+1} - x^k\right\|^2} + \sqrt{\mathbb{E}\left\|\widetilde{x}^{k+1} - x^k\right\|^2} + \sqrt{\mathbb{E}\left\|x^{k+1} - \widetilde{x}^k\right\|^2} \right. \\
& \left. + \sqrt{\mathbb{E}\left\|\widetilde{x}^k - x^k\right\|^2} + \sqrt{\mathbb{E}\left\|\nabla f(x^k) - m^k\right\|^2} + \sqrt{\mathbb{E}\left\|\nabla x^k - x^{k-1}\right\|^2} + \sqrt{\mathbb{E}\left\|m^{k+1}\right\|^2} \right).
\end{aligned}
$$

Substituting the desingularizing function $\varphi(r) = ar^{1-\vartheta}$ into (A.53), let $i \to \infty$, then we have

$$
\begin{aligned}
T^l \le & \sqrt{\mathbb{E}\left\|\widetilde{x}^l - \widetilde{x}^{l-1}\right\|^2} + \sqrt{\mathbb{E}\left\|x^l - x^{l-1}\right\|^2} + \sqrt{\mathbb{E}\left\|\widetilde{x}^l - x^{l-1}\right\|^2} + \sqrt{\mathbb{E}\left\|x^l - \widetilde{x}^{l-1}\right\|^2} \\
& + \sqrt{\mathbb{E}\left\|\widetilde{x}^{l-1} - x^{l-1}\right\|^2} + \sqrt{\mathbb{E}\left\|\nabla f(x^{l-1}) - m^{l-1}\right\|^2} + \sqrt{\mathbb{E}\left\|x^{l-1} - x^{l-2}\right\|^2} + \sqrt{\mathbb{E}\left\|m^l\right\|^2} \\
& + \frac{2\sqrt{n}}{K\rho}\sqrt{\mathbb{E}[\Upsilon_{l-1}]} + a\kappa(\mathbb{E}[F^l - \Phi_l^*])^{1-\vartheta},
\end{aligned}
$$

$$(A.54)$$

where $\kappa = \frac{4K}{A}$. Because $G^l = \Phi(\theta^l) + \mathcal{O}\left(\left\|\nabla f(x^l) - m^l\right\|^2 + \left\|\widetilde{x}^l - x^l\right\|^2 + \left\|m^l\right\|^2 + \left\|x^l - x^{l-1}\right\|^2 + \Upsilon_l\right)$, we can rewrite the final term as $\Phi(\theta^l) - \Phi_l^*$.

$$(\mathbb{E}[G^l - \Phi_l^*])^{1-\vartheta}$$

$$= \left( \mathbb{E}\left[ \Phi(\theta^l) - \Phi_l^* + \frac{4s}{\rho}\Upsilon_l + \left(M - 8s\left(2\underline{\gamma}^2 - 4\underline{\gamma} + 3\right)\right)\left\| \nabla f(x^l) - m^l \right\|^2 + \left( \frac{\sqrt{\widehat{\pi}_l} + \varepsilon}{2\overline{\alpha}} + \frac{\sqrt{\widehat{\pi}_l} + \varepsilon}{2\alpha} \right. \right. \right.$$

$$\left. \left. \left. -\tau - \frac{1}{2s} \right)\left\| \widetilde{x}^l - x^l \right\|^2 + \left( \frac{D(\mu^l)^2}{(1 - \mu^l)^2} - Z \right)\left\| m^l \right\|^2 + H\left\| x^l - x^{l-1} \right\|^2 \right] \right)^{1-\vartheta}$$

$$\overset{(1)}{\leq} \left( \mathbb{E}[\Phi(\theta^l) - \Phi_l^*] \right)^{1-\vartheta} + \left( \frac{4s}{\rho}\mathbb{E}[\Upsilon_l] \right)^{1-\vartheta} + \left( \left(M - 8s\left(2\underline{\gamma}^2 - 4\underline{\gamma} + 3\right)\right)\left\| \nabla f(x^l) - m^l \right\|^2 \right)^{1-\vartheta}$$

$$+ \left( \left( \frac{\sqrt{\widehat{\pi}_l} + \varepsilon}{2\overline{\alpha}} + \frac{\sqrt{\widehat{\pi}_l} + \varepsilon}{2\alpha} - \tau - \frac{1}{2s} \right)\left\| \widetilde{x}^l - x^l \right\|^2 \right)^{1-\vartheta} + \left( \left( \frac{D(\mu^l)^2}{(1 - \mu^l)^2} - Z \right)\left\| m^l \right\|^2 \right)^{1-\vartheta}$$

$$+ \left( H\left\| x^l - x^{l-1} \right\|^2 \right)^{1-\vartheta}. \tag{A.55}$$

Inequality (1) is due to the fact that $(a + b)^{1-\vartheta} \leq a^{1-\vartheta} + b^{1-\vartheta}$. Applying the KŁ inequality (13),

$$a\kappa \left( \mathbb{E}[\Phi(\theta^l) - \Phi_l^*] \right)^{1-\vartheta} \leq a\kappa_1 \left( \mathbb{E}\left\| \xi^l \right\| \right)^{\frac{1-\vartheta}{\vartheta}} \tag{A.56}$$

for all $\xi^l \in \partial\Phi(\theta^l)$ and we have absorbed the constant $\kappa$ into $\kappa_1$. Inequality (A.44) provides a bound on the norm of the subgradient:

$$\left( \mathbb{E}\left\| \xi^l \right\| \right)^{\frac{1-\vartheta}{\vartheta}}$$

$$\leq \left( \varrho \left( \sqrt{\mathbb{E}\left\| \widetilde{x}^l - \widetilde{x}^{l-1} \right\|^2} + \sqrt{\mathbb{E}\left\| x^l - x^{l-1} \right\|^2} + \sqrt{\mathbb{E}\left\| \widetilde{x}^l - x^{l-1} \right\|^2} + \sqrt{\mathbb{E}\left\| x^l - \widetilde{x}^{l-1} \right\|^2} \right. \right.$$

$$\left. + \sqrt{\mathbb{E}\left\| \widetilde{x}^{l-1} - x^{l-1} \right\|^2} + \sqrt{\mathbb{E}\left\| \nabla f(x^{l-1}) - m^{l-1} \right\|^2} + \sqrt{\mathbb{E}\left\| x^{l-1} - x^{l-2} \right\|^2} + \sqrt{\mathbb{E}\left\| m^l \right\|^2} \right)$$

$$\left. + \sqrt{n\mathbb{E}[\Upsilon_{l-1}]} \right)^{\frac{1-\vartheta}{\vartheta}}.$$

Let

$$\Theta^l = \varrho \left( \sqrt{\mathbb{E}\left\| \widetilde{x}^l - \widetilde{x}^{l-1} \right\|^2} + \sqrt{\mathbb{E}\left\| x^l - x^{l-1} \right\|^2} + \sqrt{\mathbb{E}\left\| \widetilde{x}^l - x^{l-1} \right\|^2} + \sqrt{\mathbb{E}\left\| x^l - \widetilde{x}^{l-1} \right\|^2} \right.$$

$$\left. + \sqrt{\mathbb{E}\left\| \widetilde{x}^{l-1} - x^{l-1} \right\|^2} + \sqrt{\mathbb{E}\left\| \nabla f(x^{l-1}) - m^{l-1} \right\|^2} + \sqrt{\mathbb{E}\left\| x^{l-1} - x^{l-2} \right\|^2} + \sqrt{\mathbb{E}\left\| m^l \right\|^2} \right)$$

$$+ \sqrt{n\mathbb{E}[\Upsilon_{l-1}]}.$$

Therefore, it follows from (A.54)-(A.56) that

$$T^l \leq \sqrt{\mathbb{E}\left\| \widetilde{x}^l - \widetilde{x}^{l-1} \right\|^2} + \sqrt{\mathbb{E}\left\| x^l - x^{l-1} \right\|^2} + \sqrt{\mathbb{E}\left\| \widetilde{x}^l - x^{l-1} \right\|^2} + \sqrt{\mathbb{E}\left\| x^l - \widetilde{x}^{l-1} \right\|^2}$$

$$+ \sqrt{\mathbb{E}\left\| \widetilde{x}^{l-1} - x^{l-1} \right\|^2} + \sqrt{\mathbb{E}\left\| \nabla f(x^{l-1}) - m^{l-1} \right\|^2} + \sqrt{\mathbb{E}\left\| x^{l-1} - x^{l-2} \right\|^2} + \sqrt{\mathbb{E}\left\| m^l \right\|^2}$$

$$+ \frac{2\sqrt{n}}{K\rho}\sqrt{\mathbb{E}[\Upsilon_{l-1}]} + a\kappa_1\Theta_l^{\frac{1-\vartheta}{\vartheta}} + a\kappa \left( \frac{4s}{\rho}\mathbb{E}[\Upsilon_l] \right)^{1-\vartheta} + a\kappa \left( \left(M - 8s\left(2\underline{\gamma}^2 - 4\underline{\gamma} + 3\right)\right) \right.$$

$$\left. \left\| \nabla f(x^l) - m^l \right\|^2 \right)^{1-\vartheta} + a\kappa \left( \left( \frac{\sqrt{\widehat{\pi}_l} + \varepsilon}{2\overline{\alpha}} + \frac{\sqrt{\widehat{\pi}_l} + \varepsilon}{2\alpha} - \tau - \frac{1}{2s} \right)\left\| \widetilde{x}^l - x^l \right\|^2 \right)^{1-\vartheta}$$

$$+ a\kappa \left( \left( \frac{D(\mu^l)^2}{(1 - \mu^l)^2} - Z \right)\left\| m^l \right\|^2 \right)^{1-\vartheta} + a\kappa \left( H\left\| x^l - x^{l-1} \right\|^2 \right)^{1-\vartheta}. \tag{A.57}$$

(i) If $\vartheta = \frac{1}{2}$, then $\left(\mathbb{E}\left\|\xi^l\right\|\right)^{\frac{1-\vartheta}{\vartheta}} = \mathbb{E}\left\|\xi^l\right\|$. Then (A.57) gives

$$
\begin{aligned}
T^l \leq & \sqrt{\mathbb{E}\left\|\widetilde{x}^l - \widetilde{x}^{l-1}\right\|^2} + \sqrt{\mathbb{E}\left\|x^l - x^{l-1}\right\|^2} + \sqrt{\mathbb{E}\left\|\widetilde{x}^l - x^{l-1}\right\|^2} + \sqrt{\mathbb{E}\left\|x^l - \widetilde{x}^{l-1}\right\|^2} \\
& + \sqrt{\mathbb{E}\left\|\widetilde{x}^{l-1} - x^{l-1}\right\|^2} + \sqrt{\mathbb{E}\left\|\nabla f(x^{l-1}) - m^{l-1}\right\|^2} + \sqrt{\mathbb{E}\left\|x^{l-1} - x^{l-2}\right\|^2} + \sqrt{\mathbb{E}\left\|m^l\right\|^2} \\
& + \frac{2\sqrt{n}}{K\rho}\sqrt{\mathbb{E}[\Upsilon_{l-1}]} + a\kappa_1\left(\varrho\left(\sqrt{\left\|\widetilde{x}^l - \widetilde{x}^{l-1}\right\|^2} + \sqrt{\left\|x^l - x^{l-1}\right\|^2} + \sqrt{\left\|\widetilde{x}^l - x^{l-1}\right\|^2}\right.\right. \\
& \left. + \sqrt{\left\|x^l - \widetilde{x}^{l-1}\right\|^2} + \sqrt{\left\|\widetilde{x}^{l-1} - x^{l-1}\right\|^2} + \sqrt{\left\|\nabla f(x^{l-1}) - m^{l-1}\right\|^2} + \sqrt{\left\|x^{l-1} - x^{l-2}\right\|^2}\right. \\
& \left.\left. + \sqrt{\left\|m^l\right\|^2}\right) + \sqrt{n\mathbb{E}[\Upsilon_{l-1}]}\right) + a\kappa\sqrt{\frac{4s}{\rho}}\sqrt{\mathbb{E}[\Upsilon_l]} + a\kappa\sqrt{M - 8s\left(2\underline{\gamma}^2 - 4\underline{\gamma} + 3\right)} \\
& \sqrt{\left\|\nabla f(x^l) - m^l\right\|^2} + a\kappa\sqrt{\frac{\sqrt{\widehat{\pi}_l} + \varepsilon}{2\overline{\alpha}} + \frac{\sqrt{\widehat{\pi}_l} + \varepsilon}{2\alpha} - \tau - \frac{1}{2s}}\sqrt{\left\|\widetilde{x}^l - x^l\right\|^2} \\
& + a\kappa\sqrt{\frac{D(\mu^l)^2}{(1-\mu^l)^2} - Z}\sqrt{\left\|m^l\right\|^2} + a\kappa\sqrt{H}\sqrt{\left\|x^l - x^{l-1}\right\|^2} \\
\leq & \left(1 + a\kappa_2\left(\varrho + \sqrt{M - 8s\left(2\underline{\gamma}^2 - 4\underline{\gamma} + 3\right)} + \sqrt{\left(\frac{\sqrt{\widehat{\pi}_l} + \varepsilon}{2\overline{\alpha}} + \frac{\sqrt{\widehat{\pi}_l} + \varepsilon}{2\alpha} - \tau - \frac{1}{2s}\right)}\right.\right. \\
& \left.\left. + \sqrt{\frac{D(\mu^l)^2}{(1-\mu^l)^2} - Z} + \sqrt{H}\right)\right)\left(\sqrt{\mathbb{E}\left\|\widetilde{x}^l - \widetilde{x}^{l-1}\right\|^2} + \sqrt{\mathbb{E}\left\|x^l - x^{l-1}\right\|^2} + \sqrt{\mathbb{E}\left\|\widetilde{x}^l - x^{l-1}\right\|^2}\right. \\
& \left. + \sqrt{\mathbb{E}\left\|x^l - \widetilde{x}^{l-1}\right\|^2} + \sqrt{\mathbb{E}\left\|\widetilde{x}^{l-1} - x^{l-1}\right\|^2} + \sqrt{\mathbb{E}\left\|\nabla f(x^{l-1}) - m^{l-1}\right\|^2} + \sqrt{\mathbb{E}\left\|x^{l-1} - x^{l-2}\right\|^2}\right. \\
& \left. + \sqrt{\mathbb{E}\left\|m^l\right\|^2}\right) + \left(\frac{2\sqrt{n}}{K\rho} + a\kappa_2\sqrt{n}\right)\sqrt{\mathbb{E}[\Upsilon_{l-1}]} + a\kappa_2\sqrt{\frac{4s}{\rho}}\sqrt{\mathbb{E}[\Upsilon_l]},
\end{aligned}
$$
(A.58)

where $\kappa_2 = \max\{\kappa_1, \kappa\}$. Using (A.45), we have that, for any constant $c > 0$,

$$
0 \leq -c\sqrt{\mathbb{E}[\Upsilon_k]} + c\left(1 - \frac{\rho}{2}\right)\sqrt{\mathbb{E}[\Upsilon_{k-1}]} + c\sqrt{V_\Upsilon}\left(\sqrt{\mathbb{E}\left\|\nabla f(x^k) - m^k\right\|^2} + \sqrt{\mathbb{E}\left\|x^k - x^{k-1}\right\|^2}\right).
$$

Combining this inequality with (A.58),

$$
\begin{aligned}
T^l \leq & \left(1 + a\kappa_2\left(\varrho + \sqrt{M - 8s\left(2\underline{\gamma}^2 - 4\underline{\gamma} + 3\right)} + \sqrt{\left(\frac{\sqrt{\widehat{\pi}_l} + \varepsilon}{2\overline{\alpha}} + \frac{\sqrt{\widehat{\pi}_l} + \varepsilon}{2\alpha} - \tau - \frac{1}{2s}\right)}\right.\right. \\
& \left.\left. + \sqrt{\frac{D(\mu^l)^2}{(1-\mu^l)^2} - Z + \sqrt{H} + c\sqrt{V_\Upsilon}}\right)\right)\left(\sqrt{\mathbb{E}\left\|\widetilde{x}^l - \widetilde{x}^{l-1}\right\|^2} + \sqrt{\mathbb{E}\left\|x^l - x^{l-1}\right\|^2} + \sqrt{\mathbb{E}\left\|\widetilde{x}^l - x^{l-1}\right\|^2}\right. \\
& \left. + \sqrt{\mathbb{E}\left\|x^l - \widetilde{x}^{l-1}\right\|^2} + \sqrt{\mathbb{E}\left\|\widetilde{x}^{l-1} - x^{l-1}\right\|^2} + \sqrt{\mathbb{E}\left\|\nabla f(x^{l-1}) - m^{l-1}\right\|^2} + \sqrt{\mathbb{E}\left\|x^{l-1} - x^{l-2}\right\|^2}\right. \\
& \left. + \sqrt{\mathbb{E}\left\|m^l\right\|^2}\right) + c\left(1 - \frac{\rho}{2} + \frac{2\sqrt{n}}{K\rho c} + \frac{a\kappa_2\sqrt{n}}{c}\right)\sqrt{\mathbb{E}[\Upsilon_{l-1}]} - c\left(1 - \frac{a\kappa_2}{c}\sqrt{\frac{4s}{\rho}}\right)\sqrt{\mathbb{E}[\Upsilon_l]}.
\end{aligned}
$$

Defining

$$
\begin{aligned}
B = & 1 + a\kappa_2\left(\varrho + \sqrt{M - 8s\left(2\underline{\gamma}^2 - 4\underline{\gamma} + 3\right)} + \sqrt{\left(\frac{\sqrt{\widehat{\pi}_l} + \varepsilon}{2\overline{\alpha}} + \frac{\sqrt{\widehat{\pi}_l} + \varepsilon}{2\alpha} - \tau - \frac{1}{2s}\right)}\right. \\
& \left. + \sqrt{\frac{D(\mu^l)^2}{(1-\mu^l)^2} - Z + \sqrt{H} + c\sqrt{V_\Upsilon}}\right),
\end{aligned}
$$

we have shown

$$
T^l + c\left(1 - \frac{a\kappa_2}{c}\sqrt{\frac{4s}{\rho}}\right)\sqrt{\mathbb{E}[\Upsilon_l]} \leq B\left(T^{l-1} - T^l\right) + c\left(1 - \frac{\rho}{2} + \frac{2\sqrt{n}}{K\rho c} + \frac{a\kappa_2\sqrt{n}}{c}\right)\sqrt{\mathbb{E}[\Upsilon_{l-1}]}.
$$

Then, we get

$$(1+B)T^l + c\left(1 - \frac{a\kappa_2}{c}\sqrt{\frac{4s}{\rho}}\right)\sqrt{\mathbb{E}[\Upsilon_l]} \le BT^{l-1} + c\left(1 - \frac{\rho}{2} + \frac{2\sqrt{n}}{K\rho c} + \frac{a\kappa_2\sqrt{n}}{c}\right)\sqrt{\mathbb{E}[\Upsilon_{l-1}]}.$$

This implies

$$T^l + \sqrt{\mathbb{E}[\Upsilon_l]} \le \max\left\{\frac{B}{1+B}, \left(1 - \frac{\rho}{2} + \frac{2\sqrt{n}}{K\rho c} + \frac{a\kappa_2\sqrt{n}}{c}\right)\left(1 - \frac{a\kappa_2}{c}\sqrt{\frac{4s}{\rho}}\right)^{-1}\right\}\left(T^{l-1} + \sqrt{\mathbb{E}[\Upsilon_{l-1}]}\right).$$

For large $c$, the second coefficient in the above expression approaches $1 - \frac{\rho}{2}$. So there exist $\zeta \in [1-\rho, 1)$ such that

$$\sum_{k=l}^{\infty}\sqrt{\mathbb{E}\left\|\widetilde{x}^k - \widetilde{x}^{k-1}\right\|^2} \le \tau^k\left(T^0 + \sqrt{\mathbb{E}[\Upsilon_0]}\right) \le d_1\tau^k$$

for some constnt $d_1$. Then using the fact that $\mathbb{E}\left\|\widetilde{x}^l - \widetilde{x}^*\right\| = \mathbb{E}\left\|\sum_{k=l+1}^{\infty}(\widetilde{x}^k - \widetilde{x}^{k-1})\right\| \le \sum_{k=l}^{\infty}\mathbb{E}\left\|\widetilde{x}^k - \widetilde{x}^{k-1}\right\|$, we proves claim (i).

(ii) Suppose $\vartheta \in (\frac{1}{2}, 1)$. Each term on the right side of (A.57) converges to zero, but at different rates. Because

$$\Theta^l = \mathcal{O}\left(\sqrt{\mathbb{E}\left\|\widetilde{x}^l - \widetilde{x}^{l-1}\right\|^2} + \sqrt{\mathbb{E}\left\|x^l - x^{l-1}\right\|^2} + \sqrt{\mathbb{E}\left\|\widetilde{x}^l - x^{l-1}\right\|^2} + \sqrt{\mathbb{E}\left\|x^l - \widetilde{x}^{l-1}\right\|^2}\right.$$

$$\left. + \sqrt{\mathbb{E}\left\|\widetilde{x}^{l-1} - x^{l-1}\right\|^2} + \sqrt{\mathbb{E}\left\|\nabla f(x^{l-1}) - m^{l-1}\right\|^2} + \sqrt{\mathbb{E}\left\|x^{l-1} - x^{l-2}\right\|^2} + \sqrt{\mathbb{E}\left\|m^l\right\|^2}\right.$$

$$\left. + \sqrt{n\mathbb{E}[\Upsilon_{l-1}]}\right)$$

and $\vartheta$ satisfies $\frac{1-\vartheta}{\vartheta} < 1$, the term $\Theta_l^{\frac{1-\vartheta}{\vartheta}}$ dominates the first five terms on the right side of (A.57) for large $l$. Also, because $\frac{1-\vartheta}{2\vartheta} < 1 - \vartheta$, $\Theta_l^{\frac{1-\vartheta}{\vartheta}}$ dominates the final four terms as well. Combining these facts, there exists a natural number $M_1$ such that for all $l \ge M_1$,

$$T^l \le P\Theta^l \tag{A.59}$$

for some constant $P > (aC)^{\frac{\vartheta}{1-\vartheta}}$. The bound of (A.46) implies

$$2\sqrt{n\mathbb{E}[\Upsilon_{l-1}]} \le \frac{4\sqrt{n}}{\rho}\left(\sqrt{\mathbb{E}[\Upsilon_{l-1}]} - \sqrt{\mathbb{E}[\Upsilon_l]} + \sqrt{V_\Upsilon}\left(\sqrt{\mathbb{E}\left\|\nabla f(x^l) - m^l\right\|^2} + \sqrt{\mathbb{E}\left\|x^l - x^{l-1}\right\|^2}\right)\right).$$

Therefore,

$$\Theta^l$$

$$= \varrho\left(\sqrt{\mathbb{E}\left\|\widetilde{x}^l - \widetilde{x}^{l-1}\right\|^2} + \sqrt{\mathbb{E}\left\|x^l - x^{l-1}\right\|^2} + \sqrt{\mathbb{E}\left\|\widetilde{x}^l - x^{l-1}\right\|^2} + \sqrt{\mathbb{E}\left\|x^l - \widetilde{x}^{l-1}\right\|^2}\right.$$

$$\left. + \sqrt{\mathbb{E}\left\|\widetilde{x}^{l-1} - x^{l-1}\right\|^2} + \sqrt{\mathbb{E}\left\|\nabla f(x^{l-1}) - m^{l-1}\right\|^2} + \sqrt{\mathbb{E}\left\|x^{l-1} - x^{l-2}\right\|^2} + \sqrt{\mathbb{E}\left\|m^l\right\|^2}\right)$$

$$+ \left(2\sqrt{n\mathbb{E}[\Upsilon_{l-1}]} - \sqrt{n\mathbb{E}[\Upsilon_{l-1}]}\right)$$

$$\le \left(\varrho + \frac{4\sqrt{nV_\Upsilon}}{\rho}\right)\left(\sqrt{\mathbb{E}\left\|\widetilde{x}^l - \widetilde{x}^{l-1}\right\|^2} + \sqrt{\mathbb{E}\left\|x^l - x^{l-1}\right\|^2} + \sqrt{\mathbb{E}\left\|\widetilde{x}^l - x^{l-1}\right\|^2} + \sqrt{\mathbb{E}\left\|x^l - \widetilde{x}^{l-1}\right\|^2}\right.$$

$$\left. + \sqrt{\mathbb{E}\left\|\widetilde{x}^{l-1} - x^{l-1}\right\|^2} + \sqrt{\mathbb{E}\left\|\nabla f(x^{l-1}) - m^{l-1}\right\|^2} + \sqrt{\mathbb{E}\left\|x^{l-1} - x^{l-2}\right\|^2} + \sqrt{\mathbb{E}\left\|m^l\right\|^2}\right)$$

$$+ \frac{4\sqrt{n}}{\rho}\left(\sqrt{\mathbb{E}[\Upsilon_{l-1}]} - \sqrt{\mathbb{E}[\Upsilon_l]}\right) - \sqrt{n\mathbb{E}[\Upsilon_{l-1}]}.$$

$$\tag{A.60}$$

Furthermore, because $\frac{\vartheta}{1-\vartheta} > 1$ and $\mathbb{E}[\Upsilon_l] \to 0$, for large enough $l$, we have $\left(\sqrt{\mathbb{E}[\Upsilon_l]}\right)^{\frac{\vartheta}{1-\vartheta}} \ll \sqrt{\mathbb{E}[\Upsilon_l]}$. This ensures that there exists a natural number $M_2$ such that for every $l \ge M_2$,

$$\left(\frac{4\sqrt{n}(1-\rho/4)}{\rho(\varrho+4\sqrt{nV_\Upsilon}/\rho)}\sqrt{\mathbb{E}[\Upsilon_l]}\right)^{\frac{\vartheta}{1-\vartheta}} \leq P\sqrt{n\mathbb{E}[\Upsilon_l]}. \tag{A.61}$$

The constant appearing on the left was chosen to simplify later arguments. Therefore, (A.59) implies

$$\left(T^l + \frac{4\sqrt{n}(1-\rho/4)}{\rho(p+4\sqrt{nV_\Upsilon}/\rho)}\sqrt{\mathbb{E}[\Upsilon_l]}\right)^{\frac{\vartheta}{1-\vartheta}}$$

$$\overset{(1)}{\leq} \frac{2^{\frac{\vartheta}{1-\vartheta}}}{2}\left(T^l\right)^{\frac{\vartheta}{1-\vartheta}} + \frac{2^{\frac{\vartheta}{1-\vartheta}}}{2}\left(\frac{4\sqrt{n}(1-\rho/4)}{\rho(p+4\sqrt{nV_\Upsilon}/\rho)}\sqrt{\mathbb{E}[\Upsilon_l]}\right)^{\frac{\vartheta}{1-\vartheta}} \overset{(2)}{\leq} \frac{2^{\frac{\vartheta}{1-\vartheta}}}{2}\left(T^l\right)^{\frac{\vartheta}{1-\vartheta}} + \frac{2^{\frac{\vartheta}{1-\vartheta}}}{2}\left(P\sqrt{n\mathbb{E}[\Upsilon_l]}\right)$$

$$\overset{(3)}{\leq} \frac{2^{\frac{\vartheta}{1-\vartheta}}}{2}\left(P\left(\varrho+\frac{4\sqrt{nV_\Upsilon}}{\rho}\right)\left(\sqrt{\mathbb{E}\left\|\widetilde{x}^l-\widetilde{x}^{l-1}\right\|^2}+\sqrt{\mathbb{E}\left\|x^l-x^{l-1}\right\|^2}+\sqrt{\mathbb{E}\left\|\widetilde{x}^l-x^{l-1}\right\|^2}\right.\right.$$

$$+\sqrt{\mathbb{E}\left\|x^l-\widetilde{x}^{l-1}\right\|^2}+\sqrt{\mathbb{E}\left\|\widetilde{x}^{l-1}-x^{l-1}\right\|^2}+\sqrt{\mathbb{E}\left\|\nabla f(x^{l-1})-m^{l-1}\right\|^2}+\sqrt{\mathbb{E}\left\|x^{l-1}-x^{l-2}\right\|^2}$$

$$\left.+\sqrt{\mathbb{E}\left\|m^l\right\|^2}\right)+\frac{4\sqrt{n}P}{\rho}\left(\sqrt{\mathbb{E}[\Upsilon_{l-1}]}-\sqrt{\mathbb{E}[\Upsilon_l]}\right)-P\sqrt{n\mathbb{E}[\Upsilon_l]}\right)+\frac{2^{\frac{\vartheta}{1-\vartheta}}}{2}\left(P\sqrt{n\mathbb{E}[\Upsilon_l]}\right)$$

$$\leq \frac{2^{\frac{\vartheta}{1-\vartheta}}}{2}\left(P\left(\varrho+\frac{4\sqrt{n}\sqrt{V_\Upsilon}}{\rho}\right)\left(\sqrt{\mathbb{E}\left\|\widetilde{x}^l-\widetilde{x}^{l-1}\right\|^2}+\sqrt{\mathbb{E}\left\|x^l-x^{l-1}\right\|^2}+\sqrt{\mathbb{E}\left\|\widetilde{x}^l-x^{l-1}\right\|^2}\right.\right.$$

$$+\sqrt{\mathbb{E}\left\|x^l-\widetilde{x}^{l-1}\right\|^2}+\sqrt{\mathbb{E}\left\|\widetilde{x}^{l-1}-x^{l-1}\right\|^2}+\sqrt{\mathbb{E}\left\|\nabla f(x^{l-1})-m^{l-1}\right\|^2}+\sqrt{\mathbb{E}\left\|x^{l-1}-x^{l-2}\right\|^2}$$

$$\left.\left.+\sqrt{\mathbb{E}\left\|m^l\right\|^2}\right)+\frac{4\sqrt{n}P(1-\rho/4)}{\rho}\left(\left(\sqrt{\mathbb{E}[\Upsilon_{l-1}]}-\sqrt{\mathbb{E}[\Upsilon_l]}\right)\right)\right).$$

Here, (1) follows by convexity of the function $x^{\frac{\vartheta}{1-\vartheta}}$ for $\vartheta \in [1/2,1)$ and $x \geq 0$, (2) is (A.61), and (3) is (A.59) combined with (A.60). We absorb the constant $\frac{2^{\frac{\vartheta}{1-\vartheta}}}{2}$ into $P$. Define

$$S^l = T^l + \frac{4\sqrt{n}(1-\rho/4)}{\rho(\varrho+4\sqrt{nV_\Upsilon}/\rho)}\sqrt{\mathbb{E}[\Upsilon_l]}.$$

$S^l$ is bounded for all $l$ because $\sum_{k=l}^\infty \sqrt{\mathbb{E}\left\|\widetilde{x}^{k+1}-\widetilde{x}^k\right\|^2}$ and $\sum_{k=l}^\infty \sqrt{\mathbb{E}\left\|\widetilde{x}^k-x^k\right\|^2}$ are bounded by (A.54). Hence, we have shown

$$S_l^{\frac{\vartheta}{1-\vartheta}} \leq P\left(p+\frac{4\sqrt{nV_\Upsilon}}{\rho}\right)(S^{l-1}-S^l). \tag{A.62}$$

The rest of the proof is almost the same as it mentioned in (Driggs et al., 2021; Attouch & Bolte, 2007). We omit the proof here.

(iii) When $\vartheta = 0$, the KŁ property (13) implies that exactly one of the following two scenarios holds: either $\mathbb{E}[\Phi(\widetilde{x}^k)] \neq \Phi_k^*$ and

$$0 < C \leq \mathbb{E}\left\|\xi^k\right\|, \ \forall \xi^k \in \partial\Phi(\widetilde{x}^k) \tag{A.63}$$

or $\mathbb{E}[\Phi(\widetilde{x}^k)] = \Phi_k^*$. We show that the above inequality can hold only for a finite number of iterations.

Using the subgradient bound (A.34), the first scenario implies

$$C^2 \leq \left( \mathbb{E} \left\| \xi^k \right\| \right)^2$$

$$\leq \left( \varrho \left( \mathbb{E} \left\| \widetilde{x}^k - \widetilde{x}^{k-1} \right\| + \mathbb{E} \left\| x^k - x^{k-1} \right\| + \mathbb{E} \left\| \widetilde{x}^k - x^{k-1} \right\| + \mathbb{E} \left\| x^k - \widetilde{x}^{k-1} \right\| + \mathbb{E} \left\| \widetilde{x}^{k-1} - x^{k-1} \right\| \right.\right.$$

$$\left.\left. + \mathbb{E} \left\| \nabla f(x^{k-1}) - m^{k-1} \right\| + \mathbb{E} \left\| x^{k-1} - x^{k-2} \right\| + \mathbb{E} \left\| m^k \right\| \right) + \Gamma_{k-1} \right)^2$$

$$\leq 9\varrho^2 \left( \mathbb{E} \left\| \widetilde{x}^k - \widetilde{x}^{k-1} \right\| \right)^2 + 9\varrho^2 \left( \mathbb{E} \left\| x^k - x^{k-1} \right\| \right)^2 + 9\varrho^2 \left( \mathbb{E} \left\| \widetilde{x}^k - x^{k-1} \right\| \right)^2 + 9\varrho^2 \left( \mathbb{E} \left\| x^k - \widetilde{x}^{k-1} \right\| \right)^2$$

$$+ 9\varrho^2 \left( \mathbb{E} \left\| \widetilde{x}^{k-1} - x^{k-1} \right\| \right)^2 + 9\varrho^2 \left( \mathbb{E} \left\| \nabla f(x^{k-1}) - m^{k-1} \right\| \right)^2 + 9\varrho^2 \left( \mathbb{E} \left\| x^{k-1} - x^{k-2} \right\| \right)^2$$

$$+ 9\varrho^2 \left( \mathbb{E} \left\| m^k \right\| \right)^2 + 9(\mathbb{E}[\Gamma_{k-1}])^2$$

$$\leq 9\varrho^2 \left( \mathbb{E} \left\| \widetilde{x}^k - \widetilde{x}^{k-1} \right\| \right)^2 + 9\varrho^2 \left( \mathbb{E} \left\| x^k - x^{k-1} \right\| \right)^2 + 9\varrho^2 \left( \mathbb{E} \left\| \widetilde{x}^k - x^{k-1} \right\| \right)^2 + 9\varrho^2 \left( \mathbb{E} \left\| x^k - \widetilde{x}^{k-1} \right\| \right)^2$$

$$+ 9\varrho^2 \left( \mathbb{E} \left\| \widetilde{x}^{k-1} - x^{k-1} \right\| \right)^2 + 9\varrho^2 \left( \mathbb{E} \left\| \nabla f(x^{k-1}) - m^{k-1} \right\| \right)^2 + 9\varrho^2 \left( \mathbb{E} \left\| x^{k-1} - x^{k-2} \right\| \right)^2$$

$$+ 9\varrho^2 \left( \mathbb{E} \left\| m^k \right\| \right)^2 + 9n\mathbb{E}[\Upsilon_{k-1}],$$

where we have used the inequality $(a_1 + a_2 + \cdots + a_t)^2 \leq t(a_1^2 + a_2^2 + \cdots + a_t^2)$ and Jensen's inequality. Applying this inequality to the decrease of $G^k$ (A.15), we obtain

$$\mathbb{E}_k G^k$$

$$\leq \mathbb{E}_k G^{k-1} - A_1 \left\| \widetilde{x}^k - \widetilde{x}^{k-1} \right\|^2 - A_2 \left\| x^k - x^{k-1} \right\|^2 - A_3 \left\| \widetilde{x}^k - x^{k-1} \right\|^2 - A_4 \left\| x^k - \widetilde{x}^{k-1} \right\|^2$$

$$- A_5 \left\| \widetilde{x}^{k-1} - x^{k-1} \right\|^2 - A_6 \left\| \nabla f(x^{k-1}) - m^{k-1} \right\|^2 - A_7 \left\| x^{k-1} - x^{k-2} \right\|^2 - A_8 \left\| m^k \right\|^2$$

$$\leq \mathbb{E}_k G^{k-1} - C^2 + \mathcal{O} \left( \left\| \widetilde{x}^k - \widetilde{x}^{k-1} \right\|^2 \right) + \mathcal{O} \left( \left\| x^k - x^{k-1} \right\|^2 \right) + \mathcal{O} \left( \left\| \widetilde{x}^k - x^{k-1} \right\|^2 \right)$$

$$+ \mathcal{O} \left( \left\| x^k - \widetilde{x}^{k-1} \right\|^2 \right) + \mathcal{O} \left( \left\| \widetilde{x}^{k-1} - x^{k-1} \right\|^2 \right) + \mathcal{O} \left( \left\| \nabla f(x^{k-1}) - m^{k-1} \right\|^2 \right)$$

$$+ \mathcal{O} \left( \left\| x^{k-1} - x^{k-2} \right\|^2 \right) + \mathcal{O} \left( \left\| m^k \right\|^2 \right) + \mathcal{O} \left( \mathbb{E}[\Upsilon_{k-1}] \right)$$

for some constant $C^2$. Because the final five terms go to zero as $k \to \infty$, there exists an index $\kappa_3$ so that the sum of these five terms is bounded above by $\frac{C^2}{2}$ for all $k \geq \kappa_3$. Therefore,

$$\mathbb{E}_k[G^k] \leq \mathbb{E}_k[G] - \frac{C^2}{2}, \ \ \forall k \geq \kappa_3.$$

Because $G^k$ is bounded below for all $k$, this inequality can only hold for $N < \infty$ steps. After $N$ steps, it is no longer possible for the bound (A.63) to hold, so it must be that $\mathbb{E}[\Phi(\widetilde{x}^k)] = \Phi_k^*$. Because $\Phi_k^* < \Phi(\widetilde{x}^*)$, $\Phi_k^* < \mathbb{E}[\Phi(\widetilde{x}^k)]$, and both $\mathbb{E}[\Phi(\widetilde{x}^k)]$, $\Phi_k^*$ converge to $\mathbb{E}[\Phi(\widetilde{x}^*)]$, we must have $\Phi_k^* = \mathbb{E}[\Phi(\widetilde{x}^k)] = \mathbb{E}[\Phi(\widetilde{x}^*)]$. $\qquad\square$

## A.4 Experimental details and additional results

Figure 4 illustrates the framework of Retinex-Net, which consists of three sequential steps for image enhancement: decomposition, adjustment, and reconstruction.

- **Decomposition**: A subnetwork (Decom-Net) decomposes the input image into reflectance and illumination components.

- **Adjustment**: An encoder-decoder based subnetwork (Enhance-Net) brightens the illumination. Multi-scale concatenation is incorporated to enable hierarchical illumination adjustment (Wei et al., 2018), while noise in the reflectance is simultaneously removed.

- **Reconstruction**: The enhanced image is generated by combining the adjusted illumination and denoised reflectance.

This structured pipeline ensures that both global illumination improvement and local detail preservation are addressed through dedicated subnetwork designs.

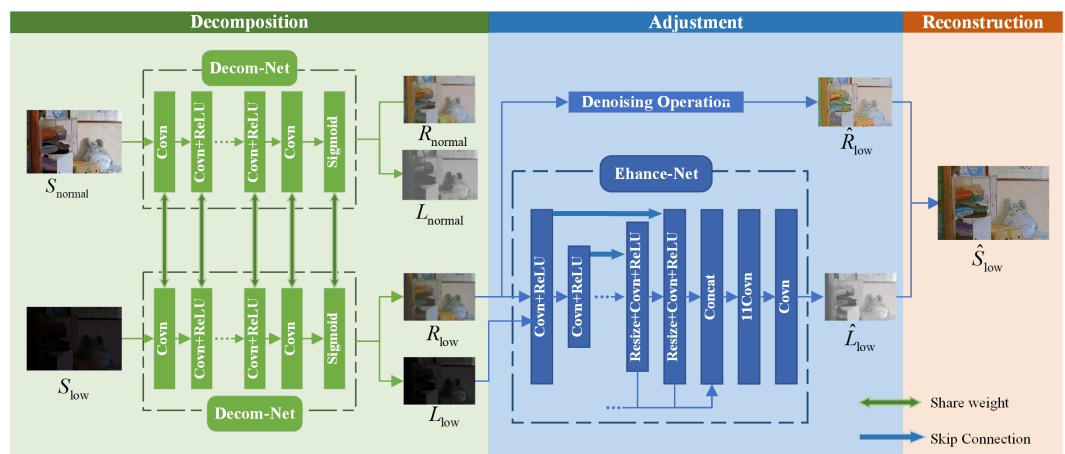

Figure 4: The training framework of Retinex-Net.

We utilize the LOw-Light paired dataset (LOL) [1], which comprises 500 low-light/normal-light image pairs. These pairs are split into 485 for training and 15 for evaluation. All raw images were resized to $400 \times 600$ and converted to Portable Network Graphics (PNG) format; Sample pairs are visualized in Figure 5. The Decom-Net architecture includes 5 convolutional layers, with ReLU activation applied between consecutive layers except for the final convolution. The Enhance-Net consists of 3 down-sampling blocks and 3 up-sampling blocks. The training protocol involves initial separate training of Decom-Net and Enhance-Net, followed by end-to-end fine-tuning using stochastic gradient descent (SGD, SAGA, SARAH) with back-propagation. Hyperparameters are set as: batch size $= 16$, patch size $= 96 \times 96$. This dataset configuration and network training strategy ensure consistent evaluation benchmarks and stable optimization of subnetwork components.

Next, we conduct two additional comparative experiments. In the first group, we compare our STNAdam-SARAH with LIME, a famous customised algorithm of LIE, by evaluating a group of low-light/normal-light image pairs from the LOL dataset. In the second group, we compare our STNAdam-SGD, STNAdam-SAGA and STNAdam-SARAH with the two Adam-type algorithms: SAdam and SNAdam.

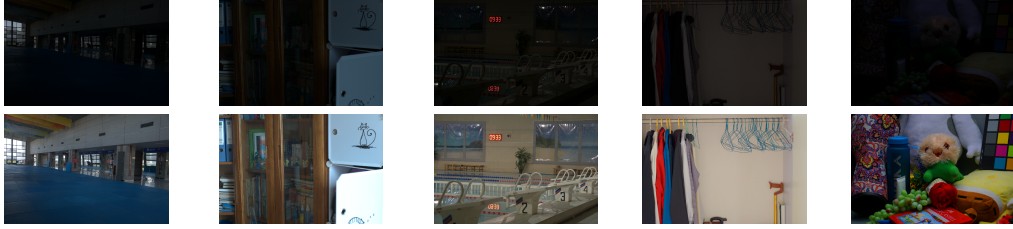

Figure 5: Several examples for low/normal-light image pairs in LOL dataset.

We first report the visual comparison results of the first group in Figure 6, which displays reflectance maps and illumination maps decomposed by STNAdam-SARAH and LIME. Then, for the second group, we compare the decomposition results of different Adam variants, shown in Figure 7. From Figures 6-7, we make the following observations.

- Our STNAdam-SARAH effectively mitigates issues of uneven illumination, whereas the LIME algorithm retains substantial illumination-related information within its reflectance map, such as ground shadows. This is because by comparing (b) with (d) in Figure 6, the reflectance of the low-light image of STNAdam-SARAH closely aligns with that of the

---

[1] https://datasets.activeloop.ai/docs/ml/datasets/lol-dataset/

normal-light image, with the primary discrepancy being amplified noise in dark regions-consistent with real-world low-light artifacts. Moreover, by comparing (c) with (e) in Figure 6, illumination maps of STNAdam-SARAH effectively capture the lightness and shadow distributions of the input images.

- By comparing (a)-(e) in Figure 7, we found that images generated by our three STNAdam algorithms exhibit exceptional contrast, characterized by well-saturated greenery and clearer spreadsheet textures, particularly with STNAdam-SARAH.

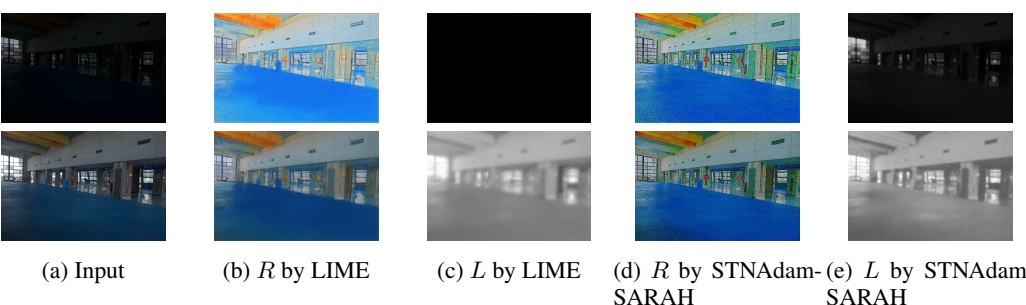

| (a) Input | (b) $R$ by LIME | (c) $L$ by LIME | (d) $R$ by STNAdam-SARAH | (e) $L$ by STNAdam-SARAH |

Figure 6: The decomposition results of STNAdam-SARAH and LIME on the LOL dataset.

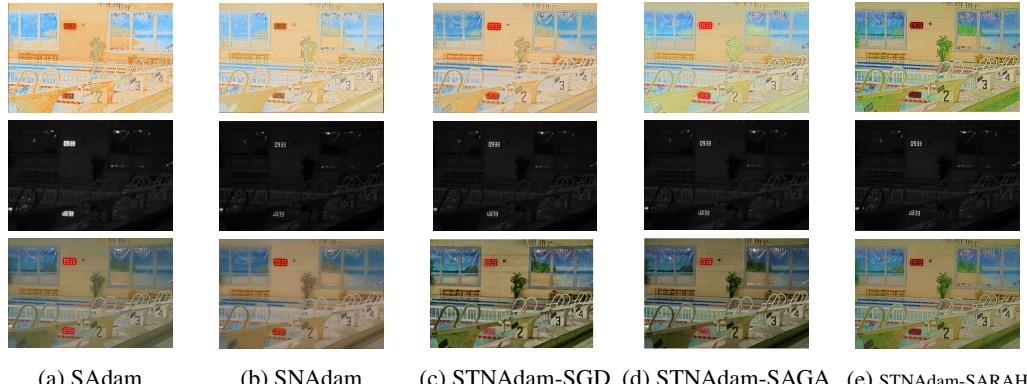

| (a) SAdam | (b) SNAdam | (c) STNAdam-SGD | (d) STNAdam-SAGA | (e) STNAdam-SARAH |

Figure 7: Comparison results of LIE with adjusted input, where $\hat{R}_{Low}$, $\hat{L}_{Low}$ and $\hat{S}_{Low}$ are reported from top row to bottom row, respectively.

In summary, these decompositions validate that our STNAdam algorithm effectively separates reflectance (content) from illumination (lighting), outperforming alternatives in preserving content consistency while isolating lighting effects.

