# OpenReview forum: "STNAdam: Stochastic Two-track Nesterov-Accelerated Adaptive Momentum Estimation"
_ICLR.cc/2026/Conference — Submitted to ICLR 2026_

### Official Review · Reviewer_csH3 · 2025-10-25

**Soundness:** 3
**Presentation:** 2
**Contribution:** 2
**Rating:** 4
**Confidence:** 4

**Summary:**

This paper proposes a Two-track Nesterov-accelerated Adaptive Momentum Estimation algorithm for solving “nonconvex + weakly convex” composite problems. For the first time, this algorithm combines Nesterov acceleration with the Adam optimization framework, introducing a two-track iteration scheme that expands the update neighborhood during the algorithm's iteration process. This enables the algorithm to find better iteration directions. Furthermore, the paper presents corresponding theoretical results and validates the algorithm's effectiveness through practical scenarios.

**Strengths:**

1. Unlike existing Adam acceleration variants, STNAdam simultaneously updates both the extrapolation trajectory and the conventional update trajectory. Through the interaction between Nesterov momentum and Adam-style adaptive adjustment, it expands the update neighborhood and continuously explores more optimal iteration directions, offering a novel approach to solving “non-convex + weakly convex” composite optimization problems. Furthermore, the algorithm permits random gradients to be supplied by any variance-reducing gradient estimator (e.g., SVRG, SAGA, SARAH), while its internal hyperparameters can be dynamically adjusted within iteration-related finite intervals. This enhances the algorithm's flexibility and applicability.

2. The paper provides a rigorous theoretical analysis based on the KL condition and experimentally validates the proposed algorithm. By comparing it with traditional methods, the effectiveness of the algorithm is demonstrated.

**Weaknesses:**

1. Due to the introduction of the two-track system, new weighting parameters have been incorporated into the algorithm. These weighting parameters are constrained by other parameters. Are all these parameters well-defined? Could there be instances where parameter values fall outside their intended domain or become meaningless?

2. Although the article provides a theoretical analytical framework, its readability is relatively low and lacks some necessary explanatory notes. It could benefit from adding more intuitive explanations or reorganizing the writing in this section.

3. The paper demonstrates convergence under the KL property, but does not explain how the “non-convex + weakly convex” conditions of the objective function satisfy the KL property.

4. The paper proposes using different variance-reduction estimators to estimate stochastic gradients, but it does not provide corresponding results for different variance estimators. Does this imply that the convergence performance of different variance estimators is largely similar?

5. There are some typos in the article: for example, in line 375 $\min_{i \in [8]} \{A_i\} > 0$; in Theorem 2 (iii), $m$ doesn't seem to work, and so on. Please carefully check the formulas in the paper.

**Questions:**

See the weakness.

---

### Official Review · Reviewer_SUNG · 2025-10-31

**Soundness:** 3
**Presentation:** 3
**Contribution:** 2
**Rating:** 4
**Confidence:** 2

**Summary:**

This paper introduces **STNAdam**, a stochastic optimization algorithm that combines *two coupled tracks*—an extrapolation (Nesterov-like) track and an adaptive (Adam-like) track—within a unified framework.
The algorithm accommodates a broad class of **variance-reduced (VR) gradient estimators** such as SGD, SAGA, and SARAH.

On the theoretical side, the authors analyze the convergence of STNAdam under the **Kurdyka–Łojasiewicz (KŁ) framework**, establishing almost-sure convergence to stationary points and characterizing the rate depending on the KŁ exponent.
On the experimental side, they apply the method to a **low-light image enhancement (LIE)** task, showing numerical and visual improvements over several Adam-style baselines.

The paper is well organized. However, the **Motivations, experimental breadth, and practical justification** are not yet strong enough to meet the ICLR acceptance bar.

**Strengths:**

- The “two-track” coupling of Nesterov acceleration and adaptive moment estimation is an interesting and potentially generalizable idea. It extends the family of Adam-like optimizers in a principled way.
- The overall flow (algorithm → theory → experiments) is easy to follow, and most proofs are well scaffolded.

**Weaknesses:**

-  While the two-track formulation is conceptually interesting, the theoretical contribution is **incremental**: the convergence analysis largely follows standard KŁ-based arguments used in many recent nonconvex adaptive/VR methods (e.g., Adan, NAdam, AdaBelief).
      No essential improvement in complexity or rate is shown compared to existing results.
      Extending from convex to weakly convex functions is straightforward under this framework and does not introduce new analytical challenges.
-  Experiments are conducted only on a **small LIE dataset (LOL)** with 500 image pairs. This cannot convincingly demonstrate effectiveness for large-scale deep learning training—the primary target scenario for Adam-type methods.
   - Missing key baselines such as **AdamW, Adan, AdaBelief, Ranger, and SGD+Momentum**.
   - The reported runtime (“Time = 10⁻⁵ s”) is suspiciously small.
   - The VR technique, while theoretically appealing, is **rarely favored in deep learning practice** due to high memory and computational costs and potential degradation of generalization. This limitation should be explicitly acknowledged and empirically examined.

**Questions:**

- What is the concrete benefit of maintaining two parallel tracks? How large is the additional computational or memory cost per iteration compared to Adam or Adan?
-    Since VR estimators are rarely adopted in deep neural network training, can the authors justify the practical utility of this framework in such settings? Have they tested plain SGD-like estimators?
-    Were all baselines tuned with comparable effort? Please include modern Adam variants and report wall-clock time, memory, and convergence curves.

---

### Official Review · Reviewer_CUKC · 2025-10-31

**Soundness:** 1
**Presentation:** 2
**Contribution:** 2
**Rating:** 2
**Confidence:** 3

**Summary:**

This paper presents a stochastic two-track adaptive optimizer that combines a regular update path and an extrapolation path, coupled through Nesterov-style momentum and Adam-like adaptive scaling. It can incorporate common variance-reduced estimators (e.g., SAGA, SARAH) to stabilize stochastic optimization. The authors provide convergence results under the KŁ framework and evaluate the method on a low-light image enhancement task, showing improvements over standard baselines.

**Strengths:**

- The idea of combining Nesterov extrapolation, adaptive scaling, and variance reduction into a unified two-track scheme is interesting and offers a fresh perspective on adaptive optimization.

- The theoretical framework is grounded in the KŁ framework and mathematically consistent within its assumptions, though some of them are quite restrictive (see Weaknesses for details).

- Overall, the integration of adaptive moment estimation with variance reduction could inspire follow-up work on designing more stable stochastic optimizers.

**Weaknesses:**

- The algorithm tracks only a scalar second moment instead of Adam’s coordinate-wise second moment, performing global RMS scaling rather than true adaptive preconditioning. This sacrifices Adam’s robustness under anisotropic or sparse gradients, making the "Adam-style" label misleading.

- The extrapolation path never evaluates gradients at the extrapolated point $\tilde{x}^k$, reusing $\nabla f(x^k)$. Without look-ahead gradients, it loses the curvature-awareness that yields Nesterov acceleration, so the “two-track” design offers little real benefit.

- Parameters $\gamma, \alpha, \lambda$ are randomly drawn from complex, state-dependent intervals that can even become invalid. It’s unclear how this randomness is handled in practice.

- The paper treats $\ell_{1/2}$-norm as weakly convex, but it is not: its curvature diverges to $-\infty$ near zero, violating the assumption used in the proofs. Hence, the theoretical guarantees do not actually cover the experimental setup.

- Lemma 5 asserts that $\mathbb{E}[ \Phi]$ preserves the KŁ property by averaging over all mini-batches, but this relies on a purely formal argument ignoring stochastic dependence and VR estimators. The assumption is unverified yet central to the convergence proof.

- The abstract claims almost sure convergence to a stationary point, but Theorems 1 and 2 prove only expected convergence. The a.s. statement overstates what the analysis actually establishes.

- The convergence proof requires $\mu < 0.5$ for positivity in descent inequalities, conflicting with practical Adam settings where the first-moment coefficient is often chosen near 0.9. The theory thus does not apply to realistic training regimes.

- Key baseline details (learning-rate schedules, warm-up, AdamW usage) are missing. Computational overhead of SAGA/SARAH is unreported, undermining fairness and reproducibility.

- The paper reports STNAdam-SGD/SAGA/SARAH results but does not isolate which component (two-track, Nesterov, VR, or adaptivity) drives the gain. Without such comparisons, the source of improvement and true novelty remain unclear.

**Questions:**

- Why does STNAdam adopt a scalar second-moment estimate instead of the standard coordinate-wise (diagonal) form of Adam? Have you verified whether using a diagonal variant changes performance or stability?

- The extrapolation track never evaluates gradients at the look-ahead point $\tilde{x}^k$. Could you explain why no "look-ahead" gradient is used, and whether computing or approximating $\nabla f (\tilde{x}^k)$ affects the claimed acceleration?

- How are the random selections of $\gamma, \lambda, \alpha$ implemented in practice? What distribution is used for sampling, and how does this affect reproducibility?

- Since $\ell_{1/2}$-norm is not weakly convex, how can the theoretical guarantees apply to experiments using it? Have you verified results using MCP and SCAD, which actually satisfy the weak convexity assumption?

- On what conditions does the expected objective $\mathbb{E}[\Phi]$ preserve the KŁ property under stochastic or variance-reduced sampling? Is this assumption formally justified for dependent estimators like SARAH/SAGA?

- The proof requires $\mu < 0.5$, but practical Adam settings typically use $\mu \approx 0.9$. Did you use $\mu <0.5$ in experiments?

- Could you provide full hyperparameter settings (learning-rate schedule, warm-up, normalization) for all baselines? What does the "Time(s)" metric in Table 2 represent, and how was it measured?

- Given the known overhead of SAGA/SARAH, how does computational complexity scale with dataset size and model dimension? Did you control for this cost when comparing runtime or performance?

- Can you provide ablation results disentangling the effects of (i) the two-track design, (ii) Nesterov momentum, (iii) variance reduction, and (iv) adaptive scaling? Without such analysis, it’s unclear what drives the observed performance improvements.

---

### Official Review · Reviewer_F9bp · 2025-11-04

**Soundness:** 2
**Presentation:** 2
**Contribution:** 2
**Rating:** 4
**Confidence:** 3

**Summary:**

The paper proposes a new method, Stochastic Two-track Nesterov-accelerated Adaptive Momentum Estimation (STNAdam), for solving “nonconvex + weakly-convex” composite optimization problems. Under the Kurdyka-Łojasiewicz (KŁ) property, the paper establishes almost-sure global convergence of STNAdam to a stationary point of the problem. Experiments on low-light image enhancement (LIE) tasks are also provided, showing the benefits of the new approach.

**Strengths:**

The main contributions of the paper are clearly presented, and the main algorithm's differences to previous approaches are nicely presented in Section 2.

I find the main idea of the paper original. It presents a novel two-track iteration framework, governed by Nesterov momentum and Adam-style adaptive conditioning interactively. As far as I am aware, I have never seen the proposed algorithms before.

**Weaknesses:**

I find the theoretical convergence analysis section quite confusing, with information that looks unnecessarily complicated for an Adam-type method.

One would expect a theorem that assumes the KL condition and smoothness to obtain the proposed result, as the abstract and intro prepare the reader for. Instead the paper introduces a bit out of the blue Assumption 1 without explanation of when this holds and why it is needed, then Lemmas 2 to 5 are presented in the main paper without explaining why this is needed for the flow of the narrative and why these are important. As far as I can see, the proofs of these lemmas (steps in the paper) use standard arguments, so i am not sure why they needed to be highlighted in the main paper and not simply the appendix.

Then Theorem 1 includes an assumption on the function $\Phi$ that does not look natural. Why this is needed for the paper? How is the associate with the main assumptions mentioned in the introduction (important information is missing).

In addition even if the paper focuses on the KL condition, this condition was never properly defined throughout the paper. The only mention of it is in Lemma 5, but without a proper definition and why one should care about this condition (problems for which this is useful to have an analysis for).

In terms of presentation, I am also not sure how Table 1 is useful. How this serves the narrative?

The numerical experiments are also far from what the rest of the paper's theory claims. How do these problems satisfy the theoretical assumptions? Is there a gap between theory and experiments there? This is fine, but it was never properly presented in the paper. From the experiments, it is also not clear to me why the new approach is better than the previous ideas for solving the same problems. Is there any intuition of why this works, or should one simply run the methods and report the results?


The authors claim that the new method adopts a novel two-track iteration framework and is essentially an enhanced version of stochastic Adam. But in the paper, this was never properly explained. The idea enhanced Adam in what sense (theory, experiments, something else)?

**Questions:**

Please see Weaknesses

---

### Meta-Review · Area_Chair_HKd4 · 2026-01-16

**Summary:**

This paper focuses on an additively composite optimization problem where we have the sum of smooth and nonconvex function $f$ and a weakly convex regularizer $g$. The authors propose an algorithm combining Nesterov acceleration, Adam-type adaptive conditioning and variance reduction and analyze the algorithm under the KL property. The reviewers pointed out several major concerns which are not addressed by the authors since a rebuttal is not submitted. Particularly, Reviewer CUKC points out that the main theoretical result is weaker than the statement (overselling), because almost sure convergence is claimed whereas only a convergence under expectation is proved. Both Reviewers F9bp and CUKC point out to the inconsistency between the theory and the empirical results and Reviewer SUNG argued that key baselines are missing. Moreover, looking at the appendix, I see that the proofs are not polished and have readability concerns. With all these concerns and the lack of a rebuttal, I recommend rejection.

**Reviewer Concerns:**

The authors did not submit a rebuttal, hence the reviewer concerns are not resolved.

**Reviewer Scores:**

The authors did not submit a rebuttal. As a result, I do not think that any of the reviewers would have changed their scores under a longer discussion period.

---

### Decision · Program_Chairs · 2026-01-26

Reject